# Learn the Time to Learn: Replay Scheduling in Continual Learning

**Marcus Klasson**                                                      *marcus.klasson@aalto.fi*
*Aalto University*

**Hedvig Kjellström**                                                          *hedvig@kth.se*
*KTH Royal Institute of Technology*

**Cheng Zhang**                                                   *cheng.zhang@microsoft.com*
*Microsoft Research*

**Reviewed on OpenReview:** *https: // openreview. net/ forum? id= Q4aAITDgdP*

## Abstract

Replay methods are known to be successful at mitigating catastrophic forgetting in continual learning scenarios despite having limited access to historical data. However, storing historical data is cheap in many real-world settings, yet replaying all historical data is often prohibited due to processing time constraints. In such settings, we propose that continual learning systems should learn the time to learn and schedule which tasks to replay at different time steps. We first demonstrate the benefits of our proposal by using Monte Carlo tree search to find a proper replay schedule, and show that the found replay schedules can outperform fixed scheduling policies when combined with various replay methods in different continual learning settings. Additionally, we propose a framework for learning replay scheduling policies with reinforcement learning. We show that the learned policies can generalize better in new continual learning scenarios compared to equally replaying all seen tasks, without added computational cost. Our study reveals the importance of learning the time to learn in continual learning, which brings current research closer to real-world needs.

## 1 Introduction

Many organizations deploying machine learning systems receive large volumes of data daily (Bailis et al., 2017; Hazelwood et al., 2018). Although all historical data are stored in the cloud in practice, retraining machine learning systems on a daily basis is prohibitive both in time and cost. In this setting, the systems often need to continuously adapt to new tasks while retaining the previously learned abilities. Continual learning (CL) methods (Delange et al., 2021; Parisi et al., 2019) address this challenge where, in particular, replay methods (Chaudhry et al., 2019; Hayes et al., 2020) have shown to be effective in achieving great prediction performance. Replay methods mitigate catastrophic forgetting by revisiting a small set of samples, which is feasible to process compared to the size of the historical data. In the traditional CL literature, replay memories are limited due to the assumption that historical data are not available. In real-world settings where historical data are always available, the requirement of small memories remains due to processing time and cost issues.

Recent research on replay-based CL has focused on the quality of memory samples (Aljundi et al., 2019b; Chaudhry et al., 2019; Nguyen et al., 2018; Rebuffi et al., 2017; Yoon et al., 2022) or data compression to increase the memory capacity (Hayes et al., 2020; Pellegrini et al., 2020). Most previous methods allocate equal memory storage space for samples from old tasks, and replay the whole memory to mitigate catastrophic forgetting. However, in life-long learning settings, this simple strategy would be inefficient as the memory must store a large number of tasks. Furthermore, the commonly used uniform selection policy of samples

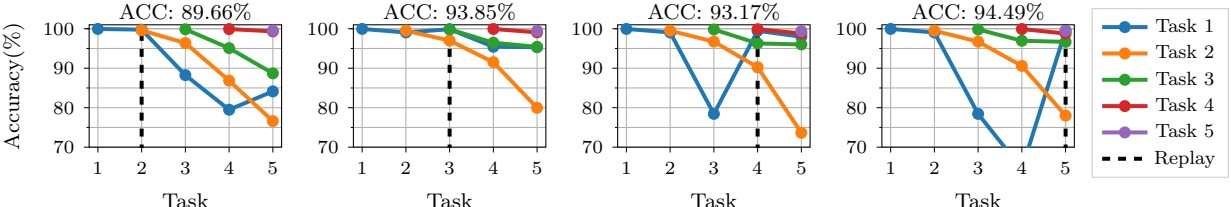

Figure 1: Task accuracies on Split MNIST (Zenke et al., 2017) when replaying only 10 samples of classes 0/1 at a single time step. The black vertical line indicates when replay is used. ACC denotes the average accuracy over all tasks after learning Task 5. Results are averaged over 5 seeds. These results show that the time to replay the previous task is critical for the final performance.

to replay ignores the time of which tasks to learn again. This stands in contrast to human learning where education methods focus on scheduling of learning new tasks and rehearsal of previous learned knowledge. For example, spaced repetition (Dempster, 1989; Ebbinghaus, 2013; Landauer & Bjork, 1977), where the time interval between rehearsal increases, has been shown to enhance memory retention over uniform rehearsal.

We argue that finding the proper schedule of which tasks to replay in fixed memory settings is critical for CL. To demonstrate our claim, we perform a simple experiment on the Split MNIST dataset (Zenke et al., 2017) where each task consists of learning the digits 0/1, 2/3, etc. arriving in sequence. Here, the replay memory contains 10 samples from task 1 and can only be replayed while learning one of the tasks. Figure 1 shows how the task performances progress over time when this memory is replayed at different tasks. In this example, the best average performance is achieved when the memory is used when learning task 5. Note that choosing different time points to replay the same memory leads to noticeably different classification performance. These results indicate that scheduling the time when to apply replay can influence the final performance significantly of a CL system.

In this paper, we propose learning the time to learn for CL systems, in which we learn replay schedules of which tasks to replay at different times inspired from human learning (Dempster, 1989). To demonstrate the benefits of replay scheduling, we perform experiments in an ideal CL environment where multiple trials are allowed to enable searching for the optimal replay schedule. We use Monte Carlo tree search (MCTS) (Coulom, 2006) to find proper replay schedules, which are evaluated by measuring the task performances of a network trained in a CL scenario in which the scheduled replay samples are used for reducing catastrophic forgetting. Furthermore, as using MCTS in realistic CL settings is infeasible, we propose a framework using reinforcement learning (RL) (Sutton & Barto, 2018) for learning replay scheduling policies. Our goal is to learn a general policy that has implicitly explored task relationships during training, such that the policy can be applied to mitigate catastrophic forgetting in new CL scenarios without additional training at test time. We evaluate the learned policy by comparing its ability to schedule the replay tasks against fixed scheduling policies, such as equally replaying all tasks. In summary, our contributions are:

- We propose a new CL setting where historical data is available while the processing time is limited, in order to adjust current CL research closer to real-world needs (Section 3.1). In this setting, we introduce replay scheduling where we learn the time of which tasks to replay (Section 3.2).

- To demonstrate the benefits of replay scheduling, we apply MCTS in an ideal CL environment where MCTS searches over a finite set of replay memories at every task (Section 3.2). We show that the found replay schedules efficiently mitigate catastrophic forgetting across multiple benchmarks for various memory selection and replay methods in various CL scenarios (Section 4.1).

- As a first step towards enabling replay scheduling for realistic CL settings, we propose an RL-based framework for learning policies for mitigating catastrophic forgetting across different CL environments (Section 3.3). We show that the learned policies can outperform equally replaying all tasks in CL scenarios with new task orders and datasets unseen during training (Section 4.2).

## 2 Related Work

In this section, we give a brief overview of various approaches in CL, especially replay methods as well as spaced repetition techniques for human CL and generalization in RL.

**Continual Learning.** Traditional CL can be divided into three main areas, namely regularization-based, architecture-based, and replay-based approaches. Regularization-based methods protect parameters influencing the performance on known tasks from wide changes and use the other parameters for learning new tasks (Adel et al., 2019; Kirkpatrick et al., 2017; Li & Hoiem, 2017; Nguyen et al., 2018; Schwarz et al., 2018; Zenke et al., 2017). Architecture-based methods mitigate catastrophic forgetting by maintaining task-specific parameters (Mallya & Lazebnik, 2018; Rusu et al., 2016; Serra et al., 2018; Xu & Zhu, 2018; Yoon et al., 2020; 2018). Replay methods mix samples from old tasks with the current dataset to mitigate catastrophic forgetting, where the replay samples are either stored in an external memory (Aljundi et al., 2019a;b; Chaudhry et al., 2019; Chrysakis & Moens, 2020; Hayes et al., 2019; 2020; Iscen et al., 2020; Isele & Cosgun, 2018; Liu et al., 2021; Rebuffi et al., 2017; Rolnick et al., 2019; Verwimp et al., 2021) or generated using a generative model (Shin et al., 2017; van de Ven & Tolias, 2018). Regularization-based approaches and dynamic architectures have been combined with replay-based approaches to methods to overcome their limitations (Buzzega et al., 2020; Chaudhry et al., 2018a;b; 2021; Douillard et al., 2020; Ebrahimi et al., 2020; Joseph & Balasubramanian, 2020; Lopez-Paz & Ranzato, 2017; Mirzadeh et al., 2021; Pan et al., 2020; Pellegrini et al., 2020; Riemer et al., 2019; von Oswald et al., 2020). Our work relates most to replay-based methods with external memory which we spend more time on describing in the next paragraph.

**Replay-based Continual Learning.** One main assumption in traditional CL has been that only a limited amount of incoming data can be stored in memory for future re-use to mitigate catastrophic forgetting of old tasks. This has led to much research effort in replay- or memory-based CL being focused on selecting high quality samples to keep in the memory (Aljundi et al., 2019b; Bang et al., 2021; Borsos et al., 2020; Chaudhry et al., 2019; Chrysakis & Moens, 2020; Hayes et al., 2019; Isele & Cosgun, 2018; Jin et al., 2021; Nguyen et al., 2018; Rebuffi et al., 2017; Sun et al., 2022; Yoon et al., 2022). An alternative to developing selection strategies has been to compress raw image data into feature representations to increase the number of memory samples that can be stored for replay (Hayes et al., 2020; Iscen et al., 2020; Pellegrini et al., 2020). More recently, methods for selecting which samples to retrieve from the memory that would most interfere with a foreseen parameter update has been proposed for mitigating catastrophic forgetting (Aljundi et al., 2019a; Shim et al., 2021). In this paper, we focus on selecting the time to replay old tasks, which has mostly been ignored in the literature. Our replay scheduling approach differs from the above mentioned works since we focus on learning to select which tasks to replay. Nevertheless, our scheduling can be combined with any replay-based method, including any memory selection and retrieval strategy.

Independently and concurrently with our work here, (Prabhu et al., 2023b;a) also focus on the setting in CL without storage constraints. In particular, Prabhu et al. (2023b) saves every incoming sample with pre-trained features and use a kNN classifier in feature space to perform fast adaptation without forgetting the old data. Prabhu et al. (2023a) study how CL methods perform under various constraints on compute and show that uniformly sampling from the historical data outperforms previous methods. In this work, we argue that using small replay memories is complimentary for reducing the amount of compute for CL methods. To effectively mitigate catastrophic forgetting with a small replay memory, motivated from human learning, we study which tasks to select for replay at different times in CL.

**Human Continual Learning.** Humans are CL systems in the sense of learning tasks and concepts sequentially. The timing of learning and rehearsal is essential for humans to memorize better (Dempster, 1989; Dunlosky et al., 2013; Willis, 2007). An example technique is spaced repetition where time intervals between rehearsal are gradually increased to improve long-term memory retention (Dempster, 1989; Ebbinghaus, 2013), which has been shown to improve memory retention better uniformly spaced rehearsal times (Hawley et al., 2008; Landauer & Bjork, 1977). Several works in CL with neural networks are inspired by human learning techniques, including spaced repetition (Amiri et al., 2017; Feng et al., 2019; Smolen et al., 2016), sleep mechanisms (Ball et al., 2020; Mallya & Lazebnik, 2018; Schwarz et al., 2018), and memory reactivation (Hayes et al., 2020; van de Ven et al., 2020). Replay scheduling is also inspired by spaced repetition, where we learn schedules of which tasks to replay at different times.

**Generalization in Reinforcement Learning.** Generalization is an active research topic in RL (Kirk et al., 2023) as RL agents tend to overfit to their training environments (Henderson et al., 2018; Zhang et al., 2018a;b). The goal is often to transfer learned policies to environments with new tasks (Finn et al., 2017; Higgins et al., 2017; Kessler et al., 2022) and action spaces (Chandak et al., 2019; 2020; Jain et al., 2020). Some approaches aim to improve generalization capabilities by generating more diverse training data (Cobbe et al., 2019; Tobin et al., 2017; Wang et al., 2020; Zhang et al., 2018a), using network regularization or inductive biases (Farebrother et al., 2018; Igl et al., 2019; Zambaldi et al., 2018), or learning dynamics models (Ball et al., 2021; Nagabandi et al., 2019). In this paper, we learn policies using RL for selecting which tasks a CL network should replay by generating training data from different CL environments. Using RL for memory management in CL has previously been studied in RMM (Liu et al., 2021) where the learned policy selects the memory partition between the old and new tasks. Our RL framework shares similarities with RMM in that our goal is to learn transferable policy without additional training cost at test time, we assume the number of tasks to be known, and that the policy is learned from generated CL environments that are different from the test environments. The main difference lies in the CL setting where we assume that historical data is accessible for replay at any time. Furthermore, while RMM keeps a partition for every seen task throughout the CL scenario, our method selects which tasks to replay at every task.

## 3 Method

Here, we describe our new problem setting of CL where historical data is available while the processing time is limited when learning new tasks. In Section 3.1 and 3.2, we present the considered problem setting, as well as our idea of learning schedules over which tasks to replay at different time steps to mitigate catastrophic forgetting. Section 3.2 also describes how we use MCTS (Coulom, 2006) to study the benefits of replay scheduling in CL. In Section 3.3, we present an RL-based framework for learning replay scheduling policies that can generalize to different CL scenarios.

### 3.1 Problem Setting

We focus on a slightly new setting in CL, where we assume that all historical data is available for mitigating catastrophic forgetting since data storage is cheap. However, as this data volume is typically huge, retraining on all historical data whenever the CL system must adapt to new tasks is impractical. Therefore, we assume there are processing time constraints which limits the system to sample a small replay memory from the historical data only once when adapting to new tasks. The challenge becomes how to select which old tasks to fill the replay memory with, such that the CL system achieves the best possible accuracy and minimize forgetting across all tasks.

The notation of our problem setting resembles the traditional CL setting for image classification. We let the network $f_{\phi}$, parameterized by $\phi$, learn $T$ tasks sequentially from the datasets $\mathcal{D}_1, \ldots, \mathcal{D}_T$ arriving one at a time. The $t$-th dataset $\mathcal{D}_t = \{(\boldsymbol{x}_t^{(i)}, y_t^{(i)})\}_{i=1}^{N_t}$ consists of $N_t$ samples where $\boldsymbol{x}_t^{(i)}$ and $y_t^{(i)}$ are the $i$-th data point and class label respectively. Furthermore, each dataset is split into a training, validation, and test set, i.e., $\mathcal{D}_t = \{\mathcal{D}_t^{(train)}, \mathcal{D}_t^{(val)}, \mathcal{D}_t^{(test)}\}$. The objective at task $t$ is to minimize the loss $\ell(f_{\phi}(\boldsymbol{x}_t), y_t)$ where $\ell(\cdot)$ is the cross-entropy loss in our case.

We assume that historical data from old tasks is accessible at any task $t$. However, due to processing time constraints, we can only use a small replay memory $\mathcal{M}$ of $M$ historical samples for replay when learning a new task. The challenge then becomes how to select the $M$ replay samples to efficiently retain knowledge of old tasks. We focus on selecting the samples on task-level by deciding on the task proportion $(p_1, \ldots, p_{t-1})$ of samples to fetch from each task, where $p_i \geq 0$ is the proportion of $M$ samples from task $i$ to place in $\mathcal{M}$ and $\sum_{i=1}^{t-1} p_i = 1$. To simplify the selection of which tasks to replay, we assume that the number of tasks $T$ to learn is known and construct a discrete set of possible task proportions to select for constructing $\mathcal{M}$.

### 3.2 Replay Scheduling in Continual Learning

In this section, we describe our setup for enabling the scheduling for selecting replay memories at different time steps. We define a replay schedule as a sequence $S = (\boldsymbol{p}_1, \ldots, \boldsymbol{p}_{T-1})$, where the task proportions

$\boldsymbol{p}_i = (p_1, \ldots, p_{T-1})$ for $1 \le i \le T-1$ are used for determining how many samples from seen tasks with which to fill the replay memory at task $i$. We construct an action space with a discrete number of choices of task proportions that can be selected at each task: At task $t$, we have $t-1$ historical tasks that we can choose samples from. We create $t-1$ bins $\boldsymbol{b}_t = [b_1, \ldots, b_{t-1}]$ and sample a task index for each bin $b_i \in \{1, \ldots, t-1\}$. The bins are treated as interchangeable and we only keep the unique choices. For example, at task 3, we have seen task 1 and 2, so the unique choices of vectors are $[1,1], [1,2], [2,2]$, where $[1,1]$ indicates that all memory samples are from task 1, $[1,2]$ indicates that half memory is from task 1 and the other half are from task etc. We count the number of occurrences of each task index in $\boldsymbol{b}_t$ and divide by $t-1$ to obtain the task proportion, i.e., $\boldsymbol{p}_t = \mathtt{bincount}(\boldsymbol{b}_t)/(t-1)$. We round the number of replay samples from task $i$, i.e., $p_i \cdot M$, up or down accordingly to keep the memory size $M$ fixed when filling the memory. From this specification, we can build a tree of different replay schedules to evaluate with the network.

Figure 2 shows an example of a replay schedule tree with Split MNIST-where the memory size is $M = 8$. Each level corresponds to a CL task, and we show some examples of possible replay memories in the tree that can be evaluated at each task. A replay schedule is represented as a path traversal of different replay memory compositions from task 1 to task 5. At task 1, the memory $\mathcal{M}_1 = \emptyset$ is empty, while $\mathcal{M}_2$ is filled with samples from task 1 at task 2. The memory $\mathcal{M}_3$ can be composed with samples from either task 1 or 2, or equally fill $\mathcal{M}_3$ with samples from both tasks. All possible paths in the tree are valid replay schedules. We show three examples of possible

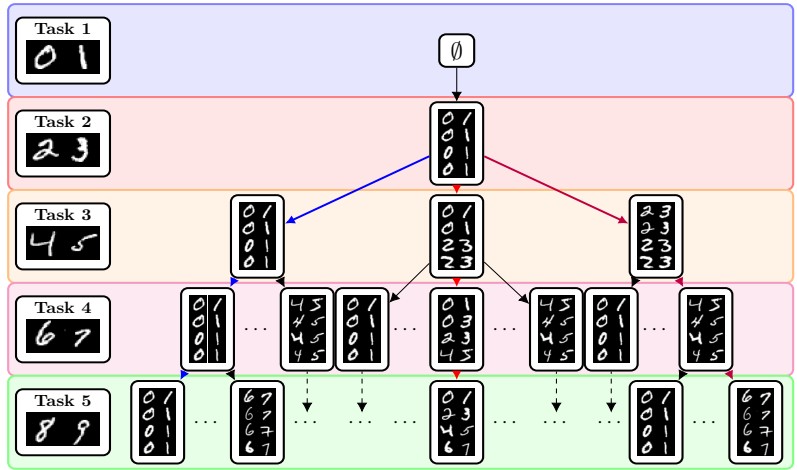

Figure 2: Tree-shaped action space of possible replay memories of size $M = 8$ at every task from the discretization method described in Section 3.2 for Split MNIST.

schedules in Figure 2 for illustration: the blue path represents a replay schedule where only task 1 samples are replayed. The red path represents using memories with equally distributed tasks, and the purple path represents a schedule where the memory is only filled with samples from the most previous task.

**Monte Carlo Tree Search for Replay Schedules.** The tree-shaped action space of task proportions grows fast with the number of tasks. This complicates studying replay scheduling in datasets with long task-horizons, where the action space is too large for using exhaustive searches. We propose to use MCTS since it has been successful in applications with large action spaces (Browne et al., 2012; Chaudhry & Lee, 2018; Silver et al., 2016). We apply MCTS in an ideal setting with multiple rollouts allowed to demonstrate that replay scheduling can be essential for the final CL performance, where MCTS concentrates the search in directions with promising outcomes in the CL environment.

Each replay memory composition in the action space corresponds to a node that MCTS can visit, as can be seen in Figure 2. At level $t$, the node $v_t$ is related to a task proportion $\boldsymbol{p}_t$ used for retrieving a replay memory composition from the historical data at task $t$. One MCTS rollout corresponds to traversing through all tree levels $1, \ldots, T$ to select the replay schedule $S$ to use during the CL training. Each task proportion $\boldsymbol{p}_t$ from every visited node is stored in $S$ during the rollout. When reaching level $T$, we start the CL training and use $S$ for constructing the replay memories at each task. Next, we briefly outline the MCTS steps for performing the search (details in Appendix A.1):

- **Selection.** During a rollout, the current node $v_t$ either moves randomly to an unvisited child, or selects the next child node $v_{t+1}$ by evaluating the Upper Confidence Tree (UCT) (Kocsis & Szepesvári, 2006) with the function from Chaudhry & Lee (2018):

$$UCT(v_t, v_{t+1}) = \max(q(v_{t+1})) + C\sqrt{2\log(n(v_t))/n(v_{t+1})}, \tag{1}$$

where $q(\cdot)$ is the reward function, $C$ the exploration constant, and $n(\cdot)$ the number of node visits.

- **Expansion.** Whenever the current node $v_t$ has unvisited child nodes, the search tree is expanded with one of the unvisited child nodes $v_{t+1}$ selected with uniform sampling.

- **Simulation and Reward.** After expansion, the succeeding nodes are selected randomly until reaching a terminal node $v_T$. The task proportions from the visited rollout nodes constitutes the replay schedule $S$. After training the network using $S$ for replay, we calculate the reward for the rollout given by $r = \frac{1}{T}\sum_{i=1}^{T} A_{T,i}^{(val)}$, where $A_{T,i}^{(val)}$ is the validation accuracy of task $i$ at task $T$.

- **Backpropagation.** Reward $r$ is backpropagated from the expanded node $v_t$ to the root $v_1$, where the reward function $q(\cdot)$ and number of visits $n(\cdot)$ are updated at each node.

### 3.3 Policy Learning Framework for Replay Scheduling

In this section, we present an RL-based framework for learning replay scheduling policies. We focus on learning a general policy that can be applied in any CL scenario to mitigate catastrophic forgetting to avoid re-training the policy for every new CL dataset. Our intuition is that there may exist general patterns regarding replay scheduling, e.g., that tasks that are harder or have been forgotten should be replayed more often. We aim to implicitly explore such task properties by using the task performances of the CL network as states for the policy to select which tasks to replay. Representing the states with task performances also enables transferring the learned policy to reduce forgetting in unseen CL environments. Next, we present our modeling approach for learning the replay scheduling policies using RL.

**CL Environment.** We model the CL environments as Markov Decision Processes (Bellman, 1957) (MDPs) where each MDP is represented as a tuple $E_i = (\mathcal{S}_i, \mathcal{A}, P_i, R_i, \mu_i, \gamma)$ consisting of the state space $\mathcal{S}_i$, action space $\mathcal{A}$, state transition probability $P_i(s'|s,a)$, reward function $R_i(s,a)$, initial state distribution $\mu_i(s_1)$, and discount factor $\gamma$. Each environment $E_i$ contains a network $f_\phi$ and task datasets $\mathcal{D}_{1:T}$ where the $t$-th dataset is learned at time step $t$. Note that each task dataset is split into a training, validation, and test set.

- **States:** The state $s_t$ is defined as the task accuracies $A_{t,1:t}$ evaluated at task $t$, such that $s_t = [A_{t,1}, ..., A_{t,t}, 0, ..., 0]$ where zero-padding is used on future tasks. We obtain the states by evaluating the classifier on the validation datasets to avoid overfitting the policy to the training data.

- **Actions:** We use the same action space as for MCTS (see Section 3.2), such that $a_t \in \mathcal{A}$ corresponds to a task proportion $\boldsymbol{p}_t$ used for sampling the replay memory $\mathcal{M}_t$.

- **Reward:** We use a dense reward defined as the average validation accuracies at task $t$, i.e., $r_t = \frac{1}{t}\sum_{i=1}^{t} A_{t,i}^{(val)}$, to ease exploration in the action space. This is similar to previous works combining RL with CL (Liu et al., 2021; Xu & Zhu, 2018), where the goal for the agent is to maximize the task validation accuracy during an episode.

The state transition distribution $P_i(s'|s,a)$ represents the dynamics of the environment, which depend on the initialization of $f_\phi$ and the task order in $\mathcal{D}_{1:T}$.

**Training and Evaluation.** The policy interacts with the CL environments by selecting which tasks the network $f_\phi$ should replay to mitigate catastrophic forgetting. The state $s_t$ is obtained by evaluating $f_\phi$ on the validation sets $\mathcal{D}_{1:t}^{(val)}$ after learning task $t$. The action $a_t$ is selected under the policy $\pi_{\boldsymbol{\theta}}(a|s_t)$, parameterized by $\boldsymbol{\theta}$, which is converted into the task proportion $\boldsymbol{p}_t$ for sampling the replay memory $\mathcal{M}_t$ from the historical datasets. The network $f_\phi$ is trained on task $t+1$ while replaying $\mathcal{M}_t$, and we obtain the reward $r_{t+1}$ and the next state $s_{t+1}$ by evaluating $f_\phi$ on the validation sets $\mathcal{D}_{1:t+1}^{(val)}$. The collected transitions $(s_t, a_t, r_{t+1}, s_{t+1})$ are used for updating the policy, and a new episode starts after $f_\phi$ has learned the final task $T$. We let the policy interact with multiple training environments $\mathcal{E}^{(train)} = \{E_i\}_{i=1}^{K}$ sampled from a distribution of CL environments, i.e., $E_i \sim p(E)$. To generate diverse CL environments, we let each $E_i$ have different network initializations of $f_\phi$ and task orders in the datasets. Our goal is to learn a general replay scheduling policy that can be applied in new CL environments to mitigate catastrophic forgetting. Hence, in Section 4.2, we evaluate the policy in CL environments with new task orders or datasets unseen during training. The policy is applied for only a single CL episode without additional training in the test environment.

## 4  Experiments

In this section, we present the experimental results to show the importance of replay scheduling in CL. First, we demonstrate the benefits with replay scheduling by using MCTS for finding replay schedules in Section 4.1. Thereafter, we evaluate our RL-based framework using DQN (Mnih et al., 2013) and A2C (Mnih et al., 2016) for learning policies that generalize to new CL scenarios in Section 4.2. Full details on experimental settings and additional results are in Appendix C and D. Code is publicly available under the MIT license[1].

### 4.1  Results on Replay Scheduling with Monte Carlo Tree Search

In this section, we show the benefits of replay scheduling in single CL environments using MCTS. We perform extensive evaluation where we apply MCTS with different memory selection and replay methods, varying memory sizes in different CL settings (Van de Ven & Tolias, 2019), and show the potential efficiency of replay scheduling in a tiny memory setting.

**Experimental Setup.** We conduct experiments on several CL benchmark datasets: Split MNIST (LeCun et al., 1998; Zenke et al., 2017), Split FashionMNIST (Xiao et al., 2017), Split notMNIST (Bulatov, 2011), Permuted MNIST (Goodfellow et al., 2013), and Split CIFAR-100 (Krizhevsky & Hinton, 2009), and Split miniImagenet (Vinyals et al., 2016). We use a 2-layer MLP with 256 hidden units for Split MNIST, Split FashionMNIST, Split notMNIST, and Permuted MNIST. We apply the ConvNet from Schwarz et al. (2018); Vinyals et al. (2016) for Split CIFAR-100, and the reduced ResNet-18 from Lopez-Paz & Ranzato (2017) for Split miniImagenet. We use multi-head output layers and assume task labels are available at test time unless stated otherwise, except for Permuted MNIST where single-head output layer is used. We measure CL performance using ACC as the average test accuracy across tasks and BWT for forgetting (Lopez-Paz & Ranzato, 2017), i.e.,

$$\text{ACC} = \frac{1}{T}\sum_{i=1}^{T} A_{T,i} \in [0,1], \quad \text{BWT} = \frac{1}{T-1}\sum_{i=1}^{T-1} A_{T,i} - A_{i,i} \in [-1,1], \tag{2}$$

where $A_{t,i}$ is the test accuracy for task $i$ after learning task $t$. We compare MCTS to the baselines:

- **Random.** Random policy that randomly selects task proportions from the action space on how to structure the replay memory at every task.
- **Equal Task Schedule (ETS).** Policy that selects equal task proportion such that the replay memory aims to fill the memory with an equal number of samples from every seen task.
- **Heuristic Global Drop (Heur-GD).** Heuristic policy that replays tasks with validation accuracy below a certain threshold proportional to the best achieved validation accuracy on the task.

Heur-GD is based on the intuition that forgotten tasks should be replayed. The replay memory is filled with $M/k$ samples per task, where $k$ is the number of selected tasks, but skips replay if $k = 0$. MCTS and Heur-GD randomly sample 15% of the training data of each task to use for validation. For MCTS, reported results are evaluated on the test set by using replay schedules selected from the validation sets. The replay memory $\mathcal{M}_t$ of size $M$ is sampled before learning task $t$ and is replayed for every batch throughout learning task $t$. Memory sizes are set to $M = 10$ for Split MNIST, Split FashionMNIST, and Split notMNIST, and $M = 100$ for Permuted MNIST, Split CIFAR-100, and Split miniImagenet, unless stated otherwise.

We report means and standard deviations using 5 seeds on all datasets. When reporting metrics, we indicate with red, orange, and yellow highlight the 1st, 2nd, and 3rd-best performing scheduling method based on its mean value on the metric for each dataset. Appendix C contains full details on the experimental settings and additional results including Welch's $t$-tests for statistical significance between MCTS and the baselines.

**Varying Memory Size.** We show that our method can improve the CL performance across varying memory sizes in different CL scenarios. Figure 3a shows the results in the Task- and Domain-Incremental Learning (IL) scenarios, where we observe that MCTS generally obtains better task accuracies than ETS, especially for small memory sizes. Both MCTS and ETS perform better than Heur-GD as $M$ increases, which shows that Heur-GD requires careful tuning of the validation thresholds. We provide results from statistical $t$-tests in Appendix C.7 to support our observations. We also performed experiments in the Class-IL scenario

---

[1]Code: https://github.com/marcusklasson/replay_scheduling

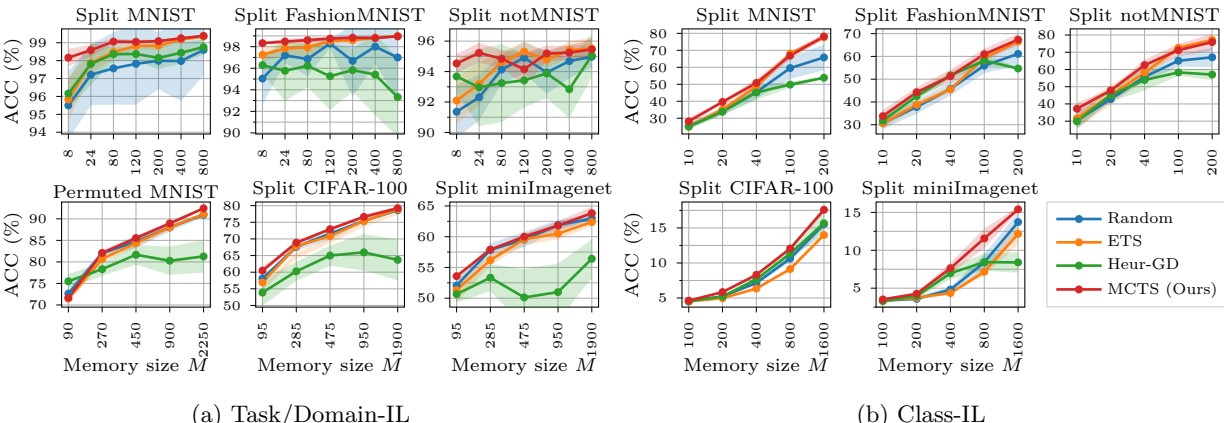

(a) Task/Domain-IL          (b) Class-IL

Figure 3: Performance comparison with ACC over various memory sizes for the methods, where (a) shows results in the Task- and Domain-Incremental Learning (IL) settings, and (b) in the Class-IL setting. The results have been averaged over 5 seeds. The results show that replay scheduling can outperform the baselines on both small and large datasets across different backbone choices in the different CL settings.

where task labels are absent. Here, the replay memory is always filled with at least 1 sample/class to avoid fully forgetting non-replayed tasks. Each scheduling method then selects which tasks to replay out of the remaining $M_{rest} = M - t \cdot n_c$ samples, where $n_c$ is the number of classes per task. Figure 3b shows that ETS approaches MCTS when $M$ increases on the 5-task datasets. However, on the more challenging Split CIFAR-100 and Split miniImagenet, MCTS outperforms ETS clearly as $M$ increases. These results show that selecting the proper replay schedule is essential in various CL scenarios with both small and large datasets across different backbone choices.

**Replay Schedule Visualization.** We visualize a replay schedule from Split CIFAR-100 with memory size $M = 100$ to gain insights into the behavior of the scheduling policy from MCTS. Figure 4 shows a bubble plot of the selected task proportions used for filling the replay memory at every task. Each circle color corresponds to a replay task, and its size represents the proportion of replay samples at the current task. The sum of points in all circles at each column is fixed at all current tasks. The task proportions vary dynamically over time in a sophisticated nonlinear way which would be hard to replace by a heuristic method. Moreover, we can observe space repetition-style scheduling on many tasks, e.g., task 1-3 are replayed with similar proportion at the initial tasks but eventually starts varying the time interval between replay. Also, task 4 and 6 need less replay in their early stages, which could potentially be that they

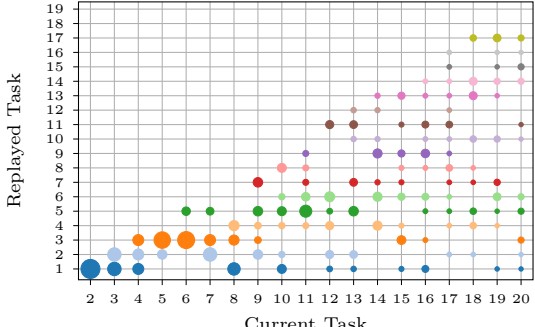

Figure 4: Replay schedule from MCTS on Split CIFAR-100 visualized as bubble plot. The task proportions vary dynamically over time which would be hard to replace with heuristics.

are simpler or correlated with other tasks. Appendix C.3 shows a similar visualization for Split MNIST.

**Scheduling with Different Memory Selection Methods.** We show that our method can be combined with any memory selection method for storing replay samples. In addition to uniform sampling, we apply various memory selection methods commonly used in the CL literature, namely *k*-means clustering and Mean-of-Features (MoF) (Rebuffi et al., 2017). Table 1 shows the results across all datasets. We indicate with red, orange, and yellow highlight the 1st, 2nd, and 3rd-best performing method based on its mean value on the metric. In Appendix C.4, we provide additional results from *k*-center selection (Nguyen et al., 2018) and *t*-tests, and also show that the forward transfer for new tasks is negligible for each method when uniform selection is used in this experiment. We note that using the replay schedule from MCTS outperforms the baselines when using the alternative selection methods, where MoF performs the best on most datasets.

Table 1: Performance comparison between MCTS (Ours) and the baselines with various memory selection methods, namely uniform sampling, $k$-means, and Mean-of-Features (MoF). We report the mean and standard deviation of ACC and BWT averaged over 5 seeds. MCTS performs better or on par than the baselines on most datasets and selection methods, where MoF yields the best performance in general.

| Memory | Schedule | Split MNIST | | Split FashionMNIST | | Split notMNIST | |
|---|---|---|---|---|---|---|---|
| | | ACC (%) | BWT (%) | ACC (%) | BWT (%) | ACC (%) | BWT (%) |
| Offline | Joint | 99.75 ± 0.06 | 0.01 ± 0.06 | 99.34 ± 0.08 | -0.01 ± 0.14 | 96.12 ± 0.57 | -0.21 ± 0.71 |
| Uniform | Random | 94.91 ± 2.52 | -6.13 ± 3.16 | 95.89 ± 2.03 | -4.33 ± 2.55 | 91.84 ± 1.48 | -5.37 ± 2.12 |
| | ETS | 94.02 ± 4.25 | -7.22 ± 5.33 | 95.81 ± 3.53 | -4.45 ± 4.34 | 91.01 ± 1.39 | -6.16 ± 1.82 |
| | Heur-GD | 96.02 ± 2.32 | -4.64 ± 2.90 | 97.09 ± 0.62 | -2.82 ± 0.84 | 91.26 ± 3.99 | -6.06 ± 4.70 |
| | MCTS (Ours) | 97.93 ± 0.56 | -2.27 ± 0.71 | 98.27 ± 0.17 | -1.29 ± 0.20 | 94.64 ± 0.39 | -1.47 ± 0.79 |
| $k$-means | Random | 92.65 ± 1.38 | -8.96 ± 1.74 | 93.11 ± 2.75 | -7.76 ± 3.42 | 93.11 ± 1.01 | -3.78 ± 1.43 |
| | ETS | 92.89 ± 3.53 | -8.66 ± 4.42 | 96.47 ± 0.85 | -3.55 ± 1.07 | 93.80 ± 0.82 | -2.84 ± 0.81 |
| | Heur-GD | 96.28 ± 1.68 | -4.32 ± 2.11 | 95.78 ± 1.50 | -4.46 ± 1.87 | 91.75 ± 0.94 | -5.60 ± 2.07 |
| | MCTS (Ours) | 98.20 ± 0.16 | -1.94 ± 0.22 | 98.48 ± 0.26 | -1.04 ± 0.31 | 93.61 ± 0.71 | -3.11 ± 0.55 |
| MoF | Random | 96.96 ± 1.34 | -3.57 ± 1.69 | 96.39 ± 1.69 | -3.66 ± 2.17 | 93.09 ± 1.40 | -3.70 ± 1.76 |
| | ETS | 97.04 ± 1.23 | -3.46 ± 1.50 | 96.48 ± 1.33 | -3.55 ± 1.73 | 92.64 ± 0.87 | -4.57 ± 1.59 |
| | Heur-GD | 96.46 ± 2.41 | -4.09 ± 3.01 | 95.84 ± 0.89 | -4.39 ± 1.15 | 93.24 ± 0.77 | -3.48 ± 1.37 |
| | MCTS (Ours) | 98.37 ± 0.24 | -1.70 ± 0.28 | 97.84 ± 0.32 | -1.81 ± 0.39 | 94.62 ± 0.42 | -1.80 ± 0.56 |

| Memory | Schedule | Permuted MNIST | | Split CIFAR-100 | | Split miniImagenet | |
|---|---|---|---|---|---|---|---|
| | | ACC (%) | BWT (%) | ACC (%) | BWT (%) | ACC (%) | BWT (%) |
| Offline | Joint | 95.34 ± 0.13 | 0.17 ± 0.18 | 84.73 ± 0.81 | -1.06 ± 0.81 | 74.03 ± 0.83 | 9.70 ± 0.68 |
| Uniform | Random | 72.59 ± 1.52 | -25.71 ± 1.76 | 53.76 ± 1.80 | -35.11 ± 1.93 | 49.89 ± 1.03 | -14.79 ± 1.14 |
| | ETS | 71.09 ± 2.31 | -27.39 ± 2.59 | 47.70 ± 2.16 | -41.69 ± 2.37 | 46.97 ± 1.24 | -18.32 ± 1.34 |
| | Heur-GD | 76.68 ± 2.13 | -20.82 ± 2.41 | 57.31 ± 1.21 | -30.76 ± 1.45 | 49.66 ± 1.10 | -12.04 ± 0.59 |
| | MCTS (Ours) | 76.34 ± 0.98 | -21.21 ± 1.16 | 56.60 ± 1.13 | -31.39 ± 1.11 | 50.20 ± 0.72 | -13.46 ± 1.22 |
| $k$-means | Random | 71.91 ± 1.24 | -26.45 ± 1.34 | 53.20 ± 1.44 | -35.77 ± 1.31 | 49.96 ± 1.46 | -14.81 ± 1.18 |
| | ETS | 69.40 ± 1.32 | -29.23 ± 1.47 | 47.51 ± 1.14 | -41.77 ± 1.30 | 45.82 ± 0.92 | -19.53 ± 1.10 |
| | Heur-GD | 75.57 ± 1.18 | -22.11 ± 1.22 | 54.31 ± 3.94 | -33.80 ± 4.24 | 49.25 ± 1.00 | -12.92 ± 1.22 |
| | MCTS (Ours) | 77.74 ± 0.80 | -19.66 ± 0.95 | 56.95 ± 0.92 | -30.92 ± 0.83 | 50.47 ± 0.85 | -13.31 ± 1.24 |
| MoF | Random | 78.80 ± 1.07 | -18.79 ± 1.16 | 62.35 ± 1.24 | -26.33 ± 1.25 | 56.02 ± 1.11 | -7.99 ± 1.13 |
| | ETS | 77.62 ± 1.12 | -20.10 ± 1.26 | 60.43 ± 1.17 | -28.22 ± 1.26 | 56.12 ± 1.12 | -8.93 ± 0.83 |
| | Heur-GD | 77.27 ± 1.45 | -20.15 ± 1.63 | 55.60 ± 2.70 | -32.57 ± 2.77 | 52.30 ± 0.59 | -9.61 ± 0.67 |
| | MCTS (Ours) | 81.58 ± 0.75 | -15.41 ± 0.86 | 64.22 ± 0.65 | -23.48 ± 1.02 | 57.70 ± 0.51 | -5.31 ± 0.55 |

Table 2: Performance comparison between scheduling methods MCTS (Ours), Random, ETS, and Heuristic combined with replay methods HAL, MER, and DER. We report the mean and standard deviation of ACC and BWT averaged over 5 seeds. * denotes results where some seed did not converge. Applying MCTS to each method can outperform the same method using the baseline schedules.

| Method | Schedule | Split MNIST | | Split CIFAR-100 | | Split miniImagenet | |
|---|---|---|---|---|---|---|---|
| | | ACC (%) | BWT (%) | ACC (%) | BWT (%) | ACC (%) | BWT (%) |
| HAL | Random | 96.32 ± 1.77 | -3.90 ± 2.28 | 35.90 ± 2.47 | -17.37 ± 3.76 | 40.86 ± 1.86 | -5.12 ± 2.23 |
| | ETS | 97.21 ± 1.25 | -2.80 ± 1.59 | 34.90 ± 2.02 | -18.92 ± 0.91 | 38.13 ± 1.18 | -8.19 ± 1.73 |
| | Heur-GD | 97.69 ± 0.19 | -2.22 ± 0.24 | 35.07 ± 1.29 | -24.76 ± 2.41 | 39.51 ± 1.49 | -5.65 ± 0.77 |
| | MCTS (Ours) | 97.96 ± 0.15 | -1.85 ± 0.18 | 40.22 ± 1.57 | -12.77 ± 1.30 | 41.39 ± 1.15 | -3.69 ± 1.86 |
| MER | Random | 93.00 ± 3.22 | -7.96 ± 4.15 | 42.68 ± 0.86 | -35.56 ± 1.39 | 32.86 ± 0.95 | -7.71 ± 0.45 |
| | ETS | 92.97 ± 1.73 | -8.52 ± 2.15 | 43.38 ± 1.81 | -34.84 ± 1.98 | 33.58 ± 1.53 | -6.80 ± 1.46 |
| | Heur-GD | 94.30 ± 2.79 | -6.46 ± 3.50 | 40.90 ± 1.70 | -44.10 ± 2.03 | 34.22 ± 1.93 | -7.57 ± 1.63 |
| | MCTS (Ours) | 96.44 ± 0.72 | -4.14 ± 0.94 | 44.29 ± 0.69 | -32.73 ± 0.88 | 32.74 ± 1.29 | -5.77 ± 1.04 |
| DER | Random | 95.91 ± 2.18 | -4.40 ± 2.46 | 56.17 ± 1.30 | -29.03 ± 1.38 | 35.13 ± 4.11 | -10.85 ± 2.92 |
| | ETS | 98.17 ± 0.35 | -2.00 ± 0.42 | 52.58 ± 1.49 | -32.93 ± 2.04 | 35.50 ± 2.84 | -10.94 ± 2.21 |
| | Heur-GD | 94.57 ± 1.71 | -6.08 ± 2.09 | 55.75 ± 1.08 | -31.27 ± 1.02 | 43.62 ± 0.88 | -8.18 ± 1.16 |
| | MCTS (Ours) | 99.02 ± 0.10 | -0.91 ± 0.13 | 58.99 ± 0.98 | -24.95 ± 0.64 | 43.46 ± 0.95 | -9.32 ± 1.37 |

**Applying Scheduling to Various Replay Methods.** In this experiment, we show that replay scheduling can be combined with any replay method to enhance the CL performance. We combine MCTS with Hindsight Anchor Learning (HAL) (Chaudhry et al., 2021), Meta-Experience Replay (MER) (Riemer et al., 2019), Dark Experience Replay (DER) (Buzzega et al., 2020). Table 2 shows the performance comparison between our the MCTS scheduling against using Random, ETS, and Heuristic schedules for each method. We indicate with red, orange, and yellow highlight the 1st, 2nd, and 3rd-best performing replay method based on its mean

Table 3: Performance comparison in memory setting where only 1 sample/class is available from the historical data for replay. The baselines replay all available samples, while MCTS selects 2 samples for Split MNIST and 50 samples for Permuted MNIST and Split miniImagenet. MCTS performs on par with the best baselines on Split MNIST and miniImagenet, and performs best on Permuted MNIST.

| Method | Split MNIST | | Permuted MNIST | | Split miniImagenet | |
|---|---|---|---|---|---|---|
| | ACC (%) | BWT (%) | ACC (%) | BWT (%) | ACC (%) | BWT (%) |
| Random | $92.56 \pm 2.90$ | $-8.97 \pm 3.62$ | $70.02 \pm 1.76$ | $-28.22 \pm 1.92$ | $48.85 \pm 1.38$ | $-14.55 \pm 1.86$ |
| A-GEM | $94.97 \pm 1.50$ | $-6.03 \pm 1.87$ | $64.71 \pm 1.78$ | $-34.41 \pm 2.05$ | $32.06 \pm 1.83$ | $-30.81 \pm 1.79$ |
| ER-Ring | $94.94 \pm 1.56$ | $-6.07 \pm 1.92$ | $69.73 \pm 1.13$ | $-28.87 \pm 1.29$ | $49.82 \pm 1.69$ | $-14.38 \pm 1.57$ |
| Uniform | $95.77 \pm 1.12$ | $-5.02 \pm 1.39$ | $69.85 \pm 1.01$ | $-28.74 \pm 1.17$ | $50.56 \pm 1.07$ | $-13.52 \pm 1.34$ |
| MCTS (Ours) | $96.07 \pm 1.60$ | $-4.59 \pm 2.01$ | $72.52 \pm 0.54$ | $-25.43 \pm 0.65$ | $50.70 \pm 0.54$ | $-12.60 \pm 1.13$ |

value on the metric. In Appendix C.5, we provide the hyperparameters for each replay method and results from statistical $t$-tests. The results confirm that replay scheduling is important for the final performance given the same memory constraints and it can benefit any existing CL framework.

**Efficiency of Replay Scheduling.** We illustrate the efficiency of replay scheduling in a setting where only 1 sample/class is available from the historical data for replay. We consider the scenario where the replay memory size is smaller than the number of classes. The replay memory size for MCTS is set to $M = 2$ for the 5-task datasets, such that only 2 samples can be selected for replay from the seen tasks. For the larger CL datasets, we set $M = 50$. We then compare against the memory efficient CL baselines A-GEM (Chaudhry et al., 2018b) and ER-Ring (Chaudhry et al., 2019), as well as uniform memory selection. We let these baselines increment their memory size, such that 1 sample/class is stored and used for replay. Table 3 shows the performance between MCTS and the baselines, where we indicate with red, orange, and yellow highlight the 1st, 2nd, and 3rd-best performing method based on its mean value on the metric. Despite using significantly fewer samples for replay, the MCTS schedule performs mostly on par with the baselines and outperforms them on Permuted MNIST. In Appendix C.6, we show that MCTS is statistically indistinguishable than the baselines on most datasets according to statistical $t$-tests. These results indicate that replay scheduling is an important research direction in CL, since storing every seen class in the memory could be inefficient in settings with large number of tasks.

**Additional Experiments.** We further demonstrate the benefits of replaying specific tasks in CL by i) showing that replay schedules found by MCTS can effectively reduce catastrophic forgetting with different replay samples from the scheduled tasks (see Appendix C.8), and ii) comparing the MCTS schedules against ETS combined with Heur-GD to assign higher proportions to hard tasks (see Appendix C.9).

### 4.2 Policy Generalization to New Continual Learning Scenarios

In this section, we evaluate how well the learned replay scheduling policies can mitigate catastrophic forgetting in new CL environments. We employ DQN and A2C for policy learning and evaluate their ability to generalize in CL environments with new task orders and datasets unseen during training.

**Experimental Setup.** We conduct experiments on the 5-task datasets Split MNIST, Split FashionMNIST, Split notMNIST, and Split CIFAR-10 (Krizhevsky & Hinton, 2009). The CL setting is in general the same as in Section 4.1. We evaluate all methods on 10 different test environments, and assess the generalization capability by ranking all methods by comparing their measured ACC per seed in each test environment, since the performance between environments can vary significantly (details in Appendix D.2). The policy is applied for only a single pass over the CL tasks at test time. We add two baselines:

- **Heuristic Local Drop (Heur-LD).** Heuristic policy that replays tasks with validation accuracy below a threshold proportional to the previous achieved validation accuracy on the task.
- **Heuristic Accuracy Threshold (Heur-AT).** Heuristic policy that replays tasks with validation accuracy below a fixed threshold.

Memory sizes are set to $M = 10$, and we average the results over 5 seeds. In the results, we indicate with red, orange, and yellow highlight the 1st, 2nd, and 3rd-best performing scheduling method based on its mean value for each metric. In Appendix D, we provide full details on the experimental settings and additional

Table 4: Performance comparison measured with average ranking (Rank) and ACC (%) between the scheduling policies in the **New Task Order** generalization experiment. The results are averaged across 10 test environments. Our learned policies using DQN and A2C performs competitively against the fixed policies (Random and ETS) and the heuristics (Heur) across the 5-task datasets.

| | S-MNIST | | S-FashionMNIST | | S-notMNIST | | S-CIFAR-10 | |
|---|---|---|---|---|---|---|---|---|
| **Schedule** | Rank ($\downarrow$) | ACC ($\uparrow$) | Rank ($\downarrow$) | ACC ($\uparrow$) | Rank ($\downarrow$) | ACC ($\uparrow$) | Rank ($\downarrow$) | ACC ($\uparrow$) |
| Random | 3.98 | $91.8 \pm 4.7$ | 3.68 | $93.9 \pm 3.8$ | 3.68 | $91.7 \pm 2.8$ | 4.91 | $81.6 \pm 6.1$ |
| ETS | 3.82 | $91.8 \pm 5.0$ | 4.74 | $93.5 \pm 3.0$ | 4.44 | $90.4 \pm 3.1$ | 5.38 | $81.0 \pm 5.9$ |
| Heur-GD | 4.53 | $91.3 \pm 4.3$ | 4.33 | $91.8 \pm 7.9$ | 3.44 | $92.3 \pm 1.5$ | 4.03 | $83.2 \pm 5.0$ |
| Heur-LD | 4.67 | $91.0 \pm 4.1$ | 3.63 | $93.8 \pm 5.0$ | 3.96 | $91.7 \pm 2.0$ | 3.63 | $83.5 \pm 4.2$ |
| Heur-AT | 4.38 | $91.5 \pm 4.0$ | 4.16 | $92.9 \pm 4.9$ | 5.50 | $89.7 \pm 2.9$ | 3.43 | $83.7 \pm 4.3$ |
| DQN (Ours) | 3.46 | $93.0 \pm 2.7$ | 3.78 | $94.2 \pm 3.8$ | 3.51 | $91.9 \pm 2.5$ | 3.83 | $83.0 \pm 4.3$ |
| A2C (Ours) | 3.16 | $93.1 \pm 3.7$ | 3.68 | $94.8 \pm 3.3$ | 3.47 | $92.1 \pm 1.8$ | 2.79 | $83.9 \pm 3.8$ |

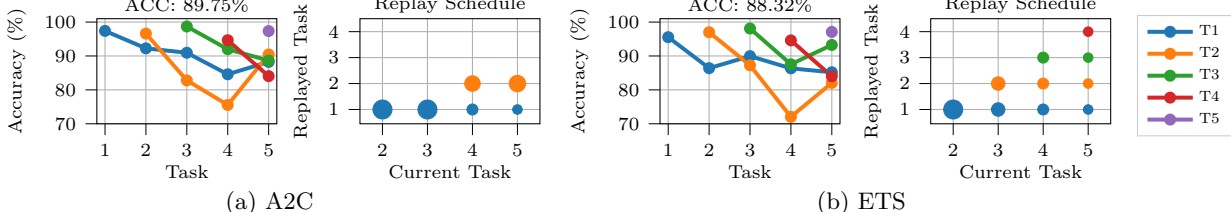

(a) A2C                                           (b) ETS

Figure 5: Task accuracies and replay schedules for A2C and ETS for a Split CIFAR-10 environment. The replay schedules are visualized as bubble plots and are from 1 seed. The learned policy by A2C can more flexibly than ETS consider replaying forgotten tasks, such as Task 2, to enhance the CL performance.

results including Welch's $t$-tests for statistical significance between our RL framework and the baselines. We also present baseline results with ETS combined with the heuristics in Appendix D.5 due to space limitations.

**Generalization to New Task Orders.** We show that the learned replay scheduling policies can generalize to CL environments with previously unseen task orders. The training and test environments are generated with unique task orders of the CL datasets. Table 4 shows the average ranking and ACC for DQN, A2C, and the baselines when being applied in the 10 test environments. Our learned policies obtain the best average ranking across most datasets, where A2C performs better than DQN in general. To provide further insights, Figure 5 shows the task accuracy progress and the corresponding replay schedule from A2C and ETS from one Split CIFAR-10 test environment. in Figure 5. The replay schedules are visualized with bubble plots showing the selected task proportion to use for composing the replay memories at each task. In Figure 5a, we observe that A2C decides to replay task 2 more than task 1 as the performance on task 2 decreases, which results in a slightly better ACC metric achieved by A2C than ETS. These results show that the learned policy can flexibly consider replaying forgotten tasks to enhance the CL performance.

**Generalization to New Datasets.** We show that the learned replay scheduling policies are capable of generalizing to CL environments with new datasets unseen in the training environments. We perform two sets of experiments with different training and testing environments for assessing the learned policy:

1. **S-notMNIST**: train with environments generated with Split MNIST and FashionMNIST and test on environments generated with Split notMNIST.
2. **S-FashionMNIST**: train with environments generated with Split MNIST and notMNIST and test on environments generated with Split FashionMNIST.

Table 5: Ranking and ACC between the scheduling policies in the **New Dataset** experiment. Our policies generalize well on S-notMNIST, but is outperformed on S-FashionMNIST.

| | S-notMNIST | | S-FashionMNIST | |
|---|---|---|---|---|
| **Schedule** | Rank ($\downarrow$) | ACC ($\uparrow$) | Rank ($\downarrow$) | ACC ($\uparrow$) |
| Random | 3.94 | $91.4 \pm 2.7$ | 4.05 | $92.6 \pm 4.7$ |
| ETS | 4.06 | $91.5 \pm 1.7$ | 3.84 | $92.8 \pm 3.7$ |
| Heur-GD | 4.61 | $89.8 \pm 4.8$ | 3.08 | $94.1 \pm 2.8$ |
| Heur-LD | 4.96 | $89.1 \pm 4.7$ | 4.96 | $90.9 \pm 4.1$ |
| Heur-AT | 4.29 | $90.2 \pm 4.9$ | 3.83 | $91.9 \pm 7.0$ |
| DQN (Ours) | 3.40 | $91.7 \pm 3.2$ | 4.12 | $92.2 \pm 5.3$ |
| A2C (Ours) | 2.74 | $92.6 \pm 1.6$ | 4.12 | $92.1 \pm 5.5$ |

Table 5 shows the average ranking and ACC for DQN, A2C, and the baselines when generalization to test environments with the new datasets. We observe that both A2C and DQN successfully generalize to Split

notMNIST compared to the baselines. However, the learned RL policies have difficulties generalizing to Split FashionMNIST environments, which could be due to high variations in the state transition dynamics, i.e., task accuracies, between training and test environments. This shows that learning the replay scheduling policies using RL inherits common challenges with generalization in RL, such as robustness to domain shifts. Potentially, the performance could be improved by generating more training environments for the agent to exhibit more variations of CL scenarios, or by using other advanced RL methods, e.g., regularization techniques (Igl et al., 2019) or online fine-tuning (Nair et al., 2020), which may generalize better.

## 5    Conclusions

We proposed learning the time to learn, i.e., in a real-world CL context, learning schedules of which tasks to replay at different times. To the best of our knowledge, we are the first to consider the time to learn in CL inspired by human learning techniques. We demonstrated the benefits with replay scheduling in CL by showing on several CL benchmarks that replay schedules found with MCTS can outperform replaying all tasks equally or relying on heuristic scheduling rules. Furthermore, we proposed an RL-based framework for learning scheduling policies as a step towards enabling replay scheduling in realistic CL settings. The learned policies are agnostic to the CL dataset, and can be applied to reduce catastrophic forgetting in new CL scenarios without additional training. Our replay scheduling approach brings current research closer to tackling real-world CL challenges where the number of tasks exceeds the replay memory size.

**Limitations and Future Work.** Generalization in RL is a challenging research topic by itself. With the current method, large amounts of diverse data and training time is required to enable the learned policy to generalize well. This can be costly since generating the CL environments is expensive as each state transition involves training the network on a CL task. Moreover, we are currently considering a discrete action space which is hard to construct, especially in large-scale CL scenarios. Thus, in future work, we would explore more advanced RL methods which can handle continuous actions and generalize well.

Finally, we study replay scheduling under the task-based CL setting where the task changes are known. Moreover, we assume that the number of tasks to learn are known beforehand and that validation sets are accessible at test time. Extending replay scheduling to task-free and online CL settings remains an open research question, where continual evaluation (De Lange et al., 2023) of class accuracies might be necessary for selecting effective replay memories. We hope that our demonstrated benefits with replay scheduling in CL will encourage more research in this direction.

### Broader Impact Statement

Concerns around privacy issues has been raised in the CL literature in settings where historical data is stored for replaying tasks over time. As discussed by Prabhu et al. (2023a), deep neural networks have been shown to be capable of reconstructing their training data (Haim et al., 2022), which means that removing access to historical data cannot solve potential privacy issues in CL on its own. Nevertheless, replay scheduling can be combined with any privacy-preserving replay method, e.g., methods that replay compressed features (Hayes et al., 2020) or generated synthetic data (Shin et al., 2017) rather than raw data. Furthermore, although learning task relationships could be beneficial, our proposed RL framework learns a dataset-agnostic policy for selecting which tasks to replay from CL performance metrics only.

A scalable scheduling method in replay-based CL would be useful for reducing compute in real-world applications but is currently missing. As storing historical data is in general affordable, scheduling methods for selecting small replay memories that effectively mitigate catastrophic forgetting are essential for CL systems. Replaying all previous tasks uniformly may be inefficient, which has been shown for human learning (Hawley et al., 2008), and creating heuristic scheduling rules can be hard to make them generalize to new scenarios. Hence, in this paper, we proposed that CL systems need to learn the time to learn from old tasks, in which we learn a scheduling policy for selecting which tasks to replay at different times. Since maintaining acceptable performance on a large number of tasks can be difficult in compute-constrained settings, we believe that replay scheduling is an important research direction for enabling real-world CL applications.

**Author Contributions**

CZ presented the idea, and MK and CZ contributed to formalizing the methodology. MK performed all the experiments, created the visualizations, and wrote most of the text. All authors took part in discussing the results and contributed to writing the manuscript.

**Acknowledgments**

This research was funded by the Promobilia Foundation (F-16500) while the first author was at KTH, and it was finished while being funded by the Finnish Center for Artificial Intelligence (FCAI). We would like to thank (in alphabetical order) Arno Solin, Christian Pek, Sofia Broomé, Ruibo Tu, and Truls Nyberg for feedback on the manuscript. We also thank Sam Devlin for early discussions on the reinforcement learning part of the paper. Finally, we thank the anonymous reviewers for their helpful suggestions and feedback.

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

## Appendix

This supplementary material is structured as follows:

- Our code is provided in the zip-file `supplementary_learn_the_time_to_learn.zip` as part of the supplementary material. The code will be made publicly available upon acceptance.

## A  Additional Methodology

In this section, we provide pseudo-code for MCTS to search for replay schedules in single CL environments in Section A.1 as well as pseudo-code for the RL-based framework for learning the replay scheduling policies in Section A.2.

### A.1  Monte Carlo Tree Search Algorithm for Replay Scheduling

We provide pseudo-code in Algorithm 1 outlining the steps for our method using Monte Carlo tree search (MCTS) to find replay schedules described in the main paper (Section 3.2). The MCTS procedure selects actions over which task proportions to fill the replay memory with at every task, where the selected task proportions are stored in the replay schedule $S$. The schedule is then passed to EVALUATEREPLAYSCHEDULE($\cdot$) where the continual learning part executes the training with replay memories filled according to the schedule. The reward for the schedule $S$ is the average validation accuracy over all tasks after learning task $T$, i.e., ACC, which is backpropagated through the tree to update the statistics of the selected nodes. The schedule $S_{best}$ yielding the best ACC score is returned to be used for evaluation on the held-out test sets.

The function GETREPLAYMEMORY($\cdot$) is the policy for retrieving the replay memory $\mathcal{M}$ from the historical data given the task proportion $\boldsymbol{p}$. The number of samples per task determined by the task proportions are rounded up or down accordingly to fill $\mathcal{M}$ with $M$ replay samples in total. The function GETTASKPROPORTION($\cdot$) simply returns the task proportion that is related to given node.

The following steps are performed during one MCTS rollout (or iteration):

1. **Selection** involves either selecting an unvisited node randomly, or selecting the next node by evaluating the UCT score (see Equation 1) if all children has been visited already. In Algorithm 1, TREEPOLICY($\cdot$) appends the task proportions $\boldsymbol{p}_t$ to the replay schedule $S$ at every selected node.

2. **Expansion** involves expanding the search tree with one of the unvisited child nodes $v_{t+1}$ selected with uniform sampling. EXPANSION($\cdot$) in Algorithm 1 appends the task proportions $\boldsymbol{p}_t$ to the replay schedule $S$ of the expanded node.

3. **Simulation** involves selecting the next nodes randomly until a terminal node $v_T$ is reached. In Algorithm 1, DEFAULTPOLICY($\cdot$) appends the task proportions $\boldsymbol{p}_t$ to the replay schedule $S$ at every randomly selected node until reaching the terminal node.

4. **Reward** The reward for the rollout is given by the ACC of the validation sets for each task. In Algorithm 1, EVALUATEREPLAYSCHEDULE($\cdot$) involves learning the tasks $t = 1, \ldots, T$ sequentially and using the replay schedule to sample the replay memories to use for mitigating catastrophic forgetting when learning a new task. The reward $r$ for the rollout is calculated after task $T$ has been learnt.

5. **Backpropagation** involves updating the reward function $q(\cdot)$ and number of visits $n(\cdot)$ from the expansion node up to the root node. See BACKRPROPAGATE($\cdot$) in Algorithm 1.

### A.2    RL Framework Algorithm

We provide pseudo-code for the RL-based framework for learning the replay scheduling policy with either DQN (Mnih et al., 2013) or A2C (Mnih et al., 2016) in Algorithm 2. The procedure collects experience from all training environments in $\mathcal{E}^{(train)}$ at every time step $t$. The datasets and classifiers are specific for each environment $E_i \in \mathcal{E}^{(train)}$. At $t = 1$, we obtain the initial state $s_1^{(i)}$ by evaluating the classifier on the validation set $\mathcal{D}_1^{(val)}$ after training the classifier on the task 1. Next, we get the replay memory for mitigating catastrophic forgetting when learning the next task $t + 1$ by 1) taking action $a_t^{(i)}$ under policy $\pi_{\boldsymbol{\theta}}$, 2) converting action $a_t^{(i)}$ into the task proportion $\boldsymbol{p}_t$, and 3) sampling the replay memory $\mathcal{M}_t$ from the historical datasets given the selected proportion. We then obtain the reward $r_t$ and the next state $s_{t+1}$ by evaluating the classifier on the validation sets $\mathcal{D}_{1:t+1}^{(val)}$ after learning task $t+1$. The collected experience from each time step is stored in the experience buffer $\mathcal{B}$ for both DQN and A2C. In UPDATEPOLICY($\cdot$), we outline the steps for updating the policy parameters $\boldsymbol{\theta}$ with either DQN or A2C.

---

**Algorithm 1** Monte Carlo Tree Search for Replay Scheduling

---

**Require:** Tree nodes $v_{1:T}$, Datasets $\mathcal{D}_{1:T}$, Learning rate $\eta$
**Require:** Replay memory size $M$
1: $\text{ACC}_{best} \leftarrow 0$, $S_{best} \leftarrow ()$
2: **while** within computational budget **do**
3:     $S \leftarrow ()$
4:     $v_t, S \leftarrow \text{TREEPOLICY}(v_1, S)$
5:     $v_T, S \leftarrow \text{DEFAULTPOLICY}(v_t, S)$
6:     $\text{ACC} \leftarrow \text{EVALUATEREPLAYSCHEDULE}(\mathcal{D}_{1:T}, S, M)$
7:     $\text{BACKPROPAGATE}(v_t, \text{ACC})$
8:     **if** $\text{ACC} > \text{ACC}_{best}$ **then**
9:         $\text{ACC}_{best} \leftarrow \text{ACC}$
10:        $S_{best} \leftarrow S$
11: **return** $\text{ACC}_{best}, S_{best}$

12: **function** $\text{TREEPOLICY}(v_t, S)$
13:     **while** $v_t$ is non-terminal **do**
14:         **if** $v_t$ not fully expanded **then**
15:             **return** $\text{EXPANSION}(v_t, S)$
16:         **else**
17:             $v_t \leftarrow \text{BESTCHILD}(v_t)$
18:             $S.\text{append}(\boldsymbol{p}_t)$, where $\boldsymbol{p}_t \leftarrow \text{GETTASKPROPORTION}(v_t)$
19:     **return** $v_t, S$

20: **function** $\text{EXPANSION}(v_t, S)$
21:     Sample $v_{t+1}$ uniformly among unvisited children of $v_t$
22:     $S.\text{append}(\boldsymbol{p}_{t+1})$, where $\boldsymbol{p}_{t+1} \leftarrow \text{GETTASKPROPORTION}(v_{t+1})$
23:     Add new child $v_{t+1}$ to node $v_t$
24:     **return** $v_{t+1}, S$

25: **function** $\text{BESTCHILD}(v_t)$
26:     $v_{t+1} = \underset{v_{t+1} \in \text{ children of } v}{\arg\max} \max(Q(v_{t+1})) + C\sqrt{\frac{2\log(N(v_t))}{N(v_{t+1})}}$
27:     **return** $v_{t+1}$

28: **function** $\text{DEFAULTPOLICY}(v_t, S)$
29:     **while** $v_t$ is non-terminal **do**
30:         Sample $v_{t+1}$ unIFormly among children of $v_t$
31:         $S.\text{append}(\boldsymbol{p}_{t+1})$, where $\boldsymbol{p}_{t+1} \leftarrow \text{GETTASKPROPORTION}(v_{t+1})$
32:         Update $v_t \leftarrow v_{t+1}$
33:     **return** $v_t, S$

34: **function** $\text{EVALUATEREPLAYSCHEDULE}(\mathcal{D}_{1:T}, S, M)$
35:     Initialize neural network $f_{\boldsymbol{\theta}}$
36:     **for** $t = 1, \ldots, T$ **do**
37:         $\boldsymbol{p} \leftarrow S[t-1]$
38:         $\mathcal{M} \leftarrow \text{GETREPLAYMEMORY}(\mathcal{D}_{1:t-1}^{(train)}, \boldsymbol{p}, M)$
39:         **for** $\mathcal{B} \sim \mathcal{D}_t^{(train)}$ **do**
40:             $\boldsymbol{\theta} \leftarrow SGD(\mathcal{B} \cup \mathcal{M}, \boldsymbol{\theta}, \eta)$
41:     $A_{1:T}^{(val)} \leftarrow \text{EVALUATEACCURACY}(f_{\boldsymbol{\theta}}, \mathcal{D}_{1:T}^{(val)})$
42:     $\text{ACC} \leftarrow \frac{1}{T}\sum_{i=1}^{T} A_{T,i}^{(val)}$
43:     **return** $\text{ACC}$

44: **function** $\text{BACKPROPAGATE}(v_t, R)$
45:     **while** $v_t$ is not root **do**
46:         $N(v_t) \leftarrow N(v_t) + 1$
47:         $Q(v_t) \leftarrow R$
48:         $v_t \leftarrow$ parent of $v_t$

---

---

**Algorithm 2** RL Framework for Learning Replay Scheduling Policy

---

**Require:** $\mathcal{E}^{(train)}$: Training environments, $\boldsymbol{\theta}$: Policy parameters, $\gamma$: Discount factor
**Require:** $\eta$: Learning rate, $n_{episodes}$: Number of episodes, $M$: Replay memory size
**Require:** $n_{steps}$: Number of steps for A2C

1:   $\mathcal{B} = \{\}$                                                        $\triangleright$ Initialize experience buffer
2:   **for** $i = 1, \ldots, n_{\text{episodes}}$ **do**
3:      **for** $t = 1, \ldots, T-1$ **do**
4:         **for** $E_i \in \mathcal{E}^{(train)}$ **do**
5:            $\mathcal{D}_{1:t+1} = \text{GETDATASETS}(E_i, t)$                $\triangleright$ Get datasets from environment $E_i$
6:            $f_{\boldsymbol{\phi}}^{(i)} = \text{GETCLASSIFIER}(E_i)$                $\triangleright$ Get classifier from environment $E_i$
7:            **if** $t == 1$ **then**
8:               $\text{TRAIN}(f_{\boldsymbol{\phi}}^{(i)}, \mathcal{D}_t^{(train)})$                $\triangleright$ Train classifier $f_{\boldsymbol{\phi}}^{(i)}$ on task 1
9:               $A_{1:t}^{(val)} = \text{EVAL}(f_{\boldsymbol{\phi}}^{(i)}, \mathcal{D}_{1:t}^{(val)})$           $\triangleright$ Evaluate classifier $f_{\boldsymbol{\phi}}^{(i)}$ on task 1
10:           $s_t^{(i)} = A_{1:t}^{(val)} = [A_{1,1}^{(val)}, 0, ..., 0]$              $\triangleright$ Get initial state
11:           $a_t^{(i)} \sim \pi_{\boldsymbol{\theta}}(a, s_t^{(i)})$                  $\triangleright$ Take action under policy $\pi_{\boldsymbol{\theta}}$
12:           $\boldsymbol{p}_t = \text{GETTASKPROPORTION}(a_t^{(i)})$
13:           $\mathcal{M}_t \sim \text{GETREPLAYMEMORY}(\mathcal{D}_{1:t}^{(train)}, \boldsymbol{p}_t, M)$
14:           $\text{TRAIN}(f_{\boldsymbol{\phi}}^{(i)}, \mathcal{D}_{t+1}^{(train)} \cup \mathcal{M}_t)$               $\triangleright$ Train classifier $f_{\boldsymbol{\phi}}^{(i)}$
15:           $A_{1:t+1}^{(val)} = \text{EVAL}(f_{\boldsymbol{\phi}}^{(i)}, \mathcal{D}_{1:t+1}^{(val)})$           $\triangleright$ Evaluate classifier $f_{\boldsymbol{\phi}}^{(i)}$
16:           $s_{t+1}^{(i)} = A_{1:t+1}^{(val)} = [A_{t+1,1}^{(val)}, ..., A_{t+1,t+1}^{(val)}, 0, ..., 0]$     $\triangleright$ Get next state
17:           $r_t^{(i)} = \frac{1}{t+1} \sum_{j=1}^{t+1} A_{1:t+1}^{(val)}$                $\triangleright$ Compute reward
18:           $\mathcal{B} = \mathcal{B} \cup \{(s_t^{(i)}, a_t^{(i)}, r_t^{(i)}, s_{t+1}^{(i)})\}$         $\triangleright$ Store transition in buffer
19:           **if** time to update policy **then**
20:              $\boldsymbol{\theta}, \mathcal{B} = \text{UPDATEPOLICY}(\boldsymbol{\theta}, \mathcal{B}, \gamma, \eta, n_{steps})$       $\triangleright$ Update policy with experience
21: **return** $\boldsymbol{\theta}$                                                 $\triangleright$ Return policy

22: **function** $\text{UPDATEPOLICY}(\boldsymbol{\theta}, \mathcal{B}, \gamma, \eta, n_{steps})$
23:      **if** DQN **then**
24:         $(s_j, a_j, r_j, s_j') \sim \mathcal{B}$                    $\triangleright$ Sample mini-batch from buffer
25:         $y_j = \begin{cases} r_j & \text{if } s_j' \text{ is terminal} \\ r_j + \gamma \max_a Q_{\boldsymbol{\theta}^-}(s_j', a) & \text{else} \end{cases}$       $\triangleright$ Compute $y_j$ with target net $\boldsymbol{\theta}^-$
26:         $\boldsymbol{\theta} = \boldsymbol{\theta} - \eta \nabla_{\boldsymbol{\theta}}(y_j - Q_{\boldsymbol{\theta}}(s_j, a_j))^2$         $\triangleright$ Update $Q$-function
27:      **else if** A2C **then**
28:         $s_t = \mathcal{B}[n_{steps}]$                      $\triangleright$ Get last state in buffer
29:         $R = \begin{cases} 0 & \text{if } s_t \text{ is terminal} \\ V_{\boldsymbol{\theta}_v}(s_t) & \text{else} \end{cases}$         $\triangleright$ Bootstrap from last state
30:         **for** $j = n_{steps} - 1, ..., 0$ **do**
31:            $s_j, a_j, r_j = \mathcal{B}[j]$          $\triangleright$ Get state, action, and reward at step $j$
32:            $R = r_j + \gamma R$
33:            $\boldsymbol{\theta} = \boldsymbol{\theta} - \eta \nabla_{\boldsymbol{\theta}} \log \pi_{\boldsymbol{\theta}}(a_j, s_j)(R - V_{\boldsymbol{\theta}_v}(s_j))$       $\triangleright$ Update policy
34:            $\boldsymbol{\theta}_v = \boldsymbol{\theta}_v - \eta \nabla_{\boldsymbol{\theta}_v}(R - V_{\boldsymbol{\theta}_v}(s_j))^2$         $\triangleright$ Update value function
35:         $\mathcal{B} = \{\}$                        $\triangleright$ Reset experience buffer
36:      **return** $\boldsymbol{\theta}, \mathcal{B}$

---

# B    Heuristic Scheduling Baselines

We implemented three heuristic scheduling baselines to compare against our proposed methods. These heuristics are based on the intuition of re-learning tasks when they have been forgotten. We keep a validation set for each task to determine whether any task should be replayed by comparing the validation accuracy against a hand-tuned threshold. If the validation accuracy is below the threshold, then the corresponding task is replayed. Let $A_{t,i}^{(val)}$ be the validation accuracy for task $t$ evaluated at time step $i$. The threshold is set differently in each of the baselines:

- **Heuristic Global Drop (Heur-GD).** Heuristic policy that replays tasks with validation accuracy below a certain threshold proportional to the best achieved validation accuracy on the task. The best achieved validation accuracy for task $i$ is given by $A_{t,i}^{(best)} = \max\{(A_{1,i}^{(val)}, \ldots, A_{t,i}^{(val)})\}$. Task $i$ is replayed if $A_{t,i}^{(val)} < \tau A_{t,i}^{(best)}$ where $\tau \in [0,1]$ is a ratio representing the degree of how much the validation accuracy of a task is allowed to drop. Note that Heur-GD (denoted as Heuristic) is the only one used in the experiments with MCTS in single CL environments in Section 4.1.

- **Heuristic Local Drop (Heur-LD).** Heuristic policy that replays tasks with validation accuracy below a threshold proportional to the previous achieved validation accuracy on the task. Task $i$ is replayed if $A_{t,i}^{(val)} < \tau A_{t-1,i}^{(val)}$ where $\tau$ again represents the degree of how much the validation accuracy of a task is allowed to drop.

- **Heuristic Accuracy Threshold (Heur-AT).** Heuristic policy that replays tasks with validation accuracy below a fixed threshold. Task $i$ is replayed if if $A_{t,i}^{(val)} < \tau$ where $\tau \in [0,1]$ represents the least tolerated accuracy before we need to replay the task.

The replay memory is filled with $M/k$ samples from each selected task, where $k$ is the number of tasks that need to be replayed according to their decrease in validation accuracy. We skip replaying any tasks if no tasks are selected for replay, i.e., $k = 0$.

**Grid search for $\tau$ in Single CL Environments.** We performed a coarse-to-fine grid search for the parameter $\tau$ on each dataset with Heur-GD to compare against the MCTS replay schedules. The best value for $\tau$ is selected according to the highest mean accuracy on the validation set averaged over 5 seeds. The validation set consists of 15% of the training data and is the same for MCTS. We use the same experimental settings as described in Appendix C. The memory sizes are set to $M = 10$ and $M = 100$ for the 5-task datasets and the 10/20-task datasets respectively, and we apply uniform sampling as the memory selection method. We provide the ranges for $\tau$ that was used on each dataset and put the best value in **bold**:

- Split MNIST: $\tau = \{0.9, 0.93, 0.95, \mathbf{0.96}, 0.97, 0.98, 0.99\}$

- Split FashionMNIST: $\tau = \{0.9, 0.93, 0.95, 0.96, \mathbf{0.97}, 0.98, 0.99\}$

- Split notMNIST: $\tau = \{0.9, 0.93, 0.95, 0.96, 0.97, \mathbf{0.98}, 0.99\}$

- Permuted MNIST: $\tau = \{0.5, 0.55, 0.6, 0.65, 0.7, \mathbf{0.75}, 0.8, 0.9, 0.95, 0.97, 0.99\}$

- Split CIFAR-100: $\tau = \{0.3, 0.4, 0.45, \mathbf{0.5}, 0.55, 0.6, 0.65, 0.7, 0.8, 0.9, 0.95, 0.97, 0.99\}$

- Split miniImagenet: $\tau = \{0.5, 0.6, 0.65, 0.7, \mathbf{0.75}, 0.8, 0.85, 0.9, 0.95, 0.97, 0.99\}$

Note that we use these values for $\tau$ on all experiments with Heur-GD for the corresponding datasets. The performance for the heuristics highly depends on careful tuning for the ratio $\tau$ when the memory size or memory selection method changes, as can be seen in Figure 3 and Table 9. We also provide the ranges for $\tau$ that was used on each dataset in the Class Incremental Learning setting and put the best value in **bold**:

- Split MNIST: $\tau = \{0.2, 0.3, 0.5, \mathbf{0.75}, 0.9\}$

- Split FashionMNIST: $\tau = \{0.2, 0.3, \mathbf{0.5}, 0.75, 0.9\}$

- Split notMNIST: $\tau = \{0.2, 0.3, \mathbf{0.5}, 0.75, 0.9\}$

Table 6: The threshold parameter $\tau$ used in the heuristic scheudling baselines Heuristic Global Drop (Heur-GD), Heuristic Local Drop (Heur-LD), and Heuristic Accuracy Threshold (Heur-AT). The search range is $\tau \in \{0.90, 0.95, 0.999\}$ for all methods and we display the number of environments used for selecting the parameter used at test time.

| | New Task Order | | | | | | | | New Dataset | | | |
| | S-MNIST | | S-FashionMNIST | | S-notMNIST | | S-CIFAR-10 | | S-notMNIST | | S-FashionMNIST | |
| Method | $\tau$ | #Envs | $\tau$ | #Envs | $\tau$ | #Envs | $\tau$ | #Envs | $\tau$ | #Envs | $\tau$ | #Envs |
|---|---|---|---|---|---|---|---|---|---|---|---|---|
| Heur-GD | 0.9 | 10 | 0.95 | 20 | 0.999 | 10 | 0.9 | 10 | 0.9 | 10 | 0.9 | 10 |
| Heur-LD | 0.9 | 10 | 0.999 | 20 | 0.999 | 10 | 0.999 | 10 | 0.95 | 10 | 0.999 | 10 |
| Heur-AT | 0.9 | 10 | 0.999 | 20 | 0.999 | 10 | 0.9 | 10 | 0.9 | 10 | 0.95 | 10 |

- Split CIFAR-100: $\tau = \{\mathbf{0.01}, 0.025, 0.05, 0.1, 0.25, 0.5\}$

- Split miniImagenet: $\tau = \{\mathbf{0.01}, 0.025, 0.05, 0.1, 0.25, 0.5\}$

**Grid search for $\tau$ in Multiple CL Environments.** We performed a grid search for the parameter $\tau$ for the three heuristic scheduling baselines for each experiment to compare against the learned replay scheduling policies. We select the parameter based on ACC scores achieved in the same number of training environments used by either DQN or A2C. The search range we use is $\tau \in \{0.90, 0.95, 0.999\}$. In Table 6, we show the selected parameter value of $\tau$ and the number of environments used for selecting the value for each method and experiment in Section 4.2. The same parameters are used to generate the results on the heuristics in Table 4 and 5.

## C  Additional Experimental Settings and Results for Replay Scheduling using MCTS

This section is structured as follows:

- Appendix C.1: Full details on the experimental settings.

- Appendix C.2: Performance progress of MCTS as sanity check.

- Appendix C.3: Visualization of replay schedule from MCTS on Split MNIST.

- Appendix C.4: Additional results on Memory Selection Methods experiment.

- Appendix C.5: Additional results on Applying Replay Scheduling to Recent Replay Methods experiment.

- Appendix C.6: Additional results on Efficiency of Replay Scheduling experiment.

- Appendix C.7: Additional results on Varying Memory Size experiment.

The additional results appendices provide Welch's t-tests for statistical significance between MCTS and the baselines.

### C.1  Experimental Settings for MCTS in Single CL Environments

Here, we provide details on the experimental settings for the experiments with MCTS in single CL environments.

**Datasets.** We conduct experiments on six datasets commonly used in the CL literature. Split MNIST (Zenke et al., 2017) is a variant of the MNIST (LeCun et al., 1998) dataset where the classes have been divided into 5 tasks incoming in the order 0/1, 2/3, 4/5, 6/7, and 8/9. Split FashionMNIST (Xiao et al., 2017) is of similar size to MNIST and consists of grayscale images of different clothes, where the classes have been divided into the 5 tasks T-shirt/Trouser, Pullover/Dress, Coat/Sandals, Shirt/Sneaker, and Bag/Ankle boots. Similar to MNIST, Split notMNIST (Bulatov, 2011) consists of 10 classes of the letters A-J with various fonts, where the classes are divided into the 5 tasks A/B, C/D, E/F, G/H, and I/J. We use training/test split provided by Ebrahimi et al. (2020) for Split notMNIST. Permuted MNIST (Goodfellow et al., 2013) dataset consists

of applying a unique random permutation of the pixels of the images in original MNIST to create each task, except for the first task that is to learn the original MNIST dataset. We reduce the original MNIST dataset to 10k samples and create 9 unique random permutations to get a 10-task version of Permuted MNIST. In Split CIFAR-100 (Krizhevsky & Hinton, 2009), the 100 classes are divided into 20 tasks with 5 classes for each task (Lopez-Paz & Ranzato, 2017; Rebuffi et al., 2017). Similarly, Split miniImagenet (Vinyals et al., 2016) consists of 100 classes randomly chosen from the original Imagenet dataset where the 100 classes are divided into 20 tasks with 5 classes per task.

**CL Network Architectures.** We use a 2-layer MLP with 256 hidden units and ReLU activation for Split MNIST, Split FashionMNIST, Split notMNIST, and Permuted MNIST. We use a multi-head output layer for each dataset except Permuted MNIST where the network uses single-head output layer. For Split CIFAR-100, we use a multi-head CNN architecture built according to the CNN in Adel et al. (2019); Schwarz et al. (2018); Vinyals et al. (2016), which consists of four 3x3 convolutional blocks, i.e. convolutional layer followed by batch normalization (Ioffe & Szegedy, 2015), with 64 filters, ReLU activations, and 2x2 Max-pooling. For Split miniImagenet, we use the reduced ResNet-18 from Lopez-Paz & Ranzato (2017) with multi-head output layer.

**CL Hyperparameters.** We train all networks with the Adam optimizer (Kingma & Ba, 2015) with learning rate $\eta = 0.001$ and hyperparameters $\beta_1 = 0.9$ and $\beta_2 = 0.999$. Note that the learning rate for Adam is not reset before training on a new task. Next, we give details on number of training epochs and batch sizes specific for each dataset:

- Split MNIST: 10 epochs/task, batch size 128.

- Split FashionMNIST: 30 epochs/task, batch size 128.

- Split notMNIST: 50 epochs/task, batch size 128.

- Permuted MNIST: 20 epochs/task, batch size 128.

- Split CIFAR-100: 25 epochs/task, batch size 256.

- Split miniImagenet: 1 epoch/task (task 1 trained for 5 epochs as warm up), batch size 32.

**Replay Memory Settings.** The replay memory $\mathcal{M}$ has a fixed size $M$ throughout the CL training sequence. The memory is filled by fetching the $M$ samples from the historical data using a memory selection, e.g., uniform sampling, $k$-means, etc. The $M$-sized replay memory is replayed in every training iteration when learning the current task by concatenating the $M$ replay samples with the $B$ current task samples, where $B$ is the batch size. If $M > B$, we randomly sample $B$ replay samples among the $M$ samples and concatenate these to the current task samples. The $B + M$ samples are weighted equally when computing the loss, regardless of which task the examples are from.

**Monte Carlo Tree Search.** We run RS-MCTS for 100 iterations in all experiments. The replay schedules used in the reported results on the held-out test sets are from the replay schedule that gave the highest reward on the validation sets. The exploration constant for UCT in Equation 1 is set to $C = 0.1$ in all experiments (Chaudhry & Lee, 2018).

**Computational Cost.** All experiments were performed on one NVIDIA GeForce RTW 2080Ti on an internal GPU cluster. The wall clock time for ETS on Split MNIST was around 1.5 minutes, and RS-MCTS and BFS takes 40 seconds on average to run one iteration, where BFS runs 1050 iterations in total for Split MNIST.

**Implementations.** We adapted the implementation released by Borsos et al. (2020) for the memory selection strategies Uniform sampling, $k$-means clustering, $k$-center clustering (**?**), and Mean-of-Features (Rebuffi et al., 2017). For HAL (Chaudhry et al., 2021), MER (Riemer et al., 2019), DER (Buzzega et al., 2020), and DER++, we follow the implementations released by Buzzega et al. (2020) for each method to apply them to our replay scheduling methods. Furthermore, we follow the implementations released by Chaudhry et al. (2019) and Mirzadeh & Ghasemzadeh (2021) for A-GEM (Chaudhry et al., 2018b) and ER-Ring (Chaudhry et al., 2019). For MCTS, we adapted the implementation from `https://github.com/int8/monte-carlo-tree-search` to search for replay schedules.

**Experimental Settings for Class Incremental Learning Experiment.** In Section 4.1 and Figure 3b, we perform experiments in the Class Incremental Learning (Class-IL) setting where task labels are missing for every data point. The memory size $M$ is fixed throughout the CL training sequence, but the replay memory is filled with at least 1 sample per class at every task, otherwise tasks that are not replayed would be fully forgotten, i.e., accuracy = 0.0%. All methods then schedule which tasks to replay out of the remaining samples $M_{rest} = M - t \cdot n_c$, where $t$ is the task number and $n_c$ is the number of classes per task in the dataset. The CL network architectures are the same as above but with a single-head output layer. The other experimental settings remain the same as described above, i.e., same learning rates, number of epochs, batch size.

**Experimental Settings for Single Task Replay Memory Experiment.** We motivated the need for replay scheduling in CL with Figure 1 in Section 1. This simple experiment was performed on Split MNIST where the replay memory only contains samples from the first task, i.e., learning the classes 0/1. Furthermore, the memory can only be replayed at one point in time and we show the performance on each task when the memory is replayed at different time steps. We set the memory size to $M = 10$ samples such that the memory holds 5 samples from both classes. We use the same network architecture and hyperparameters as described above for Split MNIST. The ACC metric above each subfigure corresponds to the ACC for training a network with the single task memory replay at different tasks. We observe that choosing different time points to replay the same memory leads to noticeably different results in the final performance, and in this example, the best final performance is achieved when the memory is used when learning task 5. Therefore, we argue that finding the proper schedule of what tasks to replay at what time in the fixed memory situation can be critical for CL.

## C.2 Performance Progress of MCTS

In the first experiments, we show that the replay schedules from MCTS yield better performance than replaying an equal amount of samples per task. The replay memory size is fixed to $M = 10$ for Split MNIST, FashionMNIST, and notMNIST, and $M = 100$ for Permuted MNIST, Split CIFAR-100, and Split miniImagenet. Uniform sampling is used as the memory selection method for all methods in this experiment. For the 5-task datasets, we provide the optimal replay schedule found from a breadth-first search (BFS) over all 1050 possible replay schedules in our action space (which corresponds to a tree with depth of 4) as an upper bound for MCTS. As the search space grows fast with the number of tasks, BFS becomes computationally infeasible when we have 10 or more tasks.

Figure 6 shows the progress of ACC over iterations by MCTS for all datasets. We also show the best ACC metrics for Random, ETS, Heuristic, and BFS (where appropriate) as straight lines. Furthermore, we include the ACC achieved by training on all seen datasets jointly at every task (Joint) for the 5-task datasets. We observe that MCTS outperforms ETS successively with more iterations. Furthermore, MCTS approaches the upper limit of BFS on the 5-task datasets. For Permuted MNIST and Split CIFAR-100, the Heuristic baseline and MCTS perform on par after 50 iterations. This shows that Heuristic with careful tuning of the validation accuracy threshold can be a strong baseline when comparing replay scheduling methods. The top row of Table 1 shows the ACC for each method for this experiment. We note that MCTS outperforms ETS significantly on most datasets and performs on par with Heuristic.

## C.3 Replay Schedule Visualization for Split MNIST

In Figure 7, we show the progress in test classification performance for each task when using ETS and MCTS with memory size $M = 10$ on Split MNIST. For comparison, we also show the performance from a network that is fine-tuning on the current task without using replay. Both ETS and MCTS overcome catastrophic forgetting to a large degree compared to the fine-tuning network. Our method MCTS further improves the performance compared to ETS with the same memory, which indicates that learning the time to learn can be more efficient against catastrophic forgetting. Especially, Task 1 and 2 seems to be the most difficult task to remember since it has the lowest final performance using the fine-tuning network. Both ETS and MCTS manage to retain their performance on Task 1 using replay, however, MCTS remembers Task 2 better than ETS by around 5%.

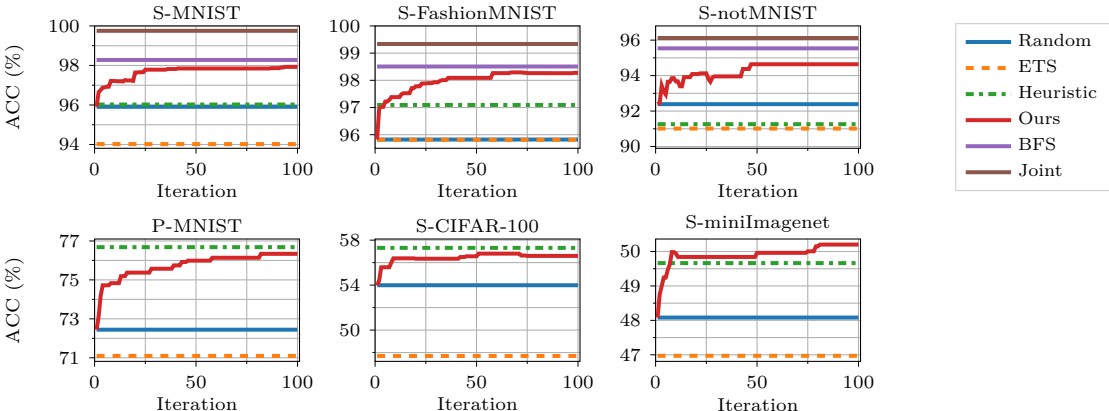

Figure 6: Average test accuracies over tasks after learning the final task (ACC) over the MCTS simulations for all datasets, where 'S' and 'P' are used as short for 'Split' and 'Permuted'. We compare performance for MCTS (Ours) against random replay schedules (Random), Equal Task Schedule (ETS), and Heuristic Global Drop (Heuristic) baselines. For the first three datasets, we show the best ACC found from a breadth-first search (BFS) as well as the ACC achieved by training on all seen datasets jointly at every task (Joint). All results have been averaged over 5 seeds. These results show that replay scheduling can improve over ETS and outperform or perform on par with Heuristic across different datasets and network architectures.

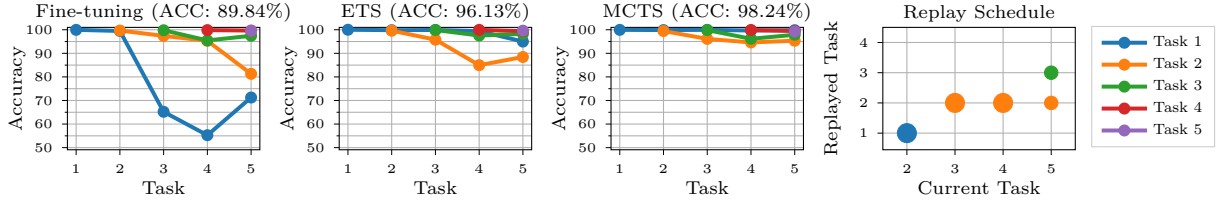

Figure 7: Comparison of test classification accuracies for Task 1-5 on Split MNIST from a network trained without replay (Fine-tuning), ETS, and MCTS. The ACC metric for each method is shown on top of each figure. We also visualize the replay schedule found by MCTS as a bubble plot to the right. The memory size is set to $M = 10$ with uniform memory selection for ETS and MCTS. Results are shown for 1 seed.

To bring more insights to this behavior, we have visualized the task proportions of the replay examples using a bubble plot showing the corresponding replay schedule from MCTS in Figure 7(right). At Task 3 and 4, we see that the schedule fills the memory with data from Task 2 and discards replaying Task 1. This helps the network to retain knowledge about Task 2 better than ETS at the cost of forgetting Task 3 slightly when learning Task 4. This shows that the learned policy has considered the difficulty level of different tasks. At the next task, the MCTS schedule has decided to rehearse Task 3 and reduces replaying Task 2 when learning Task 5. This behavior is similar to spaced repetition, where increasing the time interval between rehearsals helps memory retention. We emphasize that even on datasets with few tasks, using learned replay schedules can overcome catastrophic forgetting better than standard ETS approaches.

## C.4 Alternative Memory Selection Methods

We show that our method can be combined with any memory selection method for storing replay samples. In addition to uniform sampling, we apply various memory selection methods commonly used in the CL literature, namely $k$-means clustering, $k$-center clustering (**?**), and Mean-of-Features (MoF) (Rebuffi et al., 2017). The replay memory sizes are $M = 10$ for the 5-task datasets and $M = 100$ for the 10- and 20-task datasets. Table 9 shows the performance metrics across all datasets, where we indicate with red, orange, and yellow highlight the 1st, 2nd, and 3rd-best performing scheduling method based on its mean value for each metric. Furthermore, we present the statistical significance tests between MCTS and the baselines in

Table 7: Forward transfer (FWT) comparison between MCTS and the scheduling baselines using Uniform memory selection. Results are averaged over 5 seeds and the memory sizes are $M = 10$ and $M = 100$ for 5-task and 10/20-task datasets respectively. The FWT metric highly varies between seeds especially for the 5-task datasets, and is negligible for PermutedMNIST and Split CIFAR-100.

| Schedule | S-MNIST | S-FashionMNIST | S-notMNIST | P-MNIST | S-CIFAR-100 |
|---|---|---|---|---|---|
| Random | $7.03 \pm 5.32$ | $2.33 \pm 8.33$ | $1.28 \pm 5.83$ | $-0.02 \pm 0.79$ | $-0.24 \pm 1.05$ |
| ETS | $7.77 \pm 5.40$ | $2.55 \pm 8.35$ | $0.70 \pm 5.26$ | $-0.43 \pm 0.55$ | $-0.49 \pm 1.14$ |
| Heur-GD | $3.16 \pm 5.60$ | $3.17 \pm 6.87$ | $3.83 \pm 4.98$ | $-0.02 \pm 1.07$ | $-1.00 \pm 1.34$ |
| MCTS | $3.96 \pm 4.21$ | $5.02 \pm 12.18$ | $2.98 \pm 5.22$ | $0.06 \pm 1.06$ | $-0.13 \pm 0.67$ |

Table 8. We note that our method in general achieves significantly higher ACC comparing to the baselines showing that learning the time to learn is important.

**Forward Transfer**: We assess the forward transfer for MCTS and the baselines which tells how well the current task has benefitted from learning the previous tasks. We use the FWT metric from Lopez-Paz & Ranzato (2017) which is computed as:

$$\text{FWT} = \frac{1}{T-1} \sum_{i=2}^{T} A_{i-1,i} - b_i \in [-1, 1], \tag{3}$$

where $A_{t,i}$ is the test accuracy for task $i$ after learning task $t$ and $b_i$ is the test accuracy for task $i$ after random initialization. Table 7 shows the FWT for MCTS and the baselines using Uniform memory selection. We observe that the FWT for all methods and datasets highly varies between seeds, especially for the 5-task datasets. We believe that this large standard deviations could be reduced by using a larger memory size $M$. Furthermore, we conclude that the FWT is negligible in general across each scheduling method, which stems with the reported results on Permuted MNIST and Split CIFAR-100 in Lopez-Paz & Ranzato (2017).

Table 8: Two-tailed Welch's $t$-test results for the various memory selection methods presented in Table 9. We bold the $p$-values where $p < 0.05$ to indicate for when a method is statistically significantly better than its comparison.

| Memory | Schedule | Split MNIST | | Split FashionMNIST | | Split notMNIST | |
|---|---|---|---|---|---|---|---|
| | | $t$ | $p$ | $t$ | $p$ | $t$ | $p$ |
| Uniform | MCTS vs Random | 2.34 | 0.074 | 2.34 | 0.078 | 3.66 | **0.017** |
| | MCTS vs ETS | 1.82 | 0.140 | 1.39 | 0.236 | 5.04 | **0.005** |
| | MCTS vs Heur-GD | 1.60 | 0.178 | 3.64 | **0.017** | 1.69 | 0.166 |
| $k$-means | MCTS vs Random | 7.97 | **0.001** | 3.90 | **0.017** | 0.81 | 0.445 |
| | MCTS vs ETS | 3.00 | **0.040** | 4.54 | **0.007** | -0.36 | 0.732 |
| | MCTS vs Heur-GD | 2.27 | 0.085 | 3.55 | **0.022** | 3.15 | **0.015** |
| $k$-center | MCTS vs Random | 6.15 | **0.001** | 3.37 | **0.027** | 2.59 | 0.057 |
| | MCTS vs ETS | 4.71 | **0.007** | 2.56 | **0.037** | 2.53 | 0.062 |
| | MCTS vs Heur-GD | 2.62 | 0.057 | 2.13 | 0.099 | 3.51 | **0.019** |
| MoF | MCTS vs Random | 2.07 | 0.103 | 1.68 | 0.163 | 2.10 | 0.093 |
| | MCTS vs ETS | 2.13 | 0.095 | 1.99 | 0.110 | 4.11 | **0.007** |
| | MCTS vs Heur-GD | 1.58 | 0.188 | 4.21 | **0.008** | 3.15 | **0.019** |

| Memory | Schedule | Permuted MNIST | | Split CIFAR-100 | | Split miniImagenet | |
|---|---|---|---|---|---|---|---|
| | | $t$ | $p$ | $t$ | $p$ | $t$ | $p$ |
| Uniform | MCTS vs Random | 4.14 | **0.005** | 2.67 | **0.033** | 0.49 | 0.636 |
| | MCTS vs ETS | 4.18 | **0.007** | 7.30 | **0.000** | 4.52 | **0.003** |
| | MCTS vs Heur-GD | -0.29 | 0.780 | -0.87 | 0.412 | 0.82 | 0.441 |
| $k$-means | MCTS vs Random | 7.91 | **0.000** | 4.39 | **0.003** | 0.61 | 0.565 |
| | MCTS vs ETS | 10.83 | **0.000** | 12.93 | **0.000** | 7.42 | **0.000** |
| | MCTS vs Heur-GD | 3.05 | **0.019** | 1.31 | 0.255 | 1.87 | 0.099 |
| $k$-center | MCTS vs Random | 4.70 | **0.003** | 2.32 | 0.051 | 2.85 | **0.024** |
| | MCTS vs ETS | 7.25 | **0.000** | 7.46 | **0.000** | 6.94 | **0.000** |
| | MCTS vs Heur-GD | 1.92 | 0.100 | 0.82 | 0.437 | 3.89 | **0.005** |
| MoF | MCTS vs Random | 4.26 | **0.004** | 2.67 | **0.037** | 2.75 | **0.036** |
| | MCTS vs ETS | 5.86 | **0.001** | 5.69 | **0.001** | 2.57 | **0.045** |
| | MCTS vs Heur-GD | 5.26 | **0.002** | 6.22 | **0.002** | 13.90 | **0.000** |

Table 9: Performance comparison with ACC and BWT metrics for all datasets between MCTS (Ours) and the baselines with various memory selection methods. We provide the metrics for training on all seen task datasets jointly (Joint) as an upper bound. Furthermore, we include the results from a breadth-first search (BFS) with Uniform memory selection for the 5-task datasets. The memory size is set to $M = 10$ and $M = 100$ for the 5-task and 10/20-task datasets respectively. We report the means and standard deviations averaged over 5 seeds.

| Memory | Schedule | Split MNIST | | Split FashionMNIST | | Split notMNIST | |
|---|---|---|---|---|---|---|---|
| | | ACC (%) | BWT (%) | ACC (%) | BWT (%) | ACC (%) | BWT (%) |
| Offline | Joint | 99.75 ± 0.06 | 0.01 ± 0.06 | 99.34 ± 0.08 | -0.01 ± 0.14 | 96.12 ± 0.57 | -0.21 ± 0.71 |
| Uniform | BFS | 98.28 ± 0.49 | -1.84 ± 0.63 | 98.51 ± 0.23 | -1.03 ± 0.28 | 95.54 ± 0.67 | -1.04 ± 0.87 |
| Uniform | Random | 94.91 ± 2.52 | -6.13 ± 3.16 | 95.89 ± 2.03 | -4.33 ± 2.55 | 91.84 ± 1.48 | -5.37 ± 2.12 |
| | ETS | 94.02 ± 4.25 | -7.22 ± 5.33 | 95.81 ± 3.53 | -4.45 ± 4.34 | 91.01 ± 1.39 | -6.16 ± 1.82 |
| | Heur-GD | 96.02 ± 2.32 | -4.64 ± 2.90 | 97.09 ± 0.62 | -2.82 ± 0.84 | 91.26 ± 3.99 | -6.06 ± 4.70 |
| | MCTS | 97.93 ± 0.56 | -2.27 ± 0.71 | 98.27 ± 0.17 | -1.29 ± 0.20 | 94.64 ± 0.39 | -1.47 ± 0.79 |
| k-means | Random | 92.65 ± 1.38 | -8.96 ± 1.74 | 93.11 ± 2.75 | -7.76 ± 3.42 | 93.11 ± 1.01 | -3.78 ± 1.43 |
| | ETS | 92.89 ± 3.53 | -8.66 ± 4.42 | 96.47 ± 0.85 | -3.55 ± 1.07 | 93.80 ± 0.82 | -2.84 ± 0.81 |
| | Heur-GD | 96.28 ± 1.68 | -4.32 ± 2.11 | 95.78 ± 1.50 | -4.46 ± 1.87 | 91.75 ± 0.94 | -5.60 ± 2.07 |
| | MCTS | 98.20 ± 0.16 | -1.94 ± 0.22 | 98.48 ± 0.26 | -1.04 ± 0.31 | 93.61 ± 0.71 | -3.11 ± 0.55 |
| k-center | Random | 95.48 ± 0.82 | -5.40 ± 1.05 | 93.24 ± 2.84 | -7.64 ± 3.51 | 91.70 ± 1.94 | -5.33 ± 2.80 |
| | ETS | 94.84 ± 1.40 | -6.20 ± 1.77 | 97.28 ± 0.50 | -2.58 ± 0.66 | 91.08 ± 2.48 | -6.39 ± 3.46 |
| | Heur-GD | 94.55 ± 2.79 | -6.47 ± 3.50 | 94.08 ± 3.72 | -6.59 ± 4.57 | 92.06 ± 1.20 | -4.70 ± 2.09 |
| | MCTS | 98.24 ± 0.36 | -1.93 ± 0.44 | 98.06 ± 0.35 | -1.59 ± 0.45 | 94.26 ± 0.37 | -1.97 ± 1.02 |
| MoF | Random | 96.96 ± 1.34 | -3.57 ± 1.69 | 96.39 ± 1.69 | -3.66 ± 2.17 | 93.09 ± 1.40 | -3.70 ± 1.76 |
| | ETS | 97.04 ± 1.23 | -3.46 ± 1.50 | 96.48 ± 1.33 | -3.55 ± 1.73 | 92.64 ± 0.87 | -4.57 ± 1.59 |
| | Heur-GD | 96.46 ± 2.41 | -4.09 ± 3.01 | 95.84 ± 0.89 | -4.39 ± 1.15 | 93.24 ± 0.77 | -3.48 ± 1.37 |
| | MCTS | 98.37 ± 0.24 | -1.70 ± 0.28 | 97.84 ± 0.32 | -1.81 ± 0.39 | 94.62 ± 0.42 | -1.80 ± 0.56 |

| Memory | Schedule | Permuted MNIST | | Split CIFAR-100 | | Split miniImagenet | |
|---|---|---|---|---|---|---|---|
| | | ACC (%) | BWT (%) | ACC (%) | BWT (%) | ACC (%) | BWT (%) |
| Offline | Joint | 95.34 ± 0.13 | 0.17 ± 0.18 | 84.73 ± 0.81 | -1.06 ± 0.81 | 74.03 ± 0.83 | 9.70 ± 0.68 |
| Uniform | Random | 72.59 ± 1.52 | -25.71 ± 1.76 | 53.76 ± 1.80 | -35.11 ± 1.93 | 49.89 ± 1.03 | -14.79 ± 1.14 |
| | ETS | 71.09 ± 2.31 | -27.39 ± 2.59 | 47.70 ± 2.16 | -41.69 ± 2.37 | 46.97 ± 1.24 | -18.32 ± 1.34 |
| | Heur-GD | 76.68 ± 2.13 | -20.82 ± 2.41 | 57.31 ± 1.21 | -30.76 ± 1.45 | 49.66 ± 1.10 | -12.04 ± 0.59 |
| | MCTS | 76.34 ± 0.98 | -21.21 ± 1.16 | 56.60 ± 1.13 | -31.39 ± 1.11 | 50.20 ± 0.72 | -13.46 ± 1.22 |
| k-means | Random | 71.91 ± 1.24 | -26.45 ± 1.34 | 53.20 ± 1.44 | -35.77 ± 1.31 | 49.96 ± 1.46 | -14.81 ± 1.18 |
| | ETS | 69.40 ± 1.32 | -29.23 ± 1.47 | 47.51 ± 1.14 | -41.77 ± 1.30 | 45.82 ± 0.92 | -19.53 ± 1.10 |
| | Heur-GD | 75.57 ± 1.18 | -22.11 ± 1.22 | 54.31 ± 3.94 | -33.80 ± 4.24 | 49.25 ± 1.00 | -12.92 ± 1.22 |
| | MCTS | 77.74 ± 0.80 | -19.66 ± 0.95 | 56.95 ± 0.92 | -30.92 ± 0.83 | 50.47 ± 0.85 | -13.31 ± 1.24 |
| k-center | Random | 71.39 ± 1.87 | -27.04 ± 2.05 | 48.29 ± 2.11 | -40.88 ± 2.28 | 44.40 ± 1.35 | -20.03 ± 1.31 |
| | ETS | 69.11 ± 1.69 | -29.58 ± 1.81 | 44.13 ± 1.06 | -45.28 ± 1.04 | 41.35 ± 1.23 | -23.71 ± 1.45 |
| | Heur-GD | 74.33 ± 2.00 | -23.45 ± 2.27 | 50.32 ± 1.97 | -37.99 ± 2.14 | 44.13 ± 0.95 | -18.26 ± 1.05 |
| | MCTS | 76.55 ± 1.16 | -21.06 ± 1.32 | 51.37 ± 1.63 | -37.01 ± 1.62 | 46.76 ± 0.96 | -16.56 ± 0.90 |
| MoF | Random | 78.80 ± 1.07 | -18.79 ± 1.16 | 62.35 ± 1.24 | -26.33 ± 1.25 | 56.02 ± 1.11 | -7.99 ± 1.13 |
| | ETS | 77.62 ± 1.12 | -20.10 ± 1.26 | 60.43 ± 1.17 | -28.22 ± 1.26 | 56.12 ± 1.12 | -8.93 ± 0.83 |
| | Heur-GD | 77.27 ± 1.45 | -20.15 ± 1.63 | 55.60 ± 2.70 | -32.57 ± 2.77 | 52.30 ± 0.59 | -9.61 ± 0.67 |
| | MCTS | 81.58 ± 0.75 | -15.41 ± 0.86 | 64.22 ± 0.65 | -23.48 ± 1.02 | 57.70 ± 0.51 | -5.31 ± 0.55 |

Table 10: Hyperparameters for replay-based methods HAL, MER, DER and DER++ used in experiments on applying MCTS to recent replay-based methods in Section 4.1.

| Method | Hyperparam. | 5-task Datasets | | | 10- and 20-task Datasets | | |
|---|---|---|---|---|---|---|---|
| | | S-MNIST | S-FashionMNIST | S-notMNIST | P-MNIST | S-CIFAR-100 | S-miniImagenet |
| HAL | $\eta$ | 0.1 | 0.1 | 0.1 | 0.1 | 0.03 | 0.03 |
| | $\lambda$ | 0.1 | 0.1 | 0.1 | 0.1 | 1.0 | 0.03 |
| | $\gamma$ | 0.5 | 0.1 | 0.1 | 0.1 | 0.1 | 0.1 |
| | $\beta$ | 0.7 | 0.5 | 0.5 | 0.5 | 0.5 | 0.5 |
| | $k$ | 100 | 100 | 100 | 100 | 100 | 100 |
| MER | $\gamma$ | 1.0 | 1.0 | 1.0 | 1.0 | 1.0 | 1.0 |
| | $\beta$ | 1.0 | 0.01 | 1.0 | 1.0 | 0.1 | 0.1 |
| DER | $\alpha$ | 0.2 | 0.2 | 0.1 | 1.0 | 1.0 | 0.1 |
| DER++ | $\alpha$ | 0.2 | 0.2 | 0.1 | 1.0 | 1.0 | 0.1 |
| | $\beta$ | 1.0 | 1.0 | 1.0 | 1.0 | 1.0 | 1.0 |

### C.5 Applying Scheduling to Recent Replay Methods

In Section 4.1, we showed that MCTS can be applied to any replay method. We combined MCTS together with four recent replay methods, namely Hindsight Anchor Learning (HAL) (Chaudhry et al., 2021), Meta Experience Replay (MER) (Riemer et al., 2019), and Dark Experience Replay (DER) (Buzzega et al., 2020). We present the hyperparameters used for each method in Table 10. The hyperparameters for each method are denoted as

- **HAL.** $\eta$: learning rate, $\lambda$: regularization, $\gamma$: mean embedding strength, $\beta$: decay rate, $k$: gradient steps on anchors
- **MER.** $\gamma$: across batch meta-learning rate, $\beta$: within batch meta-learning rate
- **DER.** $\alpha$: loss coefficient for memory logits
- **DER++.** $\alpha$: loss coefficient for memory logits, $\beta$: loss coefficient for memory labels

For the experiments, we used the same architectures and hyperparameters as described in Appendix C.1 for all datasets if not mentioned otherwise. We used the Adam optimizer with learning rate $\eta = 0.001$ for MER, DER, and DER++. For HAL, we used the SGD optimizer since using Adam made the model diverge in our experiments. Table 11 shows the ACC and BWT for all methods combined with the scheduling from Random, ETS, Heuristic, and MCTS, where red, orange, and yellow highlight indicate the 1st, 2nd, and 3rd-best performing scheduling method based on its mean value for each metric. We observe that MCTS can further improve the performance for each of the replay methods across the different datasets. Table 12 shows statistical significance tests between MCTS and the baselines for every considered replay method, where we note that our method in general achieves significantly higher ACC comparing to the baselines.

Table 11: Performances averaged over 5 seeds between scheduling methods combined with replay-based methods Hindsight Anchor Learning (HAL), Meta Experience Replay (MER), Dark Experience Replay (DER), and DER++. Memory sizes are $M = 10$ and $M = 100$ for the 5-task and 10/20-task datasets respectively. Results on Heuristic where some seed did not converge is denoted by *. Applying MCTS to each method can enhance the performance compared to using the baseline schedules.

| Method | Schedule | Split MNIST ACC (%) | BWT (%) | Split FashionMNIST ACC (%) | BWT (%) | Split notMNIST ACC (%) | BWT (%) |
|---|---|---|---|---|---|---|---|
| HAL | Random | 96.32 ± 1.77 | -3.90 ± 2.28 | 90.42 ± 4.26 | -10.75 ± 5.45 | 93.50 ± 1.10 | -3.14 ± 1.56 |
| | ETS | 97.21 ± 1.25 | -2.80 ± 1.59 | 96.75 ± 0.50 | -2.84 ± 0.75 | 92.16 ± 1.82 | -5.04 ± 2.24 |
| | Heur-GD | 97.69 ± 0.19 | -2.22 ± 0.24 | *74.16 ± 11.19 | *-31.26 ± 14.00 | 93.64 ± 0.93 | -2.80 ± 1.20 |
| | MCTS | 97.96 ± 0.15 | -1.85 ± 0.18 | 97.56 ± 0.51 | -2.02 ± 0.63 | 94.47 ± 0.82 | -1.67 ± 0.64 |
| MER | Random | 93.00 ± 3.22 | -7.96 ± 4.15 | 96.20 ± 2.10 | -2.31 ± 2.59 | 89.10 ± 2.57 | -8.82 ± 3.26 |
| | ETS | 92.97 ± 1.73 | -8.52 ± 2.15 | 84.88 ± 3.85 | -3.34 ± 5.59 | 90.56 ± 0.83 | -6.11 ± 1.06 |
| | Heur-GD | 94.30 ± 2.79 | -6.46 ± 3.50 | 96.91 ± 0.62 | -1.34 ± 0.76 | 90.90 ± 1.30 | -6.24 ± 1.96 |
| | MCTS | 96.44 ± 0.72 | -4.14 ± 0.94 | 86.67 ± 4.09 | 0.85 ± 3.85 | 92.44 ± 0.77 | -3.63 ± 1.06 |
| DER | Random | 95.91 ± 2.18 | -4.40 ± 2.46 | 50.00 ± 0.00 | -12.20 ± 0.07 | 78.76 ± 12.73 | -11.91 ± 4.45 |
| | ETS | 98.17 ± 0.35 | -2.00 ± 0.42 | 97.69 ± 0.58 | -2.05 ± 0.71 | 94.74 ± 1.05 | -1.94 ± 1.17 |
| | Heur-GD | 94.57 ± 1.71 | -6.08 ± 2.09 | *72.49 ± 19.32 | *-20.88 ± 11.46 | *77.88 ± 12.58 | *-12.66 ± 4.17 |
| | MCTS | 99.02 ± 0.10 | -0.91 ± 0.13 | 98.33 ± 0.51 | -1.26 ± 0.63 | 95.02 ± 0.33 | -0.97 ± 0.81 |
| DER++ | Random | 90.09 ± 10.02 | -11.73 ± 12.38 | *50.00 ± 0.00 | *-12.20 ± 0.07 | 61.83 ± 9.84 | -14.40 ± 10.67 |
| | ETS | 97.98 ± 0.52 | -2.24 ± 0.66 | 98.12 ± 0.40 | -1.59 ± 0.52 | 94.53 ± 1.02 | -1.82 ± 1.02 |
| | Heur-GD | 92.35 ± 2.42 | -8.83 ± 2.99 | *67.31 ± 21.20 | *-24.86 ± 16.34 | 93.88 ± 1.33 | -2.86 ± 1.49 |
| | MCTS | 98.84 ± 0.21 | -1.14 ± 0.26 | 98.38 ± 0.43 | -1.17 ± 0.51 | 94.73 ± 0.20 | -1.21 ± 1.12 |

| Method | Schedule | Permuted MNIST ACC (%) | BWT (%) | Split CIFAR-100 ACC (%) | BWT (%) | Split miniImagenet ACC (%) | BWT (%) |
|---|---|---|---|---|---|---|---|
| HAL | Random | 88.93 ± 0.53 | -6.77 ± 0.64 | 35.90 ± 2.47 | -17.37 ± 3.76 | 40.86 ± 1.86 | -5.12 ± 2.23 |
| | ETS | 88.46 ± 0.86 | -7.26 ± 0.90 | 34.90 ± 2.02 | -18.92 ± 0.91 | 38.13 ± 1.18 | -8.19 ± 1.73 |
| | Heur-GD | *66.63 ± 28.50 | *-29.68 ± 27.90 | 35.07 ± 1.29 | -24.76 ± 2.41 | 39.51 ± 1.49 | -5.65 ± 0.77 |
| | MCTS | 89.14 ± 0.74 | -6.29 ± 0.74 | 40.22 ± 1.57 | -12.77 ± 1.30 | 41.39 ± 1.15 | -3.69 ± 1.86 |
| MER | Random | 87.25 ± 0.47 | -8.77 ± 0.59 | 42.68 ± 0.86 | -35.56 ± 1.39 | 32.86 ± 0.95 | -7.71 ± 0.45 |
| | ETS | 73.01 ± 0.96 | -25.19 ± 1.10 | 43.38 ± 1.81 | -34.84 ± 1.98 | 33.58 ± 1.53 | -6.80 ± 1.46 |
| | Heur-GD | 83.86 ± 3.19 | -12.48 ± 3.60 | 40.90 ± 1.70 | -44.10 ± 2.03 | 34.22 ± 1.93 | -7.57 ± 1.63 |
| | MCTS | 79.72 ± 0.71 | -17.42 ± 0.78 | 44.29 ± 0.69 | -32.73 ± 0.88 | 32.74 ± 1.29 | -5.77 ± 1.04 |
| DER | Random | 90.67 ± 0.31 | -5.20 ± 0.30 | 56.17 ± 1.30 | -29.03 ± 1.38 | 35.13 ± 4.11 | -10.85 ± 2.92 |
| | ETS | 85.71 ± 0.75 | -11.15 ± 0.87 | 52.58 ± 1.49 | -32.93 ± 2.04 | 35.50 ± 2.84 | -10.94 ± 2.21 |
| | Heur-GD | 81.56 ± 2.28 | -15.06 ± 2.51 | 55.75 ± 1.08 | -31.27 ± 1.02 | 43.62 ± 0.88 | -8.18 ± 1.16 |
| | MCTS | 90.11 ± 0.18 | -5.89 ± 0.23 | 58.99 ± 0.98 | -24.95 ± 0.64 | 43.46 ± 0.95 | -9.32 ± 1.37 |
| DER++ | Random | 89.83 ± 0.92 | -6.03 ± 0.98 | 60.90 ± 0.89 | -23.45 ± 1.34 | 46.78 ± 1.96 | 3.28 ± 1.35 |
| | ETS | 85.25 ± 0.88 | -11.60 ± 1.03 | 52.54 ± 1.06 | -33.22 ± 1.51 | 41.36 ± 2.90 | -4.07 ± 2.28 |
| | Heur-GD | 79.17 ± 2.44 | -17.68 ± 2.68 | 56.70 ± 1.27 | -30.33 ± 1.41 | 45.73 ± 0.84 | -6.09 ± 1.24 |
| | MCTS | 89.84 ± 0.22 | -6.13 ± 0.29 | 59.23 ± 0.83 | -24.61 ± 0.91 | 49.45 ± 0.68 | -3.12 ± 0.89 |

Table 12: Two-tailed Welch's $t$-test results for the replay methods results presented in Table 11. We bold the $p$-values where $p < 0.05$ to indicate for when a method is statistically significantly better than its comparison.

| Memory | Schedule | Split MNIST $t$ | $p$ | Split FashionMNIST $t$ | $p$ | Split notMNIST $t$ | $p$ |
|---|---|---|---|---|---|---|---|
| HAL | MCTS vs Random | 1.84 | 0.139 | 3.33 | **0.028** | 1.41 | 0.198 |
| | MCTS vs ETS | 1.20 | 0.295 | 2.27 | 0.053 | 2.31 | 0.064 |
| | MCTS vs Heur-GD | 2.26 | 0.056 | 4.18 | **0.014** | 1.34 | 0.218 |
| MER | MCTS vs Random | 2.08 | 0.099 | -4.15 | **0.006** | 2.48 | 0.059 |
| | MCTS vs ETS | 3.71 | **0.012** | 0.64 | 0.542 | 3.32 | **0.011** |
| | MCTS vs Heur-GD | 1.48 | 0.204 | -4.96 | **0.007** | 2.04 | 0.084 |
| DER | MCTS vs Random | 2.85 | **0.046** | 190.21 | **0.000** | 2.56 | 0.063 |
| | MCTS vs ETS | 4.64 | **0.007** | 1.64 | 0.139 | 0.51 | 0.635 |
| | MCTS vs Heur-GD | 5.21 | **0.006** | 2.67 | 0.055 | 2.73 | 0.053 |
| DER++ | MCTS vs Random | 1.75 | 0.156 | 227.62 | **0.000** | 6.68 | **0.003** |
| | MCTS vs ETS | 3.09 | **0.026** | 0.89 | 0.397 | 0.39 | 0.712 |
| | MCTS vs Heur-GD | 5.36 | **0.006** | 2.93 | **0.043** | 1.26 | 0.272 |

| Memory | Schedule | Permuted MNIST $t$ | $p$ | Split CIFAR-100 $t$ | $p$ | Split miniImagenet $t$ | $p$ |
|---|---|---|---|---|---|---|---|
| HAL | MCTS vs Random | 0.46 | 0.657 | 2.96 | 0.022 | 0.49 | 0.640 |
| | MCTS vs ETS | 1.20 | 0.266 | 4.16 | **0.004** | 3.97 | **0.004** |
| | MCTS vs Heur-GD | 1.58 | 0.189 | 5.07 | **0.001** | 2.01 | 0.082 |
| MER | MCTS vs Random | -17.61 | **0.000** | 2.92 | **0.020** | -0.14 | 0.889 |
| | MCTS vs ETS | 11.24 | **0.000** | 0.95 | 0.386 | -0.84 | 0.425 |
| | MCTS vs Heur-GD | -2.54 | 0.059 | 3.70 | **0.013** | -1.27 | 0.244 |
| DER | MCTS vs Random | -3.12 | **0.019** | 3.46 | **0.010** | 3.96 | **0.014** |
| | MCTS vs ETS | 11.36 | **0.000** | 7.18 | **0.000** | 5.31 | **0.003** |
| | MCTS vs Heur-GD | 7.47 | **0.002** | 4.44 | **0.002** | -0.25 | 0.809 |
| DER++ | MCTS vs Random | 0.03 | 0.981 | -2.75 | **0.025** | 2.57 | 0.051 |
| | MCTS vs ETS | 10.17 | **0.000** | 9.94 | **0.000** | 5.43 | **0.004** |
| | MCTS vs Heur-GD | 8.72 | **0.001** | 3.34 | **0.013** | 6.89 | **0.000** |

Table 13: Performance comparison with ACC and BWT metrics for all datasets between MCTS and the baselines in the setting where only 1 sample per class can be replayed. The memory sizes are set to $M = 10$ and $M = 100$ for the 5-task and 10/20-task datasets respectively. MCTS (Ours) and Random uses $M = 2$ and $M = 50$ for the 5-task and 10/20-task datasets respectively. We report the means and standard deviations averaged over 5 seeds. MCTS performs on par with the best baselines for both metrics on all datasets, except on Permuted MNIST where MCTS outperforms the baselines.

| | Split MNIST | | Split FashionMNIST | | Split notMNIST | |
|---|---|---|---|---|---|---|
| **Method** | ACC (%) | BWT (%) | ACC (%) | BWT (%) | ACC (%) | BWT (%) |
| Random | 92.56 ± 2.90 | -8.97 ± 3.62 | 92.70 ± 3.78 | -8.24 ± 4.75 | 89.53 ± 3.96 | -8.13 ± 5.02 |
| A-GEM | 94.97 ± 1.50 | -6.03 ± 1.87 | 94.81 ± 0.86 | -5.65 ± 1.06 | 92.27 ± 1.16 | -4.17 ± 1.39 |
| ER-Ring | 94.94 ± 1.56 | -6.07 ± 1.92 | 95.83 ± 2.15 | -4.38 ± 2.59 | 91.10 ± 1.89 | -6.27 ± 2.35 |
| Uniform | 95.77 ± 1.12 | -5.02 ± 1.39 | 97.12 ± 1.57 | -2.79 ± 1.98 | 92.14 ± 1.45 | -4.90 ± 1.41 |
| MCTS | 96.07 ± 1.60 | -4.59 ± 2.01 | 97.17 ± 0.78 | -2.64 ± 0.99 | 93.41 ± 1.11 | -3.36 ± 1.56 |

| | Permuted MNIST | | Split CIFAR-100 | | Split miniImagenet | |
|---|---|---|---|---|---|---|
| **Method** | ACC (%) | BWT (%) | ACC (%) | BWT (%) | ACC (%) | BWT (%) |
| Random | 70.02 ± 1.76 | -28.22 ± 1.92 | 48.62 ± 1.02 | -39.95 ± 1.10 | 48.85 ± 1.38 | -14.55 ± 1.86 |
| A-GEM | 64.71 ± 1.78 | -34.41 ± 2.05 | 42.22 ± 2.13 | -46.90 ± 2.21 | 32.06 ± 1.83 | -30.81 ± 1.79 |
| ER-Ring | 69.73 ± 1.13 | -28.87 ± 1.29 | 53.93 ± 1.13 | -34.91 ± 1.18 | 49.82 ± 1.69 | -14.38 ± 1.57 |
| Uniform | 69.85 ± 1.01 | -28.74 ± 1.17 | 52.63 ± 1.62 | -36.43 ± 1.81 | 50.56 ± 1.07 | -13.52 ± 1.34 |
| MCTS | 72.52 ± 0.54 | -25.43 ± 0.65 | 51.50 ± 1.19 | -37.01 ± 1.08 | 50.70 ± 0.54 | -12.60 ± 1.13 |

## C.6 Efficiency of Replay Scheduling

We illustrate the efficiency of replay scheduling in a setting where only 1 sample/class is available from the historical data for replay. Table 13 shows that MCTS, despite using significantly fewer samples for replay, performs mostly on par with the baselines and outperforms them on Permuted MNIST. Table 14 shows statistical significance tests between MCTS and the baselines for the corresponding results. Our method mostly performs on par with the baselines, except on Split CIFAR-100, but is significantly better than the baselines on Permuted MNIST.

We visualize the memory usage in the experiment on efficiency of replay scheduling in Section 4.1. For the 5-task datasets, the replay memory size for MCTS is set to $M = 2$, such that only 2 samples can be selected for replay at all times. Similarly, we set $M = 50$ for the 10- and 20-task datasets which have 100 classes to learn in total. The baselines A-GEM (Chaudhry et al., 2018b), ER-Ring (Chaudhry et al., 2019), and Uniform use an incremental memory in order to replay 1 sample/class at all tasks. We visualize the memory usage for our method and the baselines for the 5-task datasets in Figure 8. Here, the memory capacity is reached at task 2, while the baselines must increment their memory size. Figure 9 shows the memory usage for Permuted MNIST and the 20-task datasets Split CIFAR-100 and Split miniImagenet.

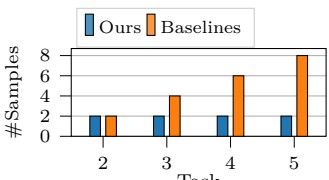

Figure 8: Number of replayed samples per task for the 5-task datasets in the tiny memory setting.

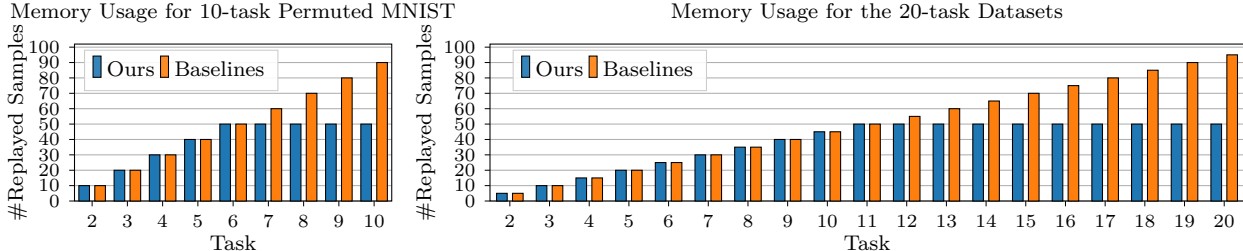

Figure 9: Number of replayed samples per task for 10-task Permuted MNIST (top) and the 20-task datasets in the experiment in Section 4.1. The fixed memory size $M = 50$ for our method is reached after learning task 6 and task 11 on the Permuted MNIST and the 20-task datasets respectively, while the baselines continue incrementing their number of replay samples per task.

Table 14: Two-tailed Welch's $t$-test results for efficiency of replay scheduling presented in Table 13. We bold the $p$-values where $p < 0.05$ to indicate for when a method is statistically significantly better than its comparison. MCTS scheduling is statistically indistinguishable with the baselines on most datasets except Split CIFAR-100, but is significantly better than the baselines on Permuted MNIST.

| Methods | Split MNIST | | Split FashionMNIST | | Split notMNIST | |
|---|---|---|---|---|---|---|
| | $t$ | $p$ | $t$ | $p$ | $t$ | $p$ |
| MCTS vs Random | 2.12 | 0.077 | 2.32 | 0.076 | 1.89 | 0.122 |
| MCTS vs A-GEM | 1.00 | 0.347 | 4.08 | **0.004** | 1.41 | 0.195 |
| MCTS vs ER-Ring | 1.01 | 0.342 | 1.18 | 0.292 | 2.11 | 0.076 |
| MCTS vs Uniform | 0.30 | 0.770 | 0.07 | 0.950 | 1.39 | 0.203 |

| Methods | Permuted MNIST | | Split CIFAR-100 | | Split miniImagenet | |
|---|---|---|---|---|---|---|
| | $t$ | $p$ | $t$ | $p$ | $t$ | $p$ |
| MCTS vs Random | 2.72 | **0.044** | 3.67 | **0.007** | 2.48 | 0.054 |
| MCTS vs A-GEM | 8.40 | **0.001** | 7.60 | **0.000** | 19.52 | **0.000** |
| MCTS vs ER-Ring | 4.46 | **0.005** | -2.96 | **0.018** | 0.99 | 0.369 |
| MCTS vs Uniform | 4.68 | **0.003** | -1.13 | **0.296** | 0.23 | 0.824 |

## C.7 Varying Memory Size in Different Continual Learning Setting

Here, we provide the ACC and BWT metrics from the varying memory size experiments from figure 3 in Section 4.1. We also provide the p-values from Welch's t-tests to show the statistical significance between the methods.

- **Domain-IL, Permuted MNIST:** Metrics in Table 15, t-tests in Table 16.
- **Task-IL, Split MNIST/FashionMNIST/notMNIST:** Metrics in Table 17, t-tests in Table 18.
- **Task-IL, Split CIFAR-100/miniImagenet:** Metrics in Table 19, t-tests in Table 20.
- **Class-IL, Split MNIST/FashionMNIST/notMNIST:** Metrics in Table 21, t-tests in Table 22.
- **Class-IL, Split CIFAR-100/miniImagenet:** Metrics in Table 23, t-tests in Table 24.

We note that MCTS mostly performs significantly better than the baselines on the 10-task Permuted MNIST and, interestingly, on the 20-task Split CIFAR-100 and Split miniImagenet in both Task-IL and Clas-IL settings, which shows the importance of replay scheduling for CL datasets with long task horizons.

Table 15: Performance comparison in the Domain Incremental Learning setting over various memory sizes for the methods on Permuted MNIST.

| Memory Size | Method | Permuted MNIST | |
|---|---|---|---|
| | | ACC (%) | BWT (%) |
| M=90 | Random | 72.63 ± 1.06 | -25.62 ± 1.20 |
| | ETS | 71.49 ± 1.15 | -26.92 ± 1.26 |
| | Heur-GD | 75.50 ± 1.64 | -22.14 ± 1.93 |
| | MCTS | 71.66 ± 0.67 | -28.70 ± 0.81 |
| M=270 | Random | 82.01 ± 0.79 | -15.24 ± 0.86 |
| | ETS | 80.68 ± 0.66 | -16.72 ± 0.77 |
| | Heur-GD | 78.32 ± 1.58 | -19.01 ± 1.83 |
| | MCTS | 82.08 ± 0.35 | -17.18 ± 0.38 |
| M=450 | Random | 84.72 ± 1.39 | -12.16 ± 1.53 |
| | ETS | 84.38 ± 1.12 | -12.54 ± 1.20 |
| | Heur-GD | 81.66 ± 2.30 | -15.27 ± 2.48 |
| | MCTS | 85.53 ± 0.42 | -13.34 ± 0.49 |
| M=900 | Random | 88.02 ± 1.03 | -8.51 ± 1.09 |
| | ETS | 88.02 ± 0.46 | -8.52 ± 0.47 |
| | Heur-GD | 80.27 ± 3.26 | -16.77 ± 3.76 |
| | MCTS | 88.94 ± 0.43 | -9.55 ± 0.47 |
| M=2250 | Random | 90.94 ± 0.46 | -5.26 ± 0.43 |
| | ETS | 91.07 ± 0.23 | -5.11 ± 0.21 |
| | Heur-GD | 81.28 ± 3.79 | -15.68 ± 4.32 |
| | MCTS | 92.45 ± 0.34 | -5.63 ± 0.41 |

Table 16: Two-tailed Welch's $t$-test results for the varying memory size experiments presented in Table 17 in the Domain Incremental Learning setting on Permuted MNIST. We bold the $p$-values where $p < 0.05$ to indicate for when a method is statistically significantly better than its comparison.

| Memory size | Methods | Permuted MNIST | |
|---|---|---|---|
| | | $t$ | $p$ |
| M=90 | MCTS vs Random | -1.55 | 0.166 |
| | MCTS vs ETS | 0.26 | 0.806 |
| | MCTS vs Heur-GD | -4.32 | **0.007** |
| M=270 | MCTS vs Random | 0.17 | 0.872 |
| | MCTS vs ETS | 3.75 | **0.009** |
| | MCTS vs Heur-GD | 4.66 | **0.008** |
| M=450 | MCTS vs Random | 1.13 | 0.313 |
| | MCTS vs ETS | 1.93 | 0.111 |
| | MCTS vs Heur-GD | 3.31 | **0.027** |
| M=900 | MCTS vs Random | 1.66 | 0.153 |
| | MCTS vs ETS | 2.92 | **0.019** |
| | MCTS vs Heur-GD | 5.28 | **0.006** |
| M=2250 | MCTS vs Random | 5.31 | **0.001** |
| | MCTS vs ETS | 6.74 | **0.000** |
| | MCTS vs Heur-GD | 5.88 | **0.004** |

Table 17: Performance comparison in the Task Incremental Learning setting over various memory sizes for the methods on the 5-task datasets.

| Memory Size | Method | Split MNIST | | Split FashionMNIST | | Split notMNIST | |
|---|---|---|---|---|---|---|---|
| | | ACC (%) | BWT (%) | ACC (%) | BWT (%) | ACC (%) | BWT (%) |
| M=8 | Random | 95.50 ± 1.92 | -5.38 ± 2.38 | 95.04 ± 2.43 | -5.42 ± 3.06 | 91.36 ± 1.76 | -6.15 ± 1.98 |
| | ETS | 95.83 ± 1.23 | -4.96 ± 1.53 | 97.25 ± 0.68 | -2.66 ± 0.82 | 92.10 ± 1.53 | -5.10 ± 1.97 |
| | Heur-GD | 96.16 ± 0.83 | -4.45 ± 1.05 | 96.29 ± 0.79 | -3.83 ± 1.05 | 93.68 ± 1.11 | -3.06 ± 1.54 |
| | MCTS | 98.17 ± 0.48 | -1.99 ± 0.61 | 98.33 ± 0.11 | -1.27 ± 0.11 | 94.53 ± 0.62 | -1.67 ± 1.03 |
| M=24 | Random | 97.21 ± 1.70 | -3.24 ± 2.14 | 97.20 ± 0.71 | -2.72 ± 0.92 | 92.31 ± 1.50 | -4.88 ± 1.75 |
| | ETS | 97.87 ± 0.58 | -2.41 ± 0.74 | 97.84 ± 0.55 | -1.91 ± 0.64 | 93.19 ± 1.41 | -3.84 ± 1.48 |
| | Heur-GD | 97.82 ± 1.17 | -2.40 ± 1.46 | 95.76 ± 2.84 | -4.48 ± 3.52 | 92.97 ± 2.55 | -3.54 ± 3.07 |
| | MCTS | 98.58 ± 0.32 | -1.47 ± 0.41 | 98.48 ± 0.24 | -1.06 ± 0.32 | 95.23 ± 0.66 | -1.08 ± 1.72 |
| M=80 | Random | 97.57 ± 2.03 | -2.78 ± 2.53 | 96.85 ± 1.58 | -3.10 ± 2.05 | 94.14 ± 0.95 | -2.50 ± 1.59 |
| | ETS | 98.47 ± 0.40 | -1.65 ± 0.49 | 97.93 ± 0.85 | -1.76 ± 0.98 | 94.63 ± 0.67 | -1.82 ± 0.74 |
| | Heur-GD | 98.36 ± 0.40 | -1.72 ± 0.50 | 96.22 ± 1.96 | -3.92 ± 2.45 | 93.24 ± 2.56 | -3.73 ± 2.71 |
| | MCTS | 99.06 ± 0.16 | -0.89 ± 0.21 | 98.60 ± 0.24 | -0.87 ± 0.34 | 94.84 ± 0.58 | -0.97 ± 1.28 |
| M=120 | Random | 97.82 ± 2.29 | -2.47 ± 2.88 | 98.29 ± 0.44 | -1.26 ± 0.56 | 94.91 ± 1.03 | -1.65 ± 1.04 |
| | ETS | 98.82 ± 0.18 | -1.22 ± 0.26 | 98.53 ± 0.28 | -0.96 ± 0.36 | 95.32 ± 0.66 | -1.21 ± 1.30 |
| | Heur-GD | 98.37 ± 0.38 | -1.71 ± 0.48 | 95.26 ± 3.14 | -5.12 ± 3.98 | 93.42 ± 1.78 | -3.47 ± 2.22 |
| | MCTS | 99.05 ± 0.11 | -0.88 ± 0.15 | 98.75 ± 0.19 | -0.77 ± 0.25 | 94.16 ± 1.08 | -1.99 ± 1.69 |
| M=200 | Random | 97.99 ± 1.59 | -2.25 ± 2.00 | 96.68 ± 3.33 | -3.38 ± 4.19 | 93.94 ± 1.40 | -2.12 ± 1.70 |
| | ETS | 98.83 ± 0.23 | -1.19 ± 0.28 | 98.60 ± 0.25 | -0.99 ± 0.29 | 94.79 ± 0.50 | -1.58 ± 1.03 |
| | Heur-GD | 98.15 ± 0.64 | -1.97 ± 0.81 | 95.83 ± 2.00 | -4.40 ± 2.52 | 93.88 ± 1.63 | -2.71 ± 1.69 |
| | MCTS | 99.09 ± 0.08 | -0.83 ± 0.11 | 98.83 ± 0.11 | -0.65 ± 0.15 | 95.19 ± 0.53 | -0.49 ± 0.47 |
| M=400 | Random | 97.98 ± 2.23 | -2.28 ± 2.80 | 98.00 ± 1.59 | -1.74 ± 2.00 | 94.68 ± 1.11 | -1.77 ± 1.09 |
| | ETS | 99.18 ± 0.10 | -0.78 ± 0.13 | 98.83 ± 0.09 | -0.71 ± 0.12 | 95.41 ± 0.56 | -0.20 ± 1.07 |
| | Heur-GD | 98.44 ± 0.60 | -1.64 ± 0.77 | 95.41 ± 3.77 | -4.93 ± 4.69 | 92.84 ± 1.88 | -3.95 ± 2.05 |
| | MCTS | 99.25 ± 0.05 | -0.63 ± 0.09 | 98.80 ± 0.11 | -0.62 ± 0.17 | 95.25 ± 0.44 | -0.97 ± 1.20 |
| M=800 | Random | 98.60 ± 1.42 | -1.51 ± 1.77 | 97.00 ± 3.93 | -2.98 ± 4.94 | 94.97 ± 0.73 | -1.86 ± 0.77 |
| | ETS | 99.34 ± 0.06 | -0.57 ± 0.04 | 98.97 ± 0.06 | -0.51 ± 0.10 | 95.52 ± 0.74 | -0.61 ± 1.21 |
| | Heur-GD | 98.76 ± 0.41 | -1.23 ± 0.55 | 93.32 ± 3.67 | -7.54 ± 4.65 | 95.06 ± 1.44 | -1.17 ± 1.30 |
| | MCTS | 99.38 ± 0.08 | -0.45 ± 0.10 | 98.96 ± 0.12 | -0.45 ± 0.18 | 95.46 ± 0.70 | -0.48 ± 0.81 |

Table 18: Two-tailed Welch's $t$-test results for the varying memory size experiments presented in Table 17 in the Task Incremental Learning setting on the 5-task datasets. We bold the $p$-values where $p < 0.05$ to indicate for when a method is statistically significantly better than its comparison.

| Memory size | Methods | Split MNIST | | Split FashionMNIST | | Split notMNIST | |
|---|---|---|---|---|---|---|---|
| | | $t$ | $p$ | $t$ | $p$ | $t$ | $p$ |
| M=8 | MCTS vs Random | 2.70 | **0.048** | 2.70 | 0.054 | 3.41 | **0.019** |
| | MCTS vs ETS | 3.53 | **0.016** | 3.13 | **0.033** | 2.94 | **0.030** |
| | MCTS vs Heur-GD | 4.18 | **0.005** | 5.12 | **0.006** | 1.34 | 0.227 |
| M=24 | MCTS vs Random | 1.59 | 0.183 | 3.39 | **0.020** | 3.55 | **0.014** |
| | MCTS vs ETS | 2.15 | 0.074 | 2.14 | **0.080** | 2.63 | **0.041** |
| | MCTS vs Heur-GD | 1.26 | 0.268 | 1.90 | 0.128 | 1.72 | 0.153 |
| M=80 | MCTS vs Random | 1.47 | 0.215 | 2.18 | 0.091 | 1.26 | 0.251 |
| | MCTS vs ETS | 2.76 | **0.038** | 1.53 | 0.191 | 0.48 | 0.645 |
| | MCTS vs Heur-GD | 3.26 | **0.021** | 2.41 | 0.072 | 1.22 | 0.285 |
| M=120 | MCTS vs Random | 1.07 | 0.344 | 1.95 | 0.105 | -1.01 | 0.343 |
| | MCTS vs ETS | 2.12 | 0.073 | 1.31 | 0.231 | -1.83 | 0.112 |
| | MCTS vs Heur-GD | 3.45 | **0.020** | 2.22 | 0.090 | 0.70 | 0.507 |
| M=200 | MCTS vs Random | 1.38 | 0.240 | 1.29 | 0.266 | 1.67 | 0.154 |
| | MCTS vs ETS | 2.06 | 0.094 | 1.69 | 0.146 | 1.11 | 0.301 |
| | MCTS vs Heur-GD | 2.91 | **0.042** | 3.00 | **0.040** | 1.53 | 0.190 |
| M=400 | MCTS vs Random | 1.14 | 0.318 | 1.00 | 0.373 | 0.94 | 0.389 |
| | MCTS vs ETS | 1.25 | 0.258 | -0.42 | 0.684 | -0.47 | 0.655 |
| | MCTS vs Heur-GD | 2.69 | 0.054 | 1.80 | 0.146 | 2.49 | 0.061 |
| M=800 | MCTS vs Random | 1.10 | 0.332 | 1.00 | 0.373 | 0.96 | 0.364 |
| | MCTS vs ETS | 0.82 | 0.435 | -0.06 | 0.955 | -0.11 | 0.912 |
| | MCTS vs Heur-GD | 3.05 | **0.035** | 3.08 | **0.037** | 0.50 | 0.636 |

Table 19: Performance comparison in the Task Incremental Learning setting over various memory sizes for the methods on the 20-task datasets.

| Memory Size | Method | Split CIFAR-100 | | Split miniImagenet | |
|---|---|---|---|---|---|
| | | ACC (%) | BWT (%) | ACC (%) | BWT (%) |
| M=95 | Random | 56.92 ± 0.72 | -31.98 ± 0.69 | 52.06 ± 0.94 | -12.49 ± 1.41 |
| | ETS | 55.19 ± 0.77 | -33.70 ± 0.82 | 51.33 ± 2.01 | -13.32 ± 2.15 |
| | Heur-GD | 53.86 ± 3.89 | -34.51 ± 4.25 | 50.66 ± 1.36 | -10.99 ± 1.39 |
| | MCTS | 60.46 ± 1.05 | -27.67 ± 1.23 | 53.59 ± 0.24 | -9.49 ± 0.51 |
| M=285 | Random | 67.79 ± 1.19 | -20.31 ± 1.12 | 57.85 ± 1.09 | -6.33 ± 1.30 |
| | ETS | 65.02 ± 0.98 | -23.36 ± 1.10 | 56.18 ± 0.83 | -8.43 ± 1.37 |
| | Heur-GD | 60.26 ± 2.88 | -27.44 ± 2.87 | 53.33 ± 2.21 | -7.96 ± 2.18 |
| | MCTS | 68.85 ± 0.56 | -18.41 ± 0.63 | 57.91 ± 0.09 | -5.18 ± 0.54 |
| M=475 | Random | 70.86 ± 1.24 | -17.19 ± 1.26 | 59.54 ± 1.25 | -4.22 ± 1.96 |
| | ETS | 69.40 ± 0.73 | -18.68 ± 0.82 | 59.60 ± 0.98 | -4.75 ± 1.31 |
| | Heur-GD | 65.00 ± 3.08 | -22.58 ± 3.20 | 50.12 ± 4.60 | -11.79 ± 5.08 |
| | MCTS | 72.93 ± 0.54 | -14.15 ± 0.78 | 60.00 ± 0.48 | -2.85 ± 0.34 |
| M=950 | Random | 75.39 ± 0.46 | -12.24 ± 0.46 | 61.87 ± 0.85 | -2.32 ± 1.06 |
| | ETS | 74.24 ± 0.61 | -13.49 ± 0.44 | 60.51 ± 0.95 | -4.01 ± 1.28 |
| | Heur-GD | 65.97 ± 5.51 | -21.40 ± 5.58 | 50.99 ± 4.64 | -11.21 ± 4.62 |
| | MCTS | 76.65 ± 0.62 | -10.31 ± 0.84 | 61.82 ± 0.69 | -1.38 ± 0.55 |
| M=1900 | Random | 78.86 ± 0.38 | -8.64 ± 0.37 | 62.95 ± 0.69 | -1.56 ± 0.99 |
| | ETS | 77.66 ± 0.38 | -9.72 ± 0.36 | 62.38 ± 0.68 | -1.55 ± 0.90 |
| | Heur-GD | 63.70 ± 6.12 | -24.07 ± 6.60 | 56.42 ± 3.15 | -5.43 ± 3.63 |
| | MCTS | 79.24 ± 0.59 | -7.43 ± 0.64 | 63.83 ± 1.04 | 0.41 ± 1.10 |

Table 20: Two-tailed Welch's $t$-test results for the varying memory size experiments presented in Table 19 in the Task Incremental Learning setting on the 20-task datasets. We bold the $p$-values where $p < 0.05$ to indicate for when a method is statistically significantly better than its comparison.

| Memory size | Methods | Split CIFAR-100 | | Split miniImagenet | |
|---|---|---|---|---|---|
| | | $t$ | $p$ | $t$ | $p$ |
| M=95 | MCTS vs Random | 5.56 | **0.001** | 3.15 | **0.029** |
| | MCTS vs ETS | 8.11 | **0.000** | 2.23 | 0.088 |
| | MCTS vs Heur-GD | 3.27 | **0.025** | 4.24 | **0.012** |
| M=285 | MCTS vs Random | 1.62 | 0.159 | 0.11 | 0.918 |
| | MCTS vs ETS | 6.76 | **0.000** | 4.14 | **0.014** |
| | MCTS vs Heur-GD | 5.87 | **0.003** | 4.13 | **0.014** |
| M=475 | MCTS vs Random | 3.05 | **0.025** | 0.68 | 0.525 |
| | MCTS vs ETS | 7.78 | **0.000** | 0.73 | 0.496 |
| | MCTS vs Heur-GD | 5.07 | **0.006** | 4.28 | **0.012** |
| M=950 | MCTS vs Random | 3.25 | **0.013** | -0.09 | 0.929 |
| | MCTS vs ETS | 5.51 | **0.001** | 2.25 | 0.058 |
| | MCTS vs Heur-GD | 3.85 | **0.017** | 4.62 | **0.009** |
| M=1900 | MCTS vs Random | 1.08 | 0.317 | 1.41 | 0.203 |
| | MCTS vs ETS | 4.53 | **0.003** | 2.34 | 0.052 |
| | MCTS vs Heur-GD | 5.05 | **0.007** | 4.47 | **0.007** |

Table 21: Performance comparison in the Class Incremental Learning setting over various memory sizes for the methods on the 5-task datasets.

| Memory Size | Method | Split MNIST | | Split FashionMNIST | | Split notMNIST | |
|---|---|---|---|---|---|---|---|
| | | ACC (%) | BWT (%) | ACC (%) | BWT (%) | ACC (%) | BWT (%) |
| M=10 | Random | 25.53 ± 1.57 | -92.74 ± 1.99 | 30.78 ± 2.74 | -85.70 ± 3.47 | 30.48 ± 5.16 | -81.80 ± 6.18 |
| | ETS | 25.10 ± 1.04 | -93.27 ± 1.32 | 30.67 ± 2.36 | -85.83 ± 3.03 | 31.67 ± 6.03 | -80.22 ± 7.36 |
| | Heur-GD | 25.05 ± 1.51 | -93.27 ± 1.85 | 31.81 ± 3.17 | -84.33 ± 3.96 | 29.86 ± 3.49 | -82.07 ± 4.26 |
| | MCTS | 28.19 ± 2.18 | -89.34 ± 2.71 | 33.77 ± 2.71 | -81.91 ± 3.45 | 37.23 ± 3.53 | -73.93 ± 3.89 |
| M=20 | Random | 34.13 ± 2.04 | -81.98 ± 2.57 | 37.80 ± 2.92 | -76.83 ± 3.62 | 42.78 ± 1.24 | -65.75 ± 1.92 |
| | ETS | 35.17 ± 2.06 | -80.65 ± 2.54 | 38.76 ± 1.58 | -75.63 ± 1.93 | 45.21 ± 5.92 | -63.10 ± 7.33 |
| | Heur-GD | 33.87 ± 2.15 | -82.20 ± 2.65 | 42.52 ± 1.98 | -70.94 ± 2.47 | 45.13 ± 3.68 | -63.32 ± 4.31 |
| | MCTS | 39.62 ± 0.73 | -75.05 ± 0.92 | 44.27 ± 2.19 | -68.71 ± 2.74 | 47.83 ± 0.82 | -59.91 ± 1.11 |
| M=40 | Random | 45.34 ± 3.71 | -67.96 ± 4.64 | 45.78 ± 2.96 | -66.84 ± 3.72 | 55.70 ± 2.96 | -49.58 ± 3.36 |
| | ETS | 48.57 ± 1.90 | -63.93 ± 2.37 | 45.54 ± 1.21 | -67.15 ± 1.48 | 58.07 ± 3.88 | -46.85 ± 4.92 |
| | Heur-GD | 45.22 ± 4.54 | -68.05 ± 5.65 | 51.67 ± 1.59 | -59.43 ± 2.02 | 53.90 ± 6.10 | -52.09 ± 7.73 |
| | MCTS | 50.79 ± 2.45 | -61.07 ± 3.08 | 51.33 ± 2.96 | -59.87 ± 3.70 | 62.55 ± 3.23 | -41.20 ± 4.81 |
| M=100 | Random | 59.53 ± 6.37 | -50.12 ± 7.96 | 55.96 ± 3.31 | -53.98 ± 4.17 | 65.06 ± 3.48 | -37.86 ± 4.30 |
| | ETS | 68.09 ± 1.06 | -39.49 ± 1.33 | 59.26 ± 0.95 | -49.92 ± 1.14 | 72.55 ± 1.65 | -28.88 ± 2.31 |
| | Heur-GD | 49.81 ± 1.58 | -62.17 ± 1.95 | 58.14 ± 2.96 | -51.24 ± 3.71 | 58.28 ± 2.02 | -46.70 ± 2.63 |
| | MCTS | 66.86 ± 2.21 | -40.96 ± 2.75 | 60.97 ± 2.43 | -47.63 ± 2.99 | 71.18 ± 3.58 | -30.27 ± 4.15 |
| M=200 | Random | 65.82 ± 7.50 | -42.27 ± 9.36 | 61.11 ± 5.19 | -47.26 ± 6.48 | 67.07 ± 5.65 | -35.36 ± 6.96 |
| | ETS | 77.73 ± 1.31 | -27.42 ± 1.63 | 66.51 ± 0.75 | -40.60 ± 0.99 | 77.15 ± 0.24 | -22.61 ± 0.89 |
| | Heur-GD | 53.85 ± 0.88 | -57.05 ± 1.11 | 54.69 ± 0.32 | -55.42 ± 0.43 | 57.05 ± 3.28 | -47.66 ± 4.09 |
| | MCTS | 78.14 ± 1.67 | -26.81 ± 2.00 | 67.38 ± 2.60 | -39.41 ± 3.26 | 75.91 ± 4.88 | -24.15 ± 5.98 |

Table 22: Two-tailed Welch's $t$-test results for the varying memory size experiments presented in Table 21 in the Class Incremental Learning setting on the 5-task datasets. We bold the $p$-values where $p < 0.05$ to indicate for when a method is statistically significantly better than its comparison.

| Memory size | Methods | Split MNIST | | Split FashionMNIST | | Split notMNIST | |
|---|---|---|---|---|---|---|---|
| | | $t$ | $p$ | $t$ | $p$ | $t$ | $p$ |
| M=10 | MCTS vs Random | 1.98 | 0.086 | 1.56 | 0.158 | 2.16 | 0.067 |
| | MCTS vs ETS | 2.56 | **0.045** | 1.73 | 0.123 | 1.59 | 0.159 |
| | MCTS vs Heur-GD | 2.37 | **0.049** | 0.94 | 0.374 | 2.97 | **0.018** |
| M=20 | MCTS vs Random | 5.06 | **0.004** | 3.54 | **0.009** | 6.80 | **0.000** |
| | MCTS vs ETS | 4.07 | **0.010** | 4.07 | **0.004** | 0.88 | 0.429 |
| | MCTS vs Heur-GD | 5.08 | **0.004** | 1.18 | 0.271 | 1.43 | 0.219 |
| M=40 | MCTS vs Random | 2.45 | **0.044** | 2.65 | **0.029** | 3.13 | **0.014** |
| | MCTS vs ETS | 1.43 | 0.192 | 3.62 | **0.014** | 1.78 | 0.115 |
| | MCTS vs Heur-GD | 2.16 | 0.073 | -0.21 | 0.843 | 2.51 | **0.046** |
| M=100 | MCTS vs Random | 2.18 | 0.082 | 2.44 | **0.043** | 2.45 | **0.040** |
| | MCTS vs ETS | -1.00 | 0.356 | 1.31 | 0.246 | -0.69 | 0.515 |
| | MCTS vs Heur-GD | 12.55 | **0.000** | 1.48 | 0.180 | 6.29 | **0.001** |
| M=200 | MCTS vs Random | 3.21 | **0.029** | 2.16 | 0.075 | 2.37 | **0.046** |
| | MCTS vs ETS | 0.39 | 0.707 | 0.65 | 0.548 | -0.51 | 0.639 |
| | MCTS vs Heur-GD | 25.79 | **0.000** | 9.69 | **0.001** | 6.41 | **0.000** |

Table 23: Performance comparison in the Class Incremental Learning setting over various memory sizes for the methods on the 20-task datasets.

| Memory Size | Method | Split CIFAR-100 | | Split miniImagenet | |
|---|---|---|---|---|---|
| | | ACC (%) | BWT (%) | ACC (%) | BWT (%) |
| M=100 | Random | 4.49 ± 0.04 | -85.73 ± 0.34 | 3.36 ± 0.23 | -60.11 ± 0.83 |
| | ETS | 4.51 ± 0.13 | -85.62 ± 0.34 | 3.34 ± 0.21 | -60.25 ± 0.47 |
| | Heur-GD | 4.55 ± 0.13 | -84.62 ± 0.44 | 3.29 ± 0.14 | -57.01 ± 0.45 |
| | MCTS | 4.62 ± 0.10 | -84.73 ± 0.23 | 3.51 ± 0.19 | -57.57 ± 1.13 |
| M=200 | Random | 5.24 ± 0.12 | -84.53 ± 0.28 | 3.60 ± 0.24 | -58.83 ± 0.81 |
| | ETS | 4.98 ± 0.14 | -85.05 ± 0.19 | 3.75 ± 0.14 | -58.92 ± 1.03 |
| | Heur-GD | 5.20 ± 0.16 | -83.33 ± 0.29 | 3.95 ± 0.36 | -54.81 ± 0.58 |
| | MCTS | 5.84 ± 0.16 | -82.89 ± 0.28 | 4.26 ± 0.19 | -55.19 ± 0.73 |
| M=400 | Random | 7.13 ± 0.25 | -81.93 ± 0.29 | 4.81 ± 0.40 | -56.36 ± 0.44 |
| | ETS | 6.35 ± 0.25 | -82.92 ± 0.23 | 4.35 ± 0.15 | -57.28 ± 0.83 |
| | Heur-GD | 7.66 ± 0.60 | -79.82 ± 0.60 | 6.97 ± 0.78 | -49.44 ± 1.04 |
| | MCTS | 8.31 ± 0.21 | -79.58 ± 0.51 | 7.64 ± 0.70 | -48.73 ± 1.71 |
| M=800 | Random | 10.64 ± 0.51 | -77.27 ± 0.62 | 8.40 ± 1.07 | -50.95 ± 1.85 |
| | ETS | 9.13 ± 0.20 | -79.54 ± 0.43 | 7.15 ± 0.51 | -53.95 ± 0.74 |
| | Heur-GD | 11.33 ± 0.47 | -74.75 ± 0.73 | 8.44 ± 1.15 | -48.89 ± 1.43 |
| | MCTS | 12.06 ± 0.24 | -74.70 ± 0.29 | 11.56 ± 1.30 | -44.30 ± 1.63 |
| M=1600 | Random | 15.54 ± 0.69 | -70.42 ± 0.55 | 13.72 ± 1.30 | -44.19 ± 1.73 |
| | ETS | 13.99 ± 0.32 | -72.42 ± 0.24 | 12.17 ± 1.40 | -47.55 ± 1.39 |
| | Heur-GD | 15.66 ± 1.84 | -67.24 ± 1.75 | 8.39 ± 1.35 | -49.35 ± 1.77 |
| | MCTS | 17.59 ± 0.42 | -66.59 ± 0.66 | 15.40 ± 0.34 | -42.27 ± 0.73 |

Table 24: Two-tailed Welch's $t$-test results for the varying memory size experiments presented in Table 23 in the Class Incremental Learning setting on the 20-task datasets. We bold the $p$-values where $p < 0.05$ to indicate for when a method is statistically significantly better than its comparison.

| Memory size | Methods | Split CIFAR-100 | | Split miniImagenet | |
|---|---|---|---|---|---|
| | | $t$ | $p$ | $t$ | $p$ |
| M=100 | MCTS vs Random | 2.35 | 0.065 | 1.03 | 0.335 |
| | MCTS vs ETS | 1.38 | 0.208 | 1.20 | 0.266 |
| | MCTS vs Heur-GD | 0.87 | 0.410 | 1.86 | 0.103 |
| M=200 | MCTS vs Random | 5.95 | **0.000** | 4.34 | **0.003** |
| | MCTS vs ETS | 8.07 | **0.000** | 4.35 | **0.003** |
| | MCTS vs Heur-GD | 5.72 | **0.000** | 1.51 | 0.182 |
| M=400 | MCTS vs Random | 7.15 | **0.000** | 7.06 | **0.000** |
| | MCTS vs ETS | 11.98 | **0.000** | 9.22 | **0.000** |
| | MCTS vs Heur-GD | 2.01 | 0.101 | 1.29 | 0.232 |
| M=800 | MCTS vs Random | 5.05 | **0.003** | 3.75 | **0.006** |
| | MCTS vs ETS | 18.79 | **0.000** | 6.32 | **0.001** |
| | MCTS vs Heur-GD | 2.77 | **0.033** | 3.60 | **0.007** |
| M=1600 | MCTS vs Random | 5.08 | **0.002** | 2.50 | 0.059 |
| | MCTS vs ETS | 13.60 | **0.000** | 4.49 | **0.008** |
| | MCTS vs Heur-GD | 2.04 | 0.104 | 10.04 | **0.000** |

Table 25: Performance comparison when applying found MCTS schedules to new seeds with different replay samples from the selected tasks by the schedule. MCTS Seed X refers as the result from applying the replay schedule found by MCTS from seed X (1-5) to the new seeds (10-14). MCTS and Heur-GD outperform the Random and ETS scheduling which demonstrates the importance of scheduling which tasks to replay.

| Method | Split MNIST | | Split FashionMNIST | | Split notMNIST | |
|---|---|---|---|---|---|---|
| | ACC (%) | BWT (%) | ACC (%) | BWT (%) | ACC (%) | BWT (%) |
| Random | 96.04 ± 1.23 | -4.67 ± 1.51 | 92.43 ± 5.93 | -8.58 ± 7.40 | 91.68 ± 4.32 | -5.25 ± 5.28 |
| ETS | 96.35 ± 0.62 | -4.30 ± 0.79 | 94.96 ± 3.27 | -5.40 ± 4.07 | 91.16 ± 3.58 | -6.13 ± 4.50 |
| Heur-GD | 97.90 ± 0.47 | -2.31 ± 0.59 | 97.04 ± 0.79 | -2.86 ± 1.01 | 91.56 ± 2.34 | -5.57 ± 3.13 |
| MCTS Seed 1 | 97.71 ± 1.21 | -2.57 ± 1.51 | 96.20 ± 1.30 | -3.96 ± 1.63 | 90.79 ± 3.13 | -7.20 ± 3.64 |
| MCTS Seed 2 | 97.80 ± 0.67 | -2.42 ± 0.82 | 97.30 ± 1.11 | -2.61 ± 1.41 | 92.15 ± 0.54 | -4.66 ± 1.09 |
| MCTS Seed 3 | 96.83 ± 0.58 | -3.61 ± 0.74 | 95.32 ± 2.96 | -4.94 ± 3.74 | 94.34 ± 1.27 | -2.71 ± 1.03 |
| MCTS Seed 4 | 96.12 ± 2.00 | -4.54 ± 2.49 | 94.56 ± 2.55 | -5.99 ± 3.31 | 89.47 ± 2.92 | -7.43 ± 3.27 |
| MCTS Seed 5 | 97.07 ± 0.96 | -3.35 ± 1.19 | 96.80 ± 0.64 | -3.11 ± 0.79 | 93.08 ± 0.84 | -3.64 ± 0.84 |
| Method | Permuted MNIST | | Split CIFAR-100 | | Split miniImagenet | |
| | ACC (%) | BWT (%) | ACC (%) | BWT (%) | ACC (%) | BWT (%) |
| Random | 72.01 ± 1.71 | -26.43 ± 1.94 | 51.29 ± 0.61 | -38.01 ± 0.61 | 49.12 ± 0.57 | -14.91 ± 0.59 |
| ETS | 70.42 ± 1.16 | -28.20 ± 1.19 | 46.69 ± 1.51 | -42.83 ± 1.66 | 46.00 ± 1.17 | -17.94 ± 1.16 |
| Heur-GD | 76.19 ± 2.81 | -21.48 ± 3.12 | 57.00 ± 2.01 | -31.25 ± 2.08 | 49.75 ± 1.90 | -11.96 ± 1.71 |
| MCTS Seed 1 | 74.59 ± 1.76 | -23.29 ± 1.98 | 54.58 ± 1.18 | -33.65 ± 1.54 | 46.60 ± 0.64 | -16.15 ± 1.12 |
| MCTS Seed 2 | 74.82 ± 1.68 | -23.00 ± 1.78 | 53.86 ± 1.11 | -34.48 ± 1.13 | 50.10 ± 0.86 | -11.89 ± 1.01 |
| MCTS Seed 3 | 72.68 ± 1.97 | -25.34 ± 2.11 | 53.21 ± 1.02 | -35.09 ± 0.81 | 47.01 ± 0.69 | -16.54 ± 0.75 |
| MCTS Seed 4 | 74.08 ± 0.83 | -23.86 ± 0.85 | 54.06 ± 1.69 | -34.52 ± 1.62 | 48.47 ± 1.17 | -14.52 ± 1.46 |
| MCTS Seed 5 | 75.05 ± 1.36 | -22.77 ± 1.55 | 55.23 ± 2.05 | -33.12 ± 2.03 | 50.30 ± 0.38 | -12.38 ± 0.55 |

## C.8 Results with MCTS Schedules Applied to New Seeds

In this section, we further show the benefits of replay scheduling by applying schedules found by MCTS on CL scenarios with different replay samples from the selected tasks by the schedule. We select 5 new seeds (seeds 10-14) and apply the replay schedules found by MCTS from independent runs (seeds 1-5) separately on the new seeds. We run similar experiments in the Task-IL setting with same setup as in Table 1 (Section 4.1) with uniform memory selection, and compare against the scheduling baselines Random, ETS, and Heur-GD executed on the same set of new seeds.

Table 25 shows the mean and std. of the evaluated ACC and BWT metrics. MCTS Seed X refers to applying the replay schedule found by MCTS from seed X to the CL experiments with the 5 new seeds. We observe that the found MCTS schedules mostly outperform Random and ETS on all datasets, especially on Split CIFAR-100 and Permuted MNIST. Heur-GD is a competitive baseline against the MCTS schedules, especially on Split CIFAR-100. However, Heur-GD requires tuning a validation accuracy threshold for every dataset which can be difficult to tune for every new memory size and memory selection method. These results further demonstrates the importance of scheduling which tasks to replay in CL to more effectively reduce catastrophic forgetting.

## C.9 Results with Equal Task Scheduling Combined with Heuristic

In this section, we present results with an additional baselines that combines Equal Task Scheduling (ETS) with the heuristic Heur-GD. This scheduling baseline replays the old tasks equally at every task, and also checks whether their validation accuracy is below a threshold according to the heuristic rule to replay harder tasks more. We compare ETS+Heur-GD against MCTS in the Task-IL setting with same setup as in Table 1 (Section 4.1). We provide the ranges for $\tau$ that was used on each dataset and put the best value in **bold**:

- Split MNIST: $\tau = \{0.9, 0.93, 0.95, 0.96, 0.97, 0.98, \mathbf{0.99}\}$
- Split FashionMNIST: $\tau = \{0.9, 0.93, 0.95, 0.96, 0.97, 0.98, \mathbf{0.99}\}$
- Split notMNIST: $\tau = \{0.9, \mathbf{0.93}, 0.95, 0.96, 0.97, 0.98, 0.99\}$
- Permuted MNIST: $\tau = \{0.5, 0.6, \mathbf{0.7}, 0.75, 0.8, 0.9, 0.99\}$
- Split CIFAR-100: $\tau = \{0.4, 0.45, 0.5, 0.55, \mathbf{0.6}, 0.65, 0.7\}$
- Split miniImagenet: $\tau = \{0.5, 0.6, \mathbf{0.7}, 0.75, 0.8, 0.9, 0.99\}$

Table 26: Performance comparison between MCTS (Ours) and the baselines, including ETS combined with Heur-GD (ETS+Heur-GD). We use uniform sampling for memory selection, and memory sizes are set to $M = 10$ and $M = 100$ for 5-task and 10/20-task datasets respectively. MCTS performs better than ETS+Heur-GD on most datasets, except on Split miniImagenet where MCTS performs on par.

| Method | Split MNIST ACC (%) | Split MNIST BWT (%) | Split FashionMNIST ACC (%) | Split FashionMNIST BWT (%) | Split notMNIST ACC (%) | Split notMNIST BWT (%) |
|---|---|---|---|---|---|---|
| Random | 94.91 ± 2.52 | -6.13 ± 3.16 | 95.89 ± 2.03 | -4.33 ± 2.55 | 91.84 ± 1.48 | -5.37 ± 2.12 |
| ETS | 94.02 ± 4.25 | -7.22 ± 5.33 | 95.81 ± 3.53 | -4.45 ± 4.34 | 91.01 ± 1.39 | -6.16 ± 1.82 |
| Heur-GD | 96.02 ± 2.32 | -4.64 ± 2.90 | 97.09 ± 0.62 | -2.82 ± 0.84 | 91.26 ± 3.99 | -6.06 ± 4.70 |
| ETS+Heur-GD | 95.19 ± 2.10 | -5.70 ± 2.64 | 96.30 ± 0.75 | -3.81 ± 0.93 | 93.16 ± 0.97 | -3.49 ± 1.48 |
| MCTS | 97.93 ± 0.56 | -2.27 ± 0.71 | 98.27 ± 0.17 | -1.29 ± 0.20 | 94.64 ± 0.39 | -1.47 ± 0.79 |

| Method | PermutedMNIST ACC (%) | PermutedMNIST BWT (%) | Split CIFAR-100 ACC (%) | Split CIFAR-100 BWT (%) | Split miniImagenet ACC (%) | Split miniImagenet BWT (%) |
|---|---|---|---|---|---|---|
| Random | 72.59 ± 1.52 | -25.71 ± 1.76 | 53.76 ± 1.80 | -35.11 ± 1.93 | 49.89 ± 1.03 | -14.79 ± 1.14 |
| ETS | 71.09 ± 2.31 | -27.39 ± 2.59 | 47.70 ± 2.16 | -41.69 ± 2.37 | 46.97 ± 1.24 | -18.32 ± 1.34 |
| Heur-GD | 76.68 ± 2.13 | -20.82 ± 2.41 | 57.31 ± 1.21 | -30.76 ± 1.45 | 49.66 ± 1.10 | -12.04 ± 0.59 |
| ETS+Heur-GD | 75.89 ± 1.69 | -21.70 ± 1.83 | 51.00 ± 0.76 | -37.33 ± 0.58 | 50.91 ± 0.62 | -12.36 ± 1.40 |
| MCTS | 76.34 ± 0.98 | -21.21 ± 1.16 | 56.60 ± 1.13 | -31.39 ± 1.11 | 50.20 ± 0.72 | -13.46 ± 1.22 |

In Table 26, we report the ACC and BWT metrics on all datasets. While ETS+Heur-GD outperforms ETS on all datasets, it is outperformed by MCTS on every dataset except Split miniImagenet. These results show the benefits of using replay scheduling policies that selectively can turn on and off replaying old tasks to focus more on remembering hard tasks. Furthermore, Heur-GD outperforms ETS+Heur-GD on most datasets, which further indicates the difficulty of coming up with well-designed heuristic scheduling policies.

# D  Additional Experimental Settings and Results for Replay Scheduling Policy Experiments

This section is structured as follows:

- Appendix D.1: Full details on the experimental settings.

- Appendix D.2: Details of the ranking method to assess the generalization abilities of the scheduling policies.

- Appendix D.3: Additional experimental results for the Replay Scheduling Policy Generalization experiments with Welch's t-tests for statistical significance between the RL algorithms (DQN and A2C) and the baselines.

- Appendix D.4: Task splits in the continual learning environments used for testing.

## D.1  Experimental Settings for RL-Based Framework

Here, we provide details on the experimental settings for the experiments with our RL-based framework where we use multiple CL environments for learning replay scheduling policies that generalize.

**Datasets.**  We conduct experiments on CL environments with four datasets common CL benchmarks, namely, Split MNIST (Zenke et al., 2017), Split Fashion-MNIST (Xiao et al., 2017), Split notMNIST (Bulatov, 2011), and Split CIFAR-10 (Krizhevsky & Hinton, 2009).  All datasets consists of 5 tasks with 2 classes/task.

**CL Network Architectures.**  We use a 2-layer MLP with 256 hidden units and ReLU activation for Split MNIST, Split FashionMNIST, and Split notMNIST. For Split CIFAR-10, we use the same ConvNet architecture as used for Split CIFAR-100 in Appendix C.1. We use a multi-head output layer for each dataset and assume task labels are available at test time for selecting the correct output head related to the task.

**CL Hyperparameters.**  We train all networks with the Adam optimizer (Kingma & Ba, 2015) with learning rate $\eta = 0.001$ and hyperparameters $\beta_1 = 0.9$ and $\beta_2 = 0.999$. Note that the learning rate for Adam is not reset before training on a new task. Next, we give details on number of training epochs and batch sizes specific for each dataset:

- Split MNIST: 10 epochs/task, batch size 128.
- Split FashionMNIST: 10 epochs/task, batch size 128.
- Split notMNIST: 20 epochs/task, batch size 128.
- Split CIFAR-10: 20 epochs/task, batch size 256.

**Generating CL Environments.**  We generate multiple CL environments with pre-set random seeds for initializing the network parameters $\phi$ and shuffling the task order. The pre-set random seeds are in the range $0 - 49$, such that we have 50 environments for each dataset. We shuffle the task order by permuting the class order and then split the classes into 5 pairs (tasks) with 2 classes/pair. For environments with seed 0, we keep the original task order in the dataset. Taking a step at task $t$ in the CL environments involves training the CL network on the $t$-th dataset with a replay memory $\mathcal{M}_t$ from the discrete action space described in Section 3.2. Therefore, to speed up the experiments with the RL algorithms, we run a breadth-first search (BFS) through the discrete action space and save the classification results for re-use during policy learning. Note that the action space has 1050 possible paths of replay schedules for the datasets with $T = 5$ tasks, which makes the environment generation time-consuming. Hence, we only generate environments where the replay memory size $M = 10$ have been used, and leave analysis of different memory sizes as future work. The CL environments we used are provided in the code submission.

**DQN and A2C Architectures.**  The input layer has size $T - 1$ where each unit is inputting the task performances since the states are represented by the validation accuracies $s_t = [A_{t,1}^{(val)}, ..., A_{t,t}^{(val)}, 0, ..., 0]$. The current task can therefore be determined by the number of non-zero state inputs. The output layer has 35 units representing the possible actions at $T = 5$ with the discrete action space we have constructed

Table 27: DQN hyperparameters for the experiments on **New Task Orders** in Section 4.2.

| Hyperparameters | Split MNIST | Split FashionMNIST | Split notMNIST | Split CIFAR-10 |
|---|---|---|---|---|
| Training Environments | 30 | 20 | 30 | 10 |
| Learning Rate | 0.0001 | 0.0003 | 0.0001 | 0.0003 |
| Optimizer | Adam | Adam | Adam | Adam |
| Buffer Size | 10k | 10k | 10k | 10k |
| Target Update per step | 500 | 500 | 500 | 500 |
| Batch Size | 32 | 32 | 32 | 32 |
| Discount Factor $\gamma$ | 1.0 | 1.0 | 1.0 | 1.0 |
| Exploration Start $\epsilon_{start}$ | 1.0 | 1.0 | 1.0 | 1.0 |
| Exploration Final $\epsilon_{final}$ | 0.02 | 0.02 | 0.02 | 0.02 |
| Exploration Annealing (episodes) | 2.5k | 2.5k | 2.5k | 2.5k |
| Training Episodes | 10k | 10k | 10k | 10k |

Table 28: A2C hyperparameters for the experiments on **New Task Orders** in Section 4.2.

| Hyperparameters | Split MNIST | Split FashionMNIST | Split notMNIST | Split CIFAR-10 |
|---|---|---|---|---|
| Training Environments | 10 | 40 | 10 | 10 |
| Learning Rate | 0.0001 | 0.0001 | 0.0001 | 0.0003 |
| Optimizer | RMSProp | RMSProp | RMSProp | RMSProp |
| Gradient Clipping | 0.5 | 0.5 | 0.5 | 0.5 |
| GAE parameter $\lambda$ | 0.95 | 0.95 | 0.95 | 0.95 |
| VF coefficient | 0.5 | 0.5 | 0.5 | 0.5 |
| Entropy coefficient | 0.01 | 0.005 | 0.01 | 0.01 |
| Number of steps $n_{steps}$ | 5 | 4 | 5 | 5 |
| Discount Factor $\gamma$ | 1.0 | 1.0 | 1.0 | 1.0 |
| Training Episodes | 100k | 100k | 100k | 100k |

in Section 3.2. We use action masking on the output units to prevent the network from selection invalid actions for constructing the replay memory at the current task. The DQN is a 2-layer MLP with 512 hidden units and ReLU activations. For A2C, we use separate networks for parameterizing the policy and the value function, where both networks are 2-layer MLPs with 64 hidden units of Tanh activations.

**DQN and A2C Hyperparameters.** We provide the hyperparameters for the both DQN and A2C in Table 27-31. Table 27 and 28 includes the hyperparameters on the New Task Order experiment for DQN and A2C respectively, while Table 30 and 31 includes the hyperparameters on the New Dataset experiment for DQN and A2C respectively. Regarding the training environments in Table 30 and 31, we use two different datasets in the training environments to increase the diversity. When Spit notMNIST is for testing, half the amount of training environments are using Split MNIST and the other half uses Split FashionMNIST. For example, in Table 31, A2C uses 10 training environments which means that there are 5 Split MNIST environments and 5 Split FashionMNIST environments. Similarly, half the amount of training environments are using Split MNIST and the other half uses Split notMNIST when the testing environments uses Split FashionMNIST.

**Computational Cost.** All experiments were performed on one NVIDIA GeForce RTW 2080Ti on an internal GPU cluster. Generating a CL environment for one seed with Split MNIST took on around 9.5 hours averaged over 10 runs of BFS. Similarly for Split CIFAR-10, generating one CL environment took on average 16.1 hours. Table 29 shows a time-cost ablation experiment w/ or w/o a DQN for selecting which tasks to replay in Split MNIST. We measured the wall clock time for training and evaluating the CL model on the 5 Split MNIST tasks w/ and w/o the DQN, and show the wall clock time averaged over 10 different DQN seeds. The time difference when w/ DQN is only 3.2 seconds, since selecting which tasks to replay is only a forward pass with the RL policy.

Table 29: Time-cost ablation experiment w/ or w/o a DQN for replay scheduling on Split MNIST.

| Time Cost | With DQN | Without DQN | Difference |
|---|---|---|---|
| Avg. Time (in sec) | 84.6 | 81.4 | 3.2 |

**Implementations.** The code for DQN was adapted from OpenAI baselines (Dhariwal et al., 2017) and the PyTorch (Paszke et al., 2019) tutorial on DQN `https://pytorch.org/tutorials/intermediate/reinforcement_q_learning.html`. For A2C, we followed the implementations released by Kostrikov (2018) and Igl et al. (2021).

Table 30: DQN hyperparameters for the experiments on **New Dataset** in Section 4.2. Split notMNIST and Split FashionMNIST indicate the dataset used in the test environments.

| Hyperparameters | S-notMNIST | S-FashionMNIST |
|---|---|---|
| Training Environments | 30 | 20 |
| Learning Rate | 0.0001 | 0.0001 |
| Optimizer | Adam | Adam |
| Buffer Size | 10k | 100k |
| Target Update per step | 500 | 500 |
| Batch Size | 32 | 32 |
| Discount Factor $\gamma$ | 1.0 | 1.0 |
| Exploration Start $\epsilon_{start}$ | 1.0 | 1.0 |
| Exploration Final $\epsilon_{final}$ | 0.02 | 0.2 |
| Exploration Annealing (episodes) | 2.5k | 2.5k |
| Training Episodes | 10k | 10k |

Table 31: A2C hyperparameters for the experiments on **New Dataset** in Section 4.2. Split notMNIST and Split FashionMNIST indicate the dataset used in the test environments.

| Hyperparameters | S-notMNIST | S-FashionMNIST |
|---|---|---|
| Training Environments | 10 | 30 |
| Learning Rate | 0.0001 | 0.0003 |
| Optimizer | RMSProp | RMSProp |
| Gradient Clipping | 0.5 | 0.5 |
| GAE parameter $\lambda$ | 0.95 | 0.95 |
| VF coefficient | 0.5 | 0.5 |
| Entropy coefficient | 0.01 | 0.005 |
| Number of steps $n_{steps}$ | 5 | 4 |
| Discount Factor $\gamma$ | 1.0 | 1.0 |
| Training Episodes | 100k | 100k |

### D.2 Ranking Method for Assessing Generalization

We use a ranking method based on the CL performance in every test environment for performance comparison between the methods in Section 4.2. We use rankings because the performances can vary greatly between environments with different task orders and datasets. To measure the CL performance in the environments, we use the average test accuracy over all tasks after learning the final task, i.e.,

$$\text{ACC} = \frac{1}{T} \sum_{i=1}^{T} A_{T,i}^{(test)},$$

where $A_{t,i}^{(test)}$ is the test accuracy of task $i$ after learning task $t$. Each method are ranked in descending order based on the ACC achieved in an environment. For example, assume that we want to compare the CL performance from using learned replay scheduling policies with DQN and A2C against a Random scheduling policy in one environment. The CL performances achieved for each method are given by

$$[\text{ACC}_{\text{Random}}, \text{ACC}_{\text{DQN}}, \text{ACC}_{\text{A2C}}] = [90\%, 99\%, 95\%].$$

We get the following ranking order between the methods based on their corresponding ACC:

$$\texttt{ranking}([\text{ACC}_{\text{Random}}, \text{ACC}_{\text{DQN}}, \text{ACC}_{\text{A2C}}]) = [3, 1, 2],$$

where DQN is ranked in 1st place, A2C in 2nd, and Random in 3rd. When there are multiple environments for evaluation, we compute the average ranking across the ranking positions in every environment for each method to compare.

The average ranking for DQN and A2C are computed over the seed for initializing the network parameters as well as the seed of the environment. Similarly, the Random baseline is affected by the seed setting the random selection of actions and the environment seed. However, the performance of the ETS and Heuristic baselines are affected by the seed of the environment as these policies are fixed. We use copied values of the performance in environments for the ETS and Heuristic baselines when we need to compare across different random seeds for Random, DQN, and A2C. We show an example of such ranking calculation for ETS, a Heuristic baseline, DQN, and A2C. Consider the following performances for one environment:

$$\begin{bmatrix} \text{ACC}_{\text{ETS}}^1 & \text{ACC}_{\text{Heur}}^1 & \text{ACC}_{\text{DQN}}^1 & \text{ACC}_{\text{A2C}}^1 \\ \text{ACC}_{\text{ETS}}^2 & \text{ACC}_{\text{Heur}}^2 & \text{ACC}_{\text{DQN}}^2 & \text{ACC}_{\text{A2C}}^2 \end{bmatrix} = \begin{bmatrix} 90\% & 95\% & 95\% & 99\% \\ * & * & 97\% & 98\% \end{bmatrix},$$

where $*$ denotes a copy of the ACC value in the first row. The subscript on ACC denotes the method and the superscript the seed used for initializing the policy network $\boldsymbol{\theta}$. Therefore, we copy the values for ETS and Heur such that the $\text{ACC}_{\text{DQN}}^2$ for seed 2 can be compared against ETS and Heur. Note that there is a tie between $\text{ACC}_{\text{Heur}}^1$ and $\text{ACC}_{\text{DQN}}^1$ as they have ACC 95%. We handle ties by assigning tied methods the average of their ranks, such that the ranks for both seeds will be

$$
\begin{aligned}
&\texttt{ranking}\left(\begin{bmatrix} \text{ACC}_{\text{ETS}}^1 & \text{ACC}_{\text{Heur}}^1 & \text{ACC}_{\text{DQN}}^1 & \text{ACC}_{\text{A2C}}^1 \\ \text{ACC}_{\text{ETS}}^2 & \text{ACC}_{\text{Heur}}^2 & \text{ACC}_{\text{DQN}}^2 & \text{ACC}_{\text{A2C}}^2 \end{bmatrix}, \texttt{axis=-1, keepdim=True}\right) \\
&=\texttt{ranking}\left(\begin{bmatrix} 90\% & 95\% & 95\% & 99\% \\ 90\% & 95\% & 97\% & 98\% \end{bmatrix}, \texttt{axis=-1, keepdim=True}\right) \\
&=\begin{bmatrix} 4 & 2.5 & 2.5 & 1 \\ 4 & 3 & 2 & 1 \end{bmatrix},
\end{aligned}
$$

where we inserted the copied values, such that $\text{ACC}_{\text{ETS}}^1 = \text{ACC}_{\text{ETS}}^2 = 90\%$ and $\text{ACC}_{\text{Heur}}^1 = \text{ACC}_{\text{Heur}}^2 = 95\%$. The mean ranking across the seeds thus becomes

$$
\texttt{mean}\left(\begin{bmatrix} 4 & 2.5 & 2.5 & 1 \\ 4 & 3 & 2 & 1 \end{bmatrix}, \texttt{axis=0}\right) = \begin{bmatrix} 4 & 2.75 & 2.25 & 1 \end{bmatrix}
$$

where A2C comes in 1st place, DQN in 2nd, Heur. in 3rd, and ETS on 4th place. We average across seeds and environments to obtain the final ranking score for each method for comparison.

### D.3 Additional Results for Replay Scheduling Policy Generalization Experiments

Here, we display the ACC and BWT metrics for each method averaged across 5 seeds and the average rank in every test environment. Note that the ACC and BWT from ETS and the heuristic scheduling baselines have standard deviation zero since these policies are fixed. Averaging the ranks over all test environments yields the corresponding average rank in Table 4 and 5. Furthermore, we provide the p-values from Welch's t-test to show whether the statistical significance of the results as recommended by Colas et al. (2019).

- **New Task Order, Split MNIST:** Metrics in Table 32, and Welch's t-test in Table 33.

- **New Task Order, Split FashionMNIST:** Metrics in Table 35, and Welch's t-test in Table 34.

- **New Task Order, Split notMNIST:** Metrics in Table 36, and Welch's t-test in Table 37.

- **New Task Order, Split CIFAR-10:** Metrics in Table 39, and Welch's t-test in Table 38.

- **New Dataset, Split notMNIST:** Metrics in Table 40, and Welch's t-test in Table 41.

- **New Dataset, Split FashionMNIST:** Metrics in Table 43, and Welch's t-test in Table 42.

In the metric tables, we indicate with red, orange, and yellow highlight the 1st, 2nd, and 3rd-best performing scheduling method based on its mean value for each metric. Note that ETS and the heuristics are deterministic scheduling methods in the experiments, which means that their ACC is the same across the varying seeds for Random, DQN, and A2C. We observe that the improvement with the learning replay scheduling policies is less significant than in the MCTS experiments, however, such behaviour is common when RL is used for generalizing to new environments.

Table 32: Performance comparison in every test environment with seed (10-19) with with **Split MNIST** for **New Task Order** experiment. Under each column named 'Test Env. Seed X', we show the mean and stddev. of ACC and BWT, and the Rank averaged over the RL seeds for the corresponding method.

| | **Test Env. Seed 10** | | | **Test Env. Seed 11** | | |
|---|---|---|---|---|---|---|
| **Method** | ACC (%) | BWT (%) | Rank | ACC (%) | BWT (%) | Rank |
| Random | 93.95 ± 2.68 | -6.95 ± 3.33 | 4.2 | 92.13 ± 1.34 | -9.46 ± 1.68 | 4.6 |
| ETS | 89.89 ± 0.00 | -12.01 ± 0.00 | 6.8 | 93.77 ± 0.00 | -7.41 ± 0.00 | 2.8 |
| Heur-GD | 94.64 ± 0.00 | -6.06 ± 0.00 | 4.5 | 91.34 ± 0.00 | -10.47 ± 0.00 | 5.6 |
| Heur-LD | 95.63 ± 0.00 | -4.80 ± 0.00 | 1.8 | 91.34 ± 0.00 | -10.47 ± 0.00 | 5.6 |
| Heur-AT | 94.64 ± 0.00 | -6.06 ± 0.00 | 4.5 | 91.34 ± 0.00 | -10.47 ± 0.00 | 5.6 |
| DQN (Ours) | 94.68 ± 1.16 | -6.00 ± 1.44 | 3.2 | 94.08 ± 1.75 | -7.01 ± 2.19 | 2.8 |
| A2C (Ours) | 95.31 ± 0.00 | -5.21 ± 0.00 | 3.0 | 96.41 ± 0.18 | -4.09 ± 0.23 | 1.0 |

| | **Test Env. Seed 12** | | | **Test Env. Seed 13** | | |
|---|---|---|---|---|---|---|
| **Method** | ACC (%) | BWT (%) | Rank | ACC (%) | BWT (%) | Rank |
| Random | 92.22 ± 2.35 | -9.16 ± 2.93 | 4.4 | 93.65 ± 2.06 | -7.47 ± 2.58 | 4.8 |
| ETS | 91.69 ± 0.00 | -9.82 ± 0.00 | 6.4 | 94.95 ± 0.00 | -5.88 ± 0.00 | 3.0 |
| Heur-GD | 94.82 ± 0.00 | -5.89 ± 0.00 | 1.7 | 94.66 ± 0.00 | -6.14 ± 0.00 | 4.7 |
| Heur-LD | 91.93 ± 0.00 | -9.50 ± 0.00 | 5.4 | 93.17 ± 0.00 | -8.00 ± 0.00 | 6.8 |
| Heur-AT | 94.82 ± 0.00 | -5.89 ± 0.00 | 1.7 | 94.66 ± 0.00 | -6.14 ± 0.00 | 4.7 |
| DQN (Ours) | 93.05 ± 1.37 | -8.05 ± 1.71 | 4.6 | 95.59 ± 1.22 | -4.94 ± 1.53 | 2.0 |
| A2C (Ours) | 93.62 ± 0.00 | -7.34 ± 0.00 | 3.8 | 95.56 ± 0.00 | -5.01 ± 0.00 | 2.0 |

| | **Test Env. Seed 14** | | | **Test Env. Seed 15** | | |
|---|---|---|---|---|---|---|
| **Method** | ACC (%) | BWT (%) | Rank | ACC (%) | BWT (%) | Rank |
| Random | 85.74 ± 3.47 | -17.19 ± 4.32 | 3.8 | 94.54 ± 1.49 | -6.20 ± 1.87 | 4.8 |
| ETS | 87.29 ± 0.00 | -15.23 ± 0.00 | 2.4 | 95.32 ± 0.00 | -5.23 ± 0.00 | 4.6 |
| Heur-GD | 81.20 ± 0.00 | -23.00 ± 0.00 | 6.3 | 95.92 ± 0.00 | -4.49 ± 0.00 | 2.7 |
| Heur-LD | 81.20 ± 0.00 | -23.00 ± 0.00 | 6.3 | 96.05 ± 0.00 | -4.30 ± 0.00 | 1.0 |
| Heur-AT | 82.36 ± 0.00 | -21.52 ± 0.00 | 4.8 | 95.92 ± 0.00 | -4.49 ± 0.00 | 2.7 |
| DQN (Ours) | 91.22 ± 3.23 | -10.45 ± 4.02 | 1.6 | 94.37 ± 0.74 | -6.45 ± 0.90 | 6.4 |
| A2C (Ours) | 88.16 ± 5.81 | -14.27 ± 7.25 | 2.8 | 94.82 ± 0.00 | -5.94 ± 0.00 | 5.8 |

| | **Test Env. Seed 16** | | | **Test Env. Seed 17** | | |
|---|---|---|---|---|---|---|
| **Method** | ACC (%) | BWT (%) | Rank | ACC (%) | BWT (%) | Rank |
| Random | 83.05 ± 3.07 | -20.78 ± 3.84 | 6.2 | 95.86 ± 0.48 | -4.52 ± 0.59 | 2.4 |
| ETS | 79.38 ± 0.00 | -25.36 ± 0.00 | 6.8 | 95.69 ± 0.00 | -4.77 ± 0.00 | 2.8 |
| Heur-GD | 91.16 ± 0.00 | -10.57 ± 0.00 | 3.2 | 93.48 ± 0.00 | -7.39 ± 0.00 | 5.8 |
| Heur-LD | 91.16 ± 0.00 | -10.57 ± 0.00 | 3.2 | 93.48 ± 0.00 | -7.39 ± 0.00 | 5.8 |
| Heur-AT | 91.16 ± 0.00 | -10.57 ± 0.00 | 3.2 | 93.48 ± 0.00 | -7.39 ± 0.00 | 5.8 |
| DQN (Ours) | 92.93 ± 1.19 | -8.41 ± 1.48 | 1.8 | 94.67 ± 2.13 | -5.91 ± 2.65 | 4.0 |
| A2C (Ours) | 91.11 ± 0.98 | -10.69 ± 1.23 | 3.6 | 96.28 ± 0.21 | -3.89 ± 0.27 | 1.4 |

| | **Test Env. Seed 18** | | | **Test Env. Seed 19** | | |
|---|---|---|---|---|---|---|
| **Method** | ACC (%) | BWT (%) | Rank | ACC (%) | BWT (%) | Rank |
| Random | 91.30 ± 2.91 | -10.28 ± 3.63 | 2.8 | 95.85 ± 2.20 | -4.71 ± 2.76 | 1.8 |
| ETS | 92.89 ± 0.00 | -8.28 ± 0.00 | 1.4 | 97.40 ± 0.00 | -2.78 ± 0.00 | 1.2 |
| Heur-GD | 87.82 ± 0.00 | -14.53 ± 0.00 | 5.8 | 88.34 ± 0.00 | -14.21 ± 0.00 | 5.0 |
| Heur-LD | 87.82 ± 0.00 | -14.53 ± 0.00 | 5.8 | 88.34 ± 0.00 | -14.21 ± 0.00 | 5.0 |
| Heur-AT | 87.82 ± 0.00 | -14.53 ± 0.00 | 5.8 | 88.34 ± 0.00 | -14.21 ± 0.00 | 5.0 |
| DQN (Ours) | 90.05 ± 0.88 | -11.74 ± 1.12 | 3.8 | 89.04 ± 1.82 | -13.28 ± 2.28 | 4.4 |
| A2C (Ours) | 91.64 ± 0.00 | -9.75 ± 0.00 | 2.6 | 87.64 ± 1.88 | -15.03 ± 2.33 | 5.6 |

Table 33: Two-tailed Welch's $t$-test results for **Split MNIST** in **New Task Order** experiment. We bold the $p$-values where $p < 0.05$ to indicate for when a method is statistically significantly better than its comparison.

| Methods | Test Env. Seed 10 $t$ | $p$ | Test Env. Seed 11 $t$ | $p$ | Test Env. Seed 12 $t$ | $p$ | Test Env. Seed 13 $t$ | $p$ | Test Env. Seed 14 $t$ | $p$ |
|---|---|---|---|---|---|---|---|---|---|---|
| DQN vs Random | 0.50 | 0.636 | 1.77 | 0.117 | 0.62 | 0.560 | 1.62 | 0.152 | 2.32 | 0.049 |
| DQN vs ETS | 8.30 | **0.001** | 0.36 | 0.740 | 1.99 | 0.117 | 1.05 | 0.352 | 2.44 | 0.071 |
| DQN vs Heur-GD | 0.08 | 0.942 | 3.13 | **0.035** | -2.57 | 0.062 | 1.53 | 0.201 | 6.21 | **0.003** |
| DQN vs Heur-LD: | -1.64 | 0.175 | 3.13 | **0.035** | 1.63 | 0.177 | 3.97 | **0.017** | 6.21 | **0.003** |
| DQN vs Heur-AT | 0.08 | 0.942 | 3.13 | **0.035** | -2.57 | 0.062 | 1.53 | 0.201 | 5.49 | **0.005** |
| DQN vs A2C | -1.08 | 0.341 | -2.65 | 0.056 | -0.83 | 0.455 | 0.06 | 0.957 | 0.92 | 0.391 |
| A2C vs Random | 1.01 | 0.369 | 6.32 | **0.003** | 1.20 | 0.298 | 1.85 | 0.138 | 0.72 | 0.499 |
| A2C vs ETS | inf | **0.000** | 28.80 | **0.000** | inf | **0.000** | inf | **0.000** | 0.30 | 0.778 |
| A2C vs Heur-GD | inf | **0.000** | 55.32 | **0.000** | -inf | **0.000** | inf | **0.000** | 2.40 | 0.075 |
| A2C vs Heur-LD | -inf | **0.000** | 55.32 | **0.000** | inf | **0.000** | inf | **0.000** | 2.40 | 0.075 |
| A2C vs Heur-AT | inf | **0.000** | 55.32 | **0.000** | -inf | **0.000** | inf | **0.000** | 2.00 | 0.117 |
| A2C vs DQN | 1.08 | 0.341 | 2.65 | 0.056 | 0.83 | 0.455 | -0.06 | 0.957 | -0.92 | 0.391 |

| Methods | Test Env. Seed 15 $t$ | $p$ | Test Env. Seed 16 $t$ | $p$ | Test Env. Seed 17 $t$ | $p$ | Test Env. Seed 18 $t$ | $p$ | Test Env. Seed 19 $t$ | $p$ |
|---|---|---|---|---|---|---|---|---|---|---|
| DQN vs Random | -0.21 | 0.844 | 6.01 | **0.002** | -1.09 | 0.332 | -0.82 | 0.452 | -4.76 | **0.002** |
| DQN vs ETS | -2.60 | 0.060 | 22.75 | **0.000** | -0.95 | 0.396 | -6.48 | **0.003** | -9.17 | **0.001** |
| DQN vs Heur-GD | -4.22 | **0.013** | 2.98 | **0.041** | 1.12 | 0.324 | 5.10 | **0.007** | 0.77 | 0.484 |
| DQN vs Heur-LD: | -4.57 | **0.010** | 2.98 | **0.041** | 1.12 | 0.324 | 5.10 | **0.007** | 0.77 | 0.484 |
| DQN vs Heur-AT | -4.22 | **0.013** | 2.98 | **0.041** | 1.12 | 0.324 | 5.10 | **0.007** | 0.77 | 0.484 |
| DQN vs A2C | -1.24 | 0.283 | 2.36 | **0.047** | -1.50 | 0.205 | -3.63 | **0.022** | 1.07 | 0.315 |
| A2C vs Random | 0.38 | 0.722 | 5.00 | 0.005 | 1.60 | 0.164 | 0.24 | 0.824 | -5.67 | **0.001** |
| A2C vs ETS | -inf | **0.000** | 23.87 | **0.000** | 5.56 | **0.005** | -inf | **0.000** | -10.39 | **0.000** |
| A2C vs Heur-GD | -inf | **0.000** | -0.10 | 0.925 | 26.08 | **0.000** | inf | **0.000** | -0.75 | 0.498 |
| A2C vs Heur-LD | -inf | **0.000** | -0.10 | 0.925 | 26.08 | **0.000** | inf | **0.000** | -0.75 | 0.498 |
| A2C vs Heur-AT | -inf | **0.000** | -0.10 | 0.925 | 26.08 | **0.000** | inf | **0.000** | -0.75 | 0.498 |
| A2C vs DQN | 1.24 | 0.283 | -2.36 | **0.047** | 1.50 | 0.205 | 3.63 | 0.022 | -1.07 | 0.315 |

Table 34: Two-tailed Welch's $t$-test results for **Split FashionMNIST** in **New Task Order** experiment. We bold the $p$-values where $p < 0.05$ to indicate for when a method is statistically significantly better than its comparison.

| Methods | Test Env. Seed 10 $t$ | $p$ | Test Env. Seed 11 $t$ | $p$ | Test Env. Seed 12 $t$ | $p$ | Test Env. Seed 13 $t$ | $p$ | Test Env. Seed 14 $t$ | $p$ |
|---|---|---|---|---|---|---|---|---|---|---|
| DQN vs Random | -0.05 | 0.962 | -0.17 | 0.871 | 0.18 | 0.864 | 3.28 | **0.024** | 0.24 | 0.819 |
| DQN vs ETS | 1.78 | 0.150 | 0.76 | 0.490 | 2.06 | 0.109 | 12.55 | **0.000** | 4.19 | **0.014** |
| DQN vs Heur-GD | -3.61 | **0.023** | -1.29 | 0.265 | -0.33 | 0.761 | 4.03 | **0.016** | -0.85 | 0.445 |
| DQN vs Heur-LD | -3.61 | **0.023** | -1.99 | 0.118 | -1.07 | 0.343 | 1.51 | 0.205 | -0.56 | 0.604 |
| DQN vs Heur-AT | 13.80 | **0.000** | -1.45 | 0.221 | -0.88 | 0.427 | -2.62 | 0.059 | 11.80 | **0.000** |
| DQN vs A2C | 22.29 | **0.000** | -3.89 | **0.018** | 0.48 | 0.659 | 1.76 | 0.154 | 0.51 | 0.635 |
| A2C vs Random | -5.14 | **0.007** | 9.64 | **0.001** | -0.11 | 0.919 | 2.90 | **0.044** | -0.36 | 0.740 |
| A2C vs ETS | -inf | **0.000** | inf | **0.000** | inf | **0.000** | inf | **0.000** | inf | **0.000** |
| A2C vs Heur-GD | -inf | **0.000** | inf | **0.000** | -inf | **0.000** | inf | **0.000** | -inf | **0.000** |
| A2C vs Heur-LD | -inf | **0.000** | inf | **0.000** | -inf | **0.000** | -inf | **0.000** | -inf | **0.000** |
| A2C vs Heur-AT | -inf | **0.000** | inf | **0.000** | -inf | **0.000** | -inf | **0.000** | inf | **0.000** |
| A2C vs DQN | -22.29 | **0.000** | 3.89 | **0.018** | -0.48 | 0.659 | -1.76 | 0.154 | -0.51 | 0.635 |

| Methods | Test Env. Seed 15 $t$ | $p$ | Test Env. Seed 16 $t$ | $p$ | Test Env. Seed 17 $t$ | $p$ | Test Env. Seed 18 $t$ | $p$ | Test Env. Seed 19 $t$ | $p$ |
|---|---|---|---|---|---|---|---|---|---|---|
| DQN vs Random | -7.48 | **0.000** | -1.05 | 0.327 | -1.11 | 0.300 | 3.23 | **0.028** | -2.26 | 0.067 |
| DQN vs ETS | -13.28 | **0.000** | -4.06 | **0.015** | 4.28 | **0.013** | 5.49 | **0.005** | -2.31 | 0.082 |
| DQN vs Heur-GD | 34.42 | **0.000** | 12.59 | **0.000** | -3.97 | **0.017** | 7.21 | **0.002** | -0.34 | 0.752 |
| DQN vs Heur-LD | -10.25 | **0.001** | 7.27 | **0.002** | -3.64 | **0.022** | 9.52 | **0.001** | 2.41 | 0.073 |
| DQN vs Heur-AT | 3.44 | **0.026** | 0.12 | 0.912 | 0.16 | 0.883 | 3.72 | **0.021** | -1.24 | 0.281 |
| DQN vs A2C | 1.29 | 0.268 | -4.15 | **0.014** | 10.37 | **0.000** | -1.01 | 0.370 | -4.23 | **0.013** |
| A2C vs Random | -9.64 | **0.001** | 4.21 | **0.013** | -14.00 | **0.000** | 3.52 | **0.024** | 3.78 | **0.019** |
| A2C vs ETS | -inf | **0.000** | 1.86 | 0.136 | -inf | **0.000** | inf | **0.000** | inf | **0.000** |
| A2C vs Heur-GD | inf | **0.000** | 356.86 | **0.000** | -inf | **0.000** | inf | **0.000** | inf | **0.000** |
| A2C vs Heur-LD | -inf | **0.000** | 243.42 | **0.000** | -inf | **0.000** | inf | **0.000** | inf | **0.000** |
| A2C vs Heur-AT | inf | **0.000** | 91.00 | **0.000** | -inf | **0.000** | inf | **0.000** | inf | **0.000** |
| A2C vs DQN | -1.29 | 0.268 | 4.15 | **0.014** | -10.37 | **0.000** | 1.01 | 0.370 | 4.23 | **0.013** |

Table 35: Performance comparison in every test environment with seed (10-19) with with **Split FashionM-NIST** for **New Task Order** experiment. Under each column named 'Test Env. Seed X', we show the mean and stddev. of ACC and BWT, and the Rank averaged over the RL seeds for the corresponding method.

| | Test Env. Seed 10 | | | Test Env. Seed 11 | | |
|---|---|---|---|---|---|---|
| **Method** | ACC (%) | BWT (%) | Rank | ACC (%) | BWT (%) | Rank |
| Random | 96.79 ± 3.02 | -3.11 ± 3.76 | 2.0 | 90.84 ± 1.64 | -8.02 ± 2.04 | 5.4 |
| ETS | 96.10 ± 0.00 | -3.98 ± 0.00 | 4.6 | 88.84 ± 0.00 | -10.55 ± 0.00 | 6.8 |
| Heur-GD | 97.96 ± 0.00 | -1.93 ± 0.00 | 2.3 | 93.21 ± 0.00 | -4.96 ± 0.00 | 4.4 |
| Heur-LD | 97.96 ± 0.00 | -1.93 ± 0.00 | 2.3 | 94.68 ± 0.00 | -3.12 ± 0.00 | 2.0 |
| Heur-AT | 91.95 ± 0.00 | -9.30 ± 0.00 | 5.8 | 93.54 ± 0.00 | -4.70 ± 0.00 | 3.0 |
| DQN (Ours) | 96.71 ± 0.69 | -3.47 ± 0.86 | 4.0 | 90.46 ± 4.25 | -8.60 ± 5.32 | 5.4 |
| A2C (Ours) | 89.02 ± 0.00 | -9.94 ± 0.00 | 7.0 | 98.74 ± 0.00 | -0.58 ± 0.00 | 1.0 |

| | Test Env. Seed 12 | | | Test Env. Seed 13 | | |
|---|---|---|---|---|---|---|
| **Method** | ACC (%) | BWT (%) | Rank | ACC (%) | BWT (%) | Rank |
| Random | 93.97 ± 4.60 | -6.60 ± 5.74 | 3.0 | 91.66 ± 3.08 | -9.38 ± 3.85 | 6.4 |
| ETS | 91.25 ± 0.00 | -10.08 ± 0.00 | 6.6 | 91.24 ± 0.00 | -9.85 ± 0.00 | 6.6 |
| Heur-GD | 94.97 ± 0.00 | -5.26 ± 0.00 | 4.2 | 95.09 ± 0.00 | -5.10 ± 0.00 | 5 |
| Heur-LD | 96.14 ± 0.00 | -3.89 ± 0.00 | 1.8 | 96.23 ± 0.00 | -3.69 ± 0.00 | 2.8 |
| Heur-AT | 95.84 ± 0.00 | -4.21 ± 0.00 | 3.0 | 98.10 ± 0.00 | -1.38 ± 0.00 | 1.2 |
| DQN (Ours) | 94.46 ± 3.12 | -6.00 ± 3.95 | 4 | 96.91 ± 0.90 | -2.83 ± 1.12 | 2.2 |
| A2C (Ours) | 93.72 ± 0.00 | -4.02 ± 0.00 | 5.4 | 96.12 ± 0.00 | -3.81 ± 0.00 | 3.8 |

| | Test Env. Seed 14 | | | Test Env. Seed 15 | | |
|---|---|---|---|---|---|---|
| **Method** | ACC (%) | BWT (%) | Rank | ACC (%) | BWT (%) | Rank |
| Random | 94.17 ± 1.37 | -3.79 ± 1.73 | 3.4 | 93.74 ± 0.95 | -4.53 ± 1.18 | 1.4 |
| ETS | 90.04 ± 0.00 | -8.92 ± 0.00 | 6.0 | 93.51 ± 0.00 | -4.81 ± 0.00 | 1.6 |
| Heur-GD | 95.37 ± 0.00 | -2.26 ± 0.00 | 1.6 | 79.33 ± 0.00 | -22.05 ± 0.00 | 7.0 |
| Heur-LD | 95.07 ± 0.00 | -2.65 ± 0.00 | 3.2 | 92.61 ± 0.00 | -5.44 ± 0.00 | 3.0 |
| Heur-AT | 81.98 ± 0.00 | -18.88 ± 0.00 | 7.0 | 88.54 ± 0.00 | -10.50 ± 0.00 | 5.8 |
| DQN (Ours) | 94.47 ± 2.12 | -3.12 ± 2.66 | 2.4 | 89.56 ± 0.59 | -9.25 ± 0.74 | 4.4 |
| A2C (Ours) | 93.93 ± 0.00 | -4.27 ± 0.00 | 4.4 | 89.18 ± 0.00 | -9.72 ± 0.00 | 4.8 |

| | Test Env. Seed 16 | | | Test Env. Seed 17 | | |
|---|---|---|---|---|---|---|
| **Method** | ACC (%) | BWT (%) | Rank | ACC (%) | BWT (%) | Rank |
| Random | 90.96 ± 1.68 | -7.18 ± 2.06 | 3.4 | 98.99 ± 0.26 | -0.66 ± 0.34 | 3.4 |
| ETS | 94.41 ± 0.00 | -2.96 ± 0.00 | 2.0 | 98.11 ± 0.00 | -1.77 ± 0.00 | 6.0 |
| Heur-GD | 73.82 ± 0.00 | -28.91 ± 0.00 | 7.0 | 99.37 ± 0.00 | -0.24 ± 0.00 | 1.0 |
| Heur-LD | 80.40 ± 0.00 | -20.66 ± 0.00 | 6.0 | 99.32 ± 0.00 | -0.30 ± 0.00 | 2.0 |
| Heur-AT | 89.24 ± 0.00 | -9.68 ± 0.00 | 4.4 | 98.74 ± 0.00 | -0.99 ± 0.00 | 4.2 |
| DQN (Ours) | 89.39 ± 2.47 | -9.49 ± 3.09 | 4.2 | 98.76 ± 0.31 | -0.97 ± 0.38 | 4.4 |
| A2C (Ours) | 94.52 ± 0.12 | -5.32 ± 0.16 | 1.0 | 97.18 ± 0.00 | -2.89 ± 0.00 | 7.0 |

| | Test Env. Seed 18 | | | Test Env. Seed 19 | | |
|---|---|---|---|---|---|---|
| **Method** | ACC (%) | BWT (%) | Rank | ACC (%) | BWT (%) | Rank |
| Random | 89.92 ± 3.44 | -11.26 ± 4.30 | 5.8 | 97.64 ± 0.79 | -1.58 ± 1.06 | 2.6 |
| ETS | 93.56 ± 0.00 | -6.74 ± 0.00 | 4.4 | 97.49 ± 0.00 | -1.81 ± 0.00 | 2.8 |
| Heur-GD | 92.92 ± 0.00 | -7.31 ± 0.00 | 5.4 | 95.79 ± 0.00 | -3.94 ± 0.00 | 5.4 |
| Heur-LD | 92.06 ± 0.00 | -8.40 ± 0.00 | 6.4 | 93.42 ± 0.00 | -6.85 ± 0.00 | 6.8 |
| Heur-AT | 94.22 ± 0.00 | -5.66 ± 0.00 | 3.0 | 96.57 ± 0.00 | -3.01 ± 0.00 | 4.2 |
| DQN (Ours) | 95.60 ± 0.74 | -3.92 ± 0.93 | 1.6 | 95.50 ± 1.72 | -4.42 ± 2.16 | 5.2 |
| A2C (Ours) | 95.98 ± 0.00 | -3.89 ± 0.00 | 1.4 | 99.14 ± 0.00 | -0.47 ± 0.00 | 1.0 |

Table 36: Performance comparison in every test environment with seed (10-19) with **Split notMNIST** for **New Task Order** experiment. Under each column named 'Test Env. Seed X', we show the mean and stddev. of ACC and BWT, and the Rank averaged over the RL seeds for the corresponding method.

| Method | Test Env. Seed 10 | | | Test Env. Seed 11 | | |
|---|---|---|---|---|---|---|
| | ACC (%) | BWT (%) | Rank | ACC (%) | BWT (%) | Rank |
| Random | 89.19 ± 3.60 | -9.93 ± 4.54 | 5.0 | 91.90 ± 1.05 | -2.01 ± 1.35 | 4.0 |
| ETS | 93.24 ± 0.00 | -4.74 ± 0.00 | 1.6 | 91.79 ± 0.00 | -2.28 ± 0.00 | 3.4 |
| Heur-GD | 91.58 ± 0.00 | -5.60 ± 0.00 | 4.0 | 90.92 ± 0.00 | -4.21 ± 0.00 | 6.8 |
| Heur-LD | 90.88 ± 0.00 | -6.51 ± 0.00 | 6.4 | 91.60 ± 0.00 | -3.31 ± 0.00 | 5.6 |
| Heur-AT | 91.36 ± 0.00 | -5.48 ± 0.00 | 5.0 | 91.70 ± 0.00 | -3.44 ± 0.00 | 4.6 |
| DQN | 93.70 ± 1.02 | -3.37 ± 0.72 | 1.8 | 92.73 ± 0.77 | -2.12 ± 0.65 | 1.8 |
| A2C | 91.78 ± 0.60 | -5.63 ± 0.49 | 4.2 | 92.66 ± 0.36 | -2.31 ± 0.33 | 1.8 |

| Method | Test Env. Seed 12 | | | Test Env. Seed 13 | | |
|---|---|---|---|---|---|---|
| | ACC (%) | BWT (%) | Rank | ACC (%) | BWT (%) | Rank |
| Random | 90.59 ± 2.85 | -7.94 ± 3.69 | 4.0 | 91.63 ± 2.40 | -3.73 ± 2.70 | 3.8 |
| ETS | 83.06 ± 0.00 | -17.48 ± 0.00 | 7.0 | 86.28 ± 0.00 | -10.76 ± 0.00 | 7.0 |
| Heur-GD | 92.89 ± 0.00 | -4.45 ± 0.00 | 2.2 | 94.06 ± 0.00 | -1.22 ± 0.00 | 1.2 |
| Heur-LD | 92.96 ± 0.00 | -3.57 ± 0.00 | 1.2 | 92.21 ± 0.00 | -3.53 ± 0.00 | 4.0 |
| Heur-AT | 91.20 ± 0.00 | -5.92 ± 0.00 | 4.6 | 88.77 ± 0.00 | -7.46 ± 0.00 | 5.8 |
| DQN | 91.59 ± 0.87 | -5.46 ± 0.95 | 3.8 | 92.30 ± 1.48 | -3.42 ± 1.86 | 3.5 |
| A2C | 91.02 ± 0.26 | -5.97 ± 0.19 | 5.2 | 93.52 ± 0.60 | -2.05 ± 0.44 | 2.7 |

| Method | Test Env. Seed 14 | | | Test Env. Seed 15 | | |
|---|---|---|---|---|---|---|
| | ACC (%) | BWT (%) | Rank | ACC (%) | BWT (%) | Rank |
| Random | 92.42 ± 0.80 | -3.95 ± 0.94 | 3.0 | 89.42 ± 1.66 | -6.65 ± 1.97 | 3.6 |
| ETS | 92.67 ± 0.00 | -3.89 ± 0.00 | 2.4 | 89.32 ± 0.00 | -6.41 ± 0.00 | 4.2 |
| Heur-GD | 93.63 ± 0.00 | -1.46 ± 0.00 | 1.0 | 92.08 ± 0.00 | -4.12 ± 0.00 | 1.0 |
| Heur-LD | 89.18 ± 0.00 | -7.02 ± 0.00 | 5.8 | 89.58 ± 0.00 | -7.25 ± 0.00 | 3.0 |
| Heur-AT | 88.14 ± 0.00 | -8.58 ± 0.00 | 6.8 | 84.47 ± 0.00 | -14.12 ± 0.00 | 6.8 |
| DQN | 90.69 ± 0.79 | -5.25 ± 0.95 | 4.8 | 86.99 ± 1.50 | -10.77 ± 1.87 | 5.6 |
| A2C | 91.47 ± 1.79 | -4.64 ± 2.34 | 4.2 | 89.04 ± 2.55 | -8.28 ± 3.07 | 3.8 |

| Method | Test Env. Seed 16 | | | Test Env. Seed 17 | | |
|---|---|---|---|---|---|---|
| | ACC (%) | BWT (%) | Rank | ACC (%) | BWT (%) | Rank |
| Random | 93.66 ± 0.54 | -0.94 ± 0.77 | 1.0 | 90.30 ± 2.35 | -7.20 ± 2.85 | 6.0 |
| ETS | 91.91 ± 0.00 | -2.65 ± 0.00 | 2.8 | 92.56 ± 0.00 | -3.99 ± 0.00 | 3.4 |
| Heur-GD | 89.22 ± 0.00 | -5.95 ± 0.00 | 4.7 | 91.30 ± 0.00 | -5.69 ± 0.00 | 5.9 |
| Heur-LD | 89.22 ± 0.00 | -5.95 ± 0.00 | 4.7 | 91.30 ± 0.00 | -5.69 ± 0.00 | 5.9 |
| Heur-AT | 86.29 ± 0.00 | -9.36 ± 0.00 | 7.0 | 93.87 ± 0.00 | -1.30 ± 0.00 | 2.0 |
| DQN | 88.94 ± 1.41 | -5.84 ± 1.93 | 5.2 | 94.53 ± 0.90 | -1.41 ± 1.11 | 1.2 |
| A2C | 91.85 ± 1.45 | -1.88 ± 1.43 | 2.6 | 92.58 ± 0.67 | -3.78 ± 0.99 | 3.6 |

| Method | Test Env. Seed 18 | | | Test Env. Seed 19 | | |
|---|---|---|---|---|---|---|
| | ACC (%) | BWT (%) | Rank | ACC (%) | BWT (%) | Rank |
| Random | 94.78 ± 2.97 | -2.78 ± 3.76 | 2.0 | 92.83 ± 2.19 | -2.62 ± 2.73 | 4.4 |
| ETS | 92.43 ± 0.00 | -5.57 ± 0.00 | 5.8 | 90.89 ± 0.00 | -5.17 ± 0.00 | 6.8 |
| Heur-GD | 92.79 ± 0.00 | -5.69 ± 0.00 | 4.4 | 94.16 ± 0.00 | -2.21 ± 0.00 | 3.2 |
| Heur-LD | 94.65 ± 0.00 | -3.71 ± 0.00 | 1.8 | 94.94 ± 0.00 | -1.01 ± 0.00 | 1.2 |
| Heur-AT | 88.15 ± 0.00 | -11.30 ± 0.00 | 7 | 93.46 ± 0.00 | -3.99 ± 0.00 | 5.4 |
| DQN | 93.32 ± 0.69 | -4.77 ± 1.02 | 3.8 | 94.08 ± 0.96 | -3.10 ± 1.02 | 3.6 |
| A2C | 93.64 ± 0.22 | -4.58 ± 0.48 | 3.2 | 93.66 ± 0.80 | -3.82 ± 0.94 | 3.4 |

Table 37: Two-tailed Welch's *t*-test results for **Split notMNIST** in **New Task Order** experiment. We bold the *p*-values where $p < 0.05$ to indicate for when a method is statistically significantly better than its comparison.

| | Test Env. Seed 10 | | Test Env. Seed 11 | | Test Env. Seed 12 | | Test Env. Seed 13 | | Test Env. Seed 14 | |
|---|---|---|---|---|---|---|---|---|---|---|
| **Methods** | *t* | *p* | *t* | *p* | *t* | *p* | *t* | *p* | *t* | *p* |
| DQN vs Random | -1.32 | 0.226 | 2.81 | **0.023** | 2.64 | **0.033** | 2.22 | 0.083 | -0.41 | 0.692 |
| DQN vs ETS | -0.36 | **0.739** | 3.08 | **0.037** | 0.68 | 0.535 | 8.68 | **0.001** | 0.20 | 0.849 |
| DQN vs Heur-GD | -1.20 | 0.296 | 6.06 | **0.004** | 9.21 | **0.001** | -7.65 | **0.002** | -0.72 | 0.512 |
| DQN vs Heur-LD: | -1.20 | 0.296 | 6.06 | **0.004** | 9.21 | **0.001** | -7.65 | **0.002** | -1.56 | 0.194 |
| DQN vs Heur-AT | -1.20 | 0.296 | 6.06 | **0.004** | 9.21 | **0.001** | -7.65 | **0.002** | -0.72 | 0.512 |
| DQN vs A2C | -0.63 | 0.554 | 0.78 | 0.458 | 1.29 | 0.245 | -3.88 | **0.005** | -1.64 | 0.172 |
| A2C vs Random | -1.04 | 0.333 | 1.74 | 0.122 | 2.10 | 0.091 | 3.57 | **0.019** | 1.68 | 0.159 |
| A2C vs ETS | 0.67 | 0.537 | 1.57 | 0.191 | -1.56 | 0.194 | 12.84 | **0.000** | 9.59 | **0.001** |
| A2C vs Heur-GD | -0.96 | 0.390 | 4.08 | **0.015** | 15.93 | **0.000** | -1.58 | 0.189 | 4.86 | **0.008** |
| A2C vs Heur-LD | -0.96 | 0.390 | 4.08 | **0.015** | 15.93 | **0.000** | -1.58 | 0.189 | 0.56 | 0.606 |
| A2C vs Heur-AT | -0.96 | 0.390 | 4.08 | **0.015** | 15.93 | **0.000** | -1.58 | 0.189 | 4.86 | 0.008 |
| A2C vs DQN | 0.63 | 0.554 | -0.78 | 0.458 | -1.29 | 0.245 | 3.88 | **0.005** | 1.64 | 0.172 |

| | Test Env. Seed 15 | | Test Env. Seed 16 | | Test Env. Seed 17 | | Test Env. Seed 18 | | Test Env. Seed 19 | |
|---|---|---|---|---|---|---|---|---|---|---|
| Methods | *t* | *p* | *t* | *p* | *t* | *p* | *t* | *p* | *t* | *p* |
| DQN vs Random | 1.82 | 0.108 | 1.05 | 0.332 | -1.16 | 0.283 | 0.85 | 0.438 | -3.30 | **0.023** |
| DQN vs ETS | 3.61 | **0.023** | 2.01 | 0.115 | -3.07 | **0.037** | 7.25 | **0.002** | -4.29 | **0.013** |
| DQN vs Heur-GD | 1.67 | 0.169 | 2.66 | 0.056 | 15.47 | **0.000** | -0.78 | 0.479 | 2.69 | 0.055 |
| DQN vs Heur-LD: | 1.67 | 0.169 | 9.49 | **0.001** | 8.95 | **0.001** | 8.72 | **0.001** | 2.69 | 0.055 |
| DQN vs Heur-AT | 1.67 | 0.169 | 2.66 | 0.056 | 0.32 | 0.767 | -0.78 | 0.479 | 2.69 | 0.055 |
| DQN vs A2C | -0.38 | 0.722 | 1.63 | 0.142 | -0.39 | 0.710 | -1.43 | 0.226 | -2.71 | 0.053 |
| A2C vs Random | 2.56 | 0.060 | -0.07 | 0.944 | -0.83 | 0.430 | 8.71 | **0.001** | -2.39 | 0.073 |
| A2C vs ETS | 18.34 | **0.000** | -0.40 | 0.707 | -2.50 | 0.067 | inf | **0.000** | -39.30 | **0.000** |
| A2C vs Heur-GD | 9.47 | **0.001** | 0.18 | 0.867 | 15.87 | **0.000** | inf | **0.000** | 135.29 | **0.000** |
| A2C vs Heur-LD | 9.47 | **0.001** | 6.23 | **0.003** | 9.41 | **0.001** | inf | **0.000** | 135.29 | **0.000** |
| A2C vs Heur-AT | 9.47 | **0.001** | 0.18 | 0.867 | 0.86 | 0.439 | inf | **0.000** | 135.29 | **0.000** |
| A2C vs DQN | 0.38 | 0.722 | -1.63 | 0.142 | 0.39 | 0.710 | 1.43 | 0.226 | 2.71 | 0.053 |

Table 38: Two-tailed Welch's *t*-test results for **Split CIFAR-10** in **New Task Order** experiment. We bold the *p*-values where $p < 0.05$ to indicate for when a method is statistically significantly better than its comparison.

| | Test Env. Seed 10 | | Test Env. Seed 11 | | Test Env. Seed 12 | | Test Env. Seed 13 | | Test Env. Seed 14 | |
|---|---|---|---|---|---|---|---|---|---|---|
| **Methods** | *t* | *p* | *t* | *p* | *t* | *p* | *t* | *p* | *t* | *p* |
| DQN vs Random | 0.70 | 0.504 | 8.12 | **0.000** | -3.77 | **0.006** | 0.48 | 0.653 | 2.03 | 0.091 |
| DQN vs ETS | -0.07 | 0.948 | 12.90 | **0.000** | -0.79 | 0.473 | 12.33 | **0.000** | 12.12 | **0.000** |
| DQN vs Heur-GD | 1.18 | 0.303 | -0.09 | 0.932 | -10.41 | **0.000** | 5.18 | **0.007** | -3.77 | **0.020** |
| DQN vs Heur-LD: | 0.61 | 0.574 | -6.72 | **0.003** | -5.31 | **0.006** | -1.29 | 0.265 | 1.24 | 0.283 |
| DQN vs Heur-AT | 1.77 | 0.152 | -5.15 | **0.007** | -9.97 | **0.001** | 0.84 | 0.448 | -3.26 | **0.031** |
| DQN vs A2C | 3.97 | **0.017** | -10.54 | **0.000** | -2.38 | 0.051 | -2.70 | **0.049** | -1.86 | 0.127 |
| A2C vs Random | -2.29 | 0.084 | 13.83 | **0.000** | -2.31 | 0.061 | 1.63 | 0.178 | 3.15 | **0.033** |
| A2C vs ETS | -inf | **0.000** | inf | **0.000** | 3.19 | **0.033** | 68.53 | **0.000** | 50.37 | **0.000** |
| A2C vs Heur-GD | -inf | **0.000** | inf | **0.000** | -12.02 | **0.000** | 36.03 | **0.000** | -6.59 | **0.003** |
| A2C vs Heur-LD | -inf | **0.000** | inf | **0.000** | -3.96 | **0.017** | 6.66 | **0.003** | 11.37 | **0.000** |
| A2C vs Heur-AT | -inf | **0.000** | inf | **0.000** | -11.34 | **0.000** | 16.34 | **0.000** | -4.77 | **0.009** |
| A2C vs DQN | -3.97 | **0.017** | 10.54 | **0.000** | 2.38 | 0.051 | 2.70 | 0.049 | 1.86 | 0.127 |

| | Test Env. Seed 15 | | Test Env. Seed 16 | | Test Env. Seed 17 | | Test Env. Seed 18 | | Test Env. Seed 19 | |
|---|---|---|---|---|---|---|---|---|---|---|
| Methods | *t* | *p* | *t* | *p* | *t* | *p* | *t* | *p* | *t* | *p* |
| DQN vs Random | -2.67 | **0.045** | 0.07 | 0.945 | 1.35 | 0.222 | 1.53 | 0.165 | 4.03 | **0.010** |
| DQN vs ETS | -1.66 | 0.172 | -1.08 | 0.342 | 2.69 | 0.055 | 0.20 | 0.850 | 5.14 | **0.007** |
| DQN vs Heur-GD | -1.38 | 0.239 | 7.09 | **0.002** | 0.56 | 0.608 | -0.38 | 0.725 | 4.68 | **0.009** |
| DQN vs Heur-LD: | -1.98 | 0.119 | 9.15 | **0.001** | -0.54 | 0.621 | -3.08 | **0.037** | 5.53 | **0.005** |
| DQN vs Heur-AT | -1.98 | 0.119 | 7.09 | **0.002** | -0.74 | 0.502 | -0.38 | 0.725 | 4.68 | **0.009** |
| DQN vs A2C | -2.65 | 0.054 | -2.90 | **0.044** | -0.33 | 0.756 | -1.86 | 0.119 | -1.31 | 0.259 |
| A2C vs Random | -0.35 | 0.740 | 0.50 | 0.643 | 3.17 | **0.034** | 4.00 | **0.009** | 4.80 | **0.009** |
| A2C vs ETS | 6.26 | **0.003** | inf | **0.000** | inf | **0.000** | 5.35 | **0.006** | inf | **0.000** |
| A2C vs Heur-GD | 7.95 | **0.001** | inf | **0.000** | inf | **0.000** | 3.95 | **0.017** | inf | **0.000** |
| A2C vs Heur-LD | 4.31 | **0.013** | inf | **0.000** | -inf | **0.000** | -2.58 | 0.061 | inf | **0.000** |
| A2C vs Heur-AT | 4.31 | **0.013** | inf | **0.000** | -inf | **0.000** | 3.95 | **0.017** | inf | **0.000** |
| A2C vs DQN | 2.65 | 0.054 | 2.90 | **0.044** | 0.33 | 0.756 | 1.86 | 0.119 | 1.31 | 0.259 |

Table 39: Performance comparison in every test environment with seed (10-19) with with **Split CIFAR-10** for **New Task Order** experiment. Under each column named 'Test Env. Seed X', we show the mean and stddev. of ACC and BWT, and the Rank averaged over the RL seeds for the corresponding method.

| | Test Env. Seed 10 | | | Test Env. Seed 11 | | |
|---|---|---|---|---|---|---|
| **Method** | ACC (%) | BWT (%) | Rank | ACC (%) | BWT (%) | Rank |
| Random | 86.35 ± 1.57 | -13.27 ± 1.98 | 4.2 | 73.05 ± 1.42 | -27.40 ± 1.91 | 6.8 |
| ETS | 87.10 ± 0.00 | -12.31 ± 0.00 | 1.8 | 75.16 ± 0.00 | -25.14 ± 0.00 | 6.2 |
| Heur-GD | 86.31 ± 0.00 | -13.19 ± 0.00 | 4.0 | 79.45 ± 0.00 | -19.79 ± 0.00 | 4.6 |
| Heur-LD | 86.67 ± 0.00 | -12.74 ± 0.00 | 3.0 | 81.64 ± 0.00 | -17.05 ± 0.00 | 2.0 |
| Heur-AT | 85.94 ± 0.00 | -13.69 ± 0.00 | 5.2 | 81.12 ± 0.00 | -17.74 ± 0.00 | 3.0 |
| DQN | 87.06 ± 1.26 | -12.42 ± 1.57 | 2.8 | 79.42 ± 0.66 | -19.64 ± 0.83 | 4.4 |
| A2C | 84.55 ± 0.00 | -15.59 ± 0.00 | 7.0 | 82.90 ± 0.00 | -15.26 ± 0.00 | 1.0 |

| | Test Env. Seed 12 | | | Test Env. Seed 13 | | |
|---|---|---|---|---|---|---|
| **Method** | ACC (%) | BWT (%) | Rank | ACC (%) | BWT (%) | Rank |
| Random | 87.86 ± 1.12 | -10.36 ± 1.44 | 3.4 | 80.07 ± 1.43 | -15.40 ± 1.78 | 3.3 |
| ETS | 85.53 ± 0.00 | -13.33 ± 0.00 | 6.6 | 76.85 ± 0.00 | -19.34 ± 0.00 | 7.0 |
| Heur-GD | 89.76 ± 0.00 | -7.69 ± 0.00 | 1.0 | 78.93 ± 0.00 | -16.48 ± 0.00 | 5.8 |
| Heur-LD | 87.52 ± 0.00 | -10.40 ± 0.00 | 3.8 | 80.81 ± 0.00 | -14.20 ± 0.00 | 2.8 |
| Heur-AT | 89.57 ± 0.00 | -8.15 ± 0.00 | 2.0 | 80.19 ± 0.00 | -14.77 ± 0.00 | 4.3 |
| DQN | 85.18 ± 0.88 | -13.31 ± 1.12 | 6.4 | 80.43 ± 0.58 | -14.47 ± 0.75 | 3.4 |
| A2C | 86.42 ± 0.56 | -11.90 ± 0.75 | 4.8 | 81.24 ± 0.13 | -13.35 ± 0.15 | 1.4 |

| | Test Env. Seed 14 | | | Test Env. Seed 15 | | |
|---|---|---|---|---|---|---|
| **Method** | ACC (%) | BWT (%) | Rank | ACC (%) | BWT (%) | Rank |
| Random | 82.04 ± 2.37 | -17.84 ± 2.95 | 6.0 | 84.86 ± 0.62 | -14.77 ± 0.79 | 2.0 |
| ETS | 77.79 ± 0.00 | -23.26 ± 0.00 | 6.8 | 83.78 ± 0.00 | -16.15 ± 0.00 | 5.2 |
| Heur-GD | 86.86 ± 0.00 | -11.70 ± 0.00 | 1.0 | 83.52 ± 0.00 | -16.29 ± 0.00 | 6.2 |
| Heur-LD | 84.00 ± 0.00 | -15.24 ± 0.00 | 4.8 | 84.08 ± 0.00 | -15.58 ± 0.00 | 3.5 |
| Heur-AT | 86.57 ± 0.00 | -12.05 ± 0.00 | 2.0 | 84.08 ± 0.00 | -15.58 ± 0.00 | 3.5 |
| DQN | 84.71 ± 1.14 | -14.62 ± 1.36 | 4.2 | 82.22 ± 1.88 | -17.83 ± 2.38 | 5.8 |
| A2C | 85.81 ± 0.32 | -13.35 ± 0.40 | 3.2 | 84.74 ± 0.31 | -14.65 ± 0.39 | 1.8 |

| | Test Env. Seed 16 | | | Test Env. Seed 17 | | |
|---|---|---|---|---|---|---|
| **Method** | ACC (%) | BWT (%) | Rank | ACC (%) | BWT (%) | Rank |
| Random | 87.60 ± 1.72 | -10.51 ± 2.15 | 4.0 | 72.33 ± 2.14 | -26.27 ± 2.73 | 6.0 |
| ETS | 87.80 ± 0.00 | -10.20 ± 0.00 | 2.8 | 70.35 ± 0.00 | -28.79 ± 0.00 | 6.6 |
| Heur-GD | 86.77 ± 0.00 | -12.23 ± 0.00 | 5.1 | 74.14 ± 0.00 | -23.30 ± 0.00 | 4.8 |
| Heur-LD | 86.51 ± 0.00 | -12.63 ± 0.00 | 6.6 | 76.08 ± 0.00 | -20.88 ± 0.00 | 2.6 |
| Heur-AT | 86.77 ± 0.00 | -12.23 ± 0.00 | 5.1 | 76.44 ± 0.00 | -20.49 ± 0.00 | 1.4 |
| DQN | 87.66 ± 0.25 | -11.02 ± 0.33 | 2.8 | 75.13 ± 3.56 | -21.96 ± 4.45 | 3.0 |
| A2C | 88.03 ± 0.00 | -10.59 ± 0.00 | 1.6 | 75.72 ± 0.00 | -21.18 ± 0.00 | 3.6 |

| | Test Env. Seed 18 | | | Test Env. Seed 19 | | |
|---|---|---|---|---|---|---|
| **Method** | ACC (%) | BWT (%) | Rank | ACC (%) | BWT (%) | Rank |
| Random | 87.43 ± 0.88 | -10.75 ± 1.03 | 6.4 | 74.89 ± 2.33 | -25.80 ± 2.89 | 7.0 |
| ETS | 88.32 ± 0.00 | -10.19 ± 0.00 | 5.8 | 77.67 ± 0.00 | -22.16 ± 0.00 | 5.0 |
| Heur-GD | 88.59 ± 0.00 | -10.34 ± 0.00 | 4.3 | 77.87 ± 0.00 | -22.98 ± 0.00 | 3.5 |
| Heur-LD | 89.85 ± 0.00 | -8.95 ± 0.00 | 1.2 | 77.50 ± 0.00 | -23.50 ± 0.00 | 6.0 |
| Heur-AT | 88.59 ± 0.00 | -10.34 ± 0.00 | 4.3 | 77.87 ± 0.00 | -22.98 ± 0.00 | 3.5 |
| DQN | 88.41 ± 0.93 | -10.73 ± 1.15 | 3.8 | 79.91 ± 0.87 | -20.52 ± 1.07 | 1.7 |
| A2C | 89.35 ± 0.39 | -9.47 ± 0.48 | 2.2 | 80.48 ± 0.00 | -19.78 ± 0.00 | 1.3 |

Table 40: Performance comparison in every test environment with seed (0-9) with with **Split notMNIST** for **New Dataset** experiment. Under each column named 'Test Env. Seed X', we show the mean and stddev. of ACC and BWT, and the Rank averaged over the RL seeds for the corresponding method.

| | Test Env. Seed 0 | | | Test Env. Seed 1 | | |
|---|---|---|---|---|---|---|
| **Method** | ACC (%) | BWT (%) | Rank | ACC (%) | BWT (%) | Rank |
| Random | 93.09 ± 2.31 | -3.66 ± 2.82 | 2.4 | 92.13 ± 0.80 | -3.91 ± 0.93 | 3.8 |
| ETS | 91.08 ± 0.00 | -6.61 ± 0.00 | 5.8 | 92.46 ± 0.00 | -3.75 ± 0.00 | 2.8 |
| Heur-GD | 92.41 ± 0.00 | -3.55 ± 0.00 | 3.8 | 91.18 ± 0.00 | -6.26 ± 0.00 | 5.8 |
| Heur-LD | 92.41 ± 0.00 | -3.55 ± 0.00 | 3.8 | 91.18 ± 0.00 | -6.26 ± 0.00 | 5.8 |
| Heur-AT | 92.41 ± 0.00 | -3.55 ± 0.00 | 3.8 | 91.18 ± 0.00 | -6.26 ± 0.00 | 5.8 |
| DQN | 90.52 ± 3.15 | -6.92 ± 3.57 | 4.6 | 93.79 ± 0.86 | -1.65 ± 1.03 | 1.6 |
| A2C | 91.62 ± 1.62 | -5.47 ± 2.49 | 3.8 | 93.26 ± 1.02 | -2.12 ± 1.41 | 2.4 |

| | Test Env. Seed 2 | | | Test Env. Seed 3 | | |
|---|---|---|---|---|---|---|
| **Method** | ACC (%) | BWT (%) | Rank | ACC (%) | BWT (%) | Rank |
| Random | 87.13 ± 3.01 | -10.47 ± 3.60 | 4.4 | 89.04 ± 3.01 | -8.60 ± 3.86 | 6.4 |
| ETS | 91.23 ± 0.00 | -5.23 ± 0.00 | 2.0 | 89.33 ± 0.00 | -8.28 ± 0.00 | 6.4 |
| Heur-GD | 82.58 ± 0.00 | -15.83 ± 0.00 | 5.8 | 95.26 ± 0.00 | -0.99 ± 0.00 | 2.0 |
| Heur-LD | 82.58 ± 0.00 | -15.83 ± 0.00 | 5.8 | 95.26 ± 0.00 | -0.99 ± 0.00 | 2.0 |
| Heur-AT | 82.58 ± 0.00 | -15.83 ± 0.00 | 5.8 | 95.26 ± 0.00 | -0.99 ± 0.00 | 2.0 |
| DQN | 91.92 ± 2.03 | -4.77 ± 2.31 | 1.6 | 92.49 ± 0.73 | -4.69 ± 0.54 | 5.2 |
| A2C | 90.46 ± 0.99 | -6.03 ± 1.24 | 2.6 | 94.61 ± 0.82 | -2.06 ± 0.46 | 4.0 |

| | Test Env. Seed 4 | | | Test Env. Seed 5 | | |
|---|---|---|---|---|---|---|
| **Method** | ACC (%) | BWT (%) | Rank | ACC (%) | BWT (%) | Rank |
| Random | 91.37 ± 1.63 | -5.04 ± 2.02 | 4.6 | 91.06 ± 1.45 | -5.87 ± 1.82 | 5.6 |
| ETS | 90.52 ± 0.00 | -5.81 ± 0.00 | 6.2 | 90.73 ± 0.00 | -6.00 ± 0.00 | 6.4 |
| Heur-GD | 91.64 ± 0.00 | -6.55 ± 0.00 | 4.3 | 91.80 ± 0.00 | -3.52 ± 0.00 | 4.0 |
| Heur-LD | 92.66 ± 0.00 | -5.28 ± 0.00 | 2.4 | 91.80 ± 0.00 | -3.52 ± 0.00 | 4.0 |
| Heur-AT | 91.64 ± 0.00 | -6.55 ± 0.00 | 4.3 | 91.80 ± 0.00 | -3.52 ± 0.00 | 4.0 |
| DQN | 90.76 ± 2.44 | -6.15 ± 3.21 | 4.4 | 92.72 ± 1.11 | -3.49 ± 1.10 | 2.2 |
| A2C | 92.80 ± 0.48 | -4.09 ± 0.48 | 1.8 | 92.94 ± 0.24 | -3.00 ± 0.29 | 1.8 |

| | Test Env. Seed 6 | | | Test Env. Seed 7 | | |
|---|---|---|---|---|---|---|
| **Method** | ACC (%) | BWT (%) | Rank | ACC (%) | BWT (%) | Rank |
| Random | 91.80 ± 2.56 | -4.63 ± 3.25 | 3.0 | 93.72 ± 0.78 | -3.20 ± 0.94 | 2.4 |
| ETS | 91.99 ± 0.00 | -5.12 ± 0.00 | 2.6 | 94.03 ± 0.00 | -1.98 ± 0.00 | 1.4 |
| Heur-GD | 91.55 ± 0.00 | -4.47 ± 0.00 | 4.7 | 88.81 ± 0.00 | -8.97 ± 0.00 | 7.0 |
| Heur-LD | 87.03 ± 0.00 | -10.13 ± 0.00 | 7 | 90.65 ± 0.00 | -5.98 ± 0.00 | 6.0 |
| Heur-AT | 91.55 ± 0.00 | -4.47 ± 0.00 | 4.7 | 93.08 ± 0.00 | -2.95 ± 0.00 | 3.8 |
| DQN | 93.32 ± 1.33 | -2.59 ± 1.27 | 2.4 | 93.17 ± 0.56 | -2.79 ± 0.55 | 4.0 |
| A2C | 91.69 ± 1.50 | -4.35 ± 1.87 | 3.6 | 93.32 ± 0.57 | -2.46 ± 0.59 | 3.4 |

| | Test Env. Seed 8 | | | Test Env. Seed 9 | | |
|---|---|---|---|---|---|---|
| **Method** | ACC (%) | BWT (%) | Rank | ACC (%) | BWT (%) | Rank |
| Random | 92.62 ± 0.30 | -4.15 ± 0.61 | 4.6 | 92.48 ± 1.34 | -5.36 ± 1.50 | 2.2 |
| ETS | 89.04 ± 0.00 | -8.36 ± 0.00 | 6.0 | 94.29 ± 0.00 | -3.55 ± 0.00 | 1.0 |
| Heur-GD | 93.56 ± 0.00 | -2.86 ± 0.00 | 2.9 | 79.09 ± 0.00 | -22.95 ± 0.00 | 5.8 |
| Heur-LD | 88.21 ± 0.00 | -9.54 ± 0.00 | 7.0 | 79.09 ± 0.00 | -22.95 ± 0.00 | 5.8 |
| Heur-AT | 93.56 ± 0.00 | -2.86 ± 0.00 | 2.9 | 79.09 ± 0.00 | -22.95 ± 0.00 | 5.8 |
| DQN | 93.12 ± 1.13 | -3.20 ± 1.38 | 3.4 | 84.95 ± 4.36 | -15.42 ± 5.12 | 4.6 |
| A2C | 93.92 ± 0.00 | -2.53 ± 0.00 | 1.2 | 90.87 ± 0.17 | -8.58 ± 0.22 | 2.8 |

Table 41: Two-tailed Welch's $t$-test results for **Split notMNIST** in **New Dataset** experiment. We bold the $p$-values where $p < 0.05$ to indicate for when a method is statistically significantly better than its comparison.

| Methods | Test Env. Seed 0 | | Test Env. Seed 1 | | Test Env. Seed 2 | | Test Env. Seed 3 | | Test Env. Seed 4 | |
|---|---|---|---|---|---|---|---|---|---|---|
| | $t$ | $p$ | $t$ | $p$ | $t$ | $p$ | $t$ | $p$ | $t$ | $p$ |
| DQN vs Random | 2.41 | **0.065** | 1.27 | 0.242 | 0.67 | 0.534 | 0.47 | 0.651 | -3.09 | **0.015** |
| DQN vs ETS | 0.89 | 0.423 | 2.45 | 0.071 | 19.64 | **0.000** | 8.12 | **0.001** | -5.03 | **0.007** |
| DQN vs Heur-GD | 4.15 | **0.014** | 4.72 | **0.009** | -3.00 | **0.040** | -2.37 | 0.077 | -7.47 | **0.002** |
| DQN vs Heur-LD: | 5.52 | **0.005** | 2.96 | **0.041** | -3.16 | **0.034** | 0.13 | 0.906 | 3.84 | **0.018** |
| DQN vs Heur-AT | 4.58 | **0.010** | 2.70 | 0.054 | 0.88 | 0.428 | 4.76 | **0.009** | 6.47 | **0.003** |
| DQN vs A2C | 3.23 | **0.016** | 0.17 | 0.870 | 1.25 | 0.271 | -1.52 | 0.186 | -0.80 | 0.457 |
| A2C vs Random | 1.42 | 0.226 | 1.36 | 0.234 | 0.30 | 0.777 | 1.52 | 0.194 | -0.97 | 0.372 |
| A2C vs ETS | -4.85 | **0.008** | 4.73 | **0.009** | 60.22 | **0.000** | 24.32 | **0.000** | -1.34 | 0.251 |
| A2C vs Heur-GD | 0.67 | 0.537 | 9.51 | **0.001** | -14.14 | **0.000** | -1.82 | 0.142 | -2.42 | 0.073 |
| A2C vs Heur-LD | 2.99 | **0.040** | 5.82 | **0.004** | -14.66 | **0.000** | 4.40 | **0.012** | 2.57 | 0.062 |
| A2C vs Heur-AT | 1.40 | 0.233 | 5.27 | **0.006** | -1.39 | 0.238 | 15.96 | **0.000** | 3.73 | **0.020** |
| A2C vs DQN | -3.23 | **0.016** | -0.17 | 0.870 | -1.25 | 0.271 | 1.52 | 0.186 | 0.80 | 0.457 |

| Methods | Test Env. Seed 5 | | Test Env. Seed 6 | | Test Env. Seed 7 | | Test Env. Seed 8 | | Test Env. Seed 9 | |
|---|---|---|---|---|---|---|---|---|---|---|
| | $t$ | $p$ | $t$ | $p$ | $t$ | $p$ | $t$ | $p$ | $t$ | $p$ |
| DQN vs Random | -2.17 | 0.062 | -6.27 | **0.001** | 3.36 | **0.019** | -0.96 | 0.387 | 1.04 | 0.340 |
| DQN vs ETS | -3.11 | **0.036** | -4.23 | **0.013** | 4.40 | **0.012** | 2.60 | 0.060 | 6.64 | **0.003** |
| DQN vs Heur-GD | -6.80 | **0.002** | -0.40 | 0.708 | 7.21 | **0.002** | 1.56 | 0.194 | -0.16 | 0.881 |
| DQN vs Heur-LD: | -3.45 | **0.026** | -0.40 | 0.708 | 7.21 | **0.002** | -3.85 | **0.018** | -1.79 | 0.149 |
| DQN vs Heur-AT | 3.37 | **0.028** | 3.77 | **0.020** | 1.47 | 0.214 | 15.03 | **0.000** | 1.29 | 0.267 |
| DQN vs A2C | -1.39 | 0.211 | -2.89 | **0.020** | 3.50 | **0.009** | -0.87 | 0.426 | 0.68 | 0.514 |
| A2C vs Random | -0.25 | 0.809 | -2.32 | 0.067 | 1.86 | 0.126 | -0.77 | 0.483 | 0.70 | 0.512 |
| A2C vs ETS | -0.22 | 0.837 | -0.07 | 0.945 | 0.06 | 0.958 | 11.20 | **0.000** | 6.87 | **0.002** |
| A2C vs Heur-GD | -2.39 | 0.075 | 3.62 | **0.022** | 3.84 | **0.018** | 7.88 | **0.001** | -1.26 | 0.277 |
| A2C vs Heur-LD | -0.42 | 0.694 | 3.62 | **0.022** | 3.84 | **0.018** | -9.37 | **0.001** | -3.20 | **0.033** |
| A2C vs Heur-AT | 3.59 | **0.023** | 7.66 | **0.002** | -3.89 | **0.018** | 50.85 | **0.000** | 0.48 | 0.659 |
| A2C vs DQN | 1.39 | 0.211 | 2.89 | **0.020** | -3.50 | **0.009** | 0.87 | 0.426 | -0.68 | 0.514 |

Table 42: Two-tailed Welch's $t$-test results for **Split FashionMNIST** in **New Dataset** experiment. We bold the $p$-values where $p < 0.05$ to indicate for when a method is statistically significantly better than its comparison.

| Methods | Test Env. Seed 0 | | Test Env. Seed 1 | | Test Env. Seed 2 | | Test Env. Seed 3 | | Test Env. Seed 4 | |
|---|---|---|---|---|---|---|---|---|---|---|
| | $t$ | $p$ | $t$ | $p$ | $t$ | $p$ | $t$ | $p$ | $t$ | $p$ |
| DQN vs Random | 1.89 | 0.125 | 0.14 | 0.895 | 1.7 | 0.153 | 0.86 | 0.418 | 1.43 | 0.194 |
| DQN vs ETS | 0.47 | 0.663 | -1.05 | 0.353 | 22.86 | **0.000** | 4.61 | **0.010** | 0.62 | 0.568 |
| DQN vs Heur-GD | -1.77 | 0.152 | -1.68 | 0.167 | -2.47 | 0.069 | -0.26 | 0.81 | -1.58 | 0.189 |
| DQN vs Heur-LD | -0.9 | 0.419 | 1.78 | 0.149 | -2.21 | 0.091 | 3.83 | **0.019** | 0.79 | 0.474 |
| DQN vs Heur-AT | -0.4 | 0.711 | -1.04 | 0.356 | -3.04 | **0.038** | -2.08 | 0.106 | 2.6 | 0.06 |
| DQN vs A2C | -1.17 | 0.297 | 6.61 | **0.000** | 5.84 | **0.001** | 2.49 | **0.048** | -4.31 | **0.012** |
| A2C vs Random | 2.22 | **0.090** | -5.27 | **0.001** | -1.35 | 0.227 | -1.67 | 0.137 | 4.94 | **0.008** |
| A2C vs ETS | 5.15 | **0.007** | -12.04 | **0.000** | 6.69 | **0.003** | -0.46 | 0.67 | 52.33 | **0.000** |
| A2C vs Heur-GD | -1.63 | 0.179 | -12.84 | **0.000** | -8.23 | **0.001** | -2.91 | **0.044** | 29.05 | **0.000** |
| A2C vs Heur-LD | 1 | 0.374 | -8.43 | **0.001** | -8.08 | **0.001** | -0.85 | 0.441 | 54.1 | **0.000** |
| A2C vs Heur-AT | 2.53 | 0.065 | -12.03 | **0.000** | -8.57 | **0.001** | -3.83 | **0.019** | 73.24 | **0.000** |
| A2C vs DQN | 1.17 | 0.297 | -6.61 | **0.000** | -5.84 | **0.001** | -2.49 | **0.048** | 4.31 | **0.012** |

| Methods | Test Env. Seed 5 | | Test Env. Seed 6 | | Test Env. Seed 7 | | Test Env. Seed 8 | | Test Env. Seed 9 | |
|---|---|---|---|---|---|---|---|---|---|---|
| | $t$ | $p$ | $t$ | $p$ | $t$ | $p$ | $t$ | $p$ | $t$ | $p$ |
| DQN vs Random | -0.87 | 0.411 | -3.75 | **0.019** | -2.95 | **0.029** | 0.08 | 0.935 | 0.25 | 0.812 |
| DQN vs ETS | -3.57 | **0.023** | -3.82 | **0.019** | -4.18 | **0.014** | -0.77 | 0.482 | -1.44 | 0.223 |
| DQN vs Heur-GD | -2.18 | 0.095 | -2.14 | 0.099 | -0.32 | 0.768 | -0.88 | 0.427 | 0.06 | 0.956 |
| DQN vs Heur-LD | 0.51 | 0.635 | -0.74 | 0.499 | -4 | **0.016** | 3.01 | **0.040** | 5.52 | **0.005** |
| DQN vs Heur-AT | -2.41 | 0.073 | 2.14 | 0.099 | -5.78 | **0.004** | -0.3 | 0.779 | 5.24 | **0.006** |
| DQN vs A2C | -10.73 | **0.000** | -3.75 | **0.020** | 0.37 | 0.72 | 0.82 | 0.443 | 3.14 | **0.014** |
| A2C vs Random | 6.46 | **0.003** | -0.1 | 0.923 | -2.62 | 0.05 | -0.65 | 0.539 | -1.93 | 0.097 |
| A2C vs ETS | 514.01 | **0.000** | -0.78 | 0.481 | -3.43 | **0.026** | -3.12 | **0.036** | -5.32 | **0.006** |
| A2C vs Heur-GD | 614.01 | **0.000** | 22.55 | **0.000** | -0.68 | 0.533 | -3.32 | **0.030** | -4.06 | **0.015** |
| A2C vs Heur-LD | 807.34 | **0.000** | 42.04 | **0.000** | -3.3 | **0.030** | 3.78 | **0.019** | 0.56 | 0.606 |
| A2C vs Heur-AT | 597.34 | **0.000** | 82.22 | **0.000** | -4.57 | **0.010** | -2.25 | 0.087 | 0.32 | 0.764 |
| A2C vs DQN | 10.73 | **0.000** | 3.75 | **0.020** | -0.37 | 0.72 | -0.82 | 0.443 | -3.14 | **0.014** |

Table 43: Performance comparison in every test environment with seed (0-9) with with **Split FashionM-NIST** for **New Dataset** experiment. Under each column named 'Test Env. Seed X', we show the mean and stddev. of ACC and BWT, and the Rank averaged over the RL seeds for the corresponding method.

| Method | Test Env. Seed 0 ACC (%) | BWT (%) | Rank | Test Env. Seed 1 ACC (%) | BWT (%) | Rank |
|---|---|---|---|---|---|---|
| Random | 94.33 ± 3.03 | -6.15 ± 3.78 | 5.8 | 92.26 ± 3.84 | -5.96 ± 4.78 | 3.8 |
| ETS | 97.10 ± 0.00 | -2.70 ± 0.00 | 6.0 | 94.10 ± 0.00 | -3.59 ± 0.00 | 2.6 |
| Heur-GD | 97.90 ± 0.00 | -1.59 ± 0.00 | 1.6 | 95.01 ± 0.00 | -2.69 ± 0.00 | 1.4 |
| Heur-LD | 97.59 ± 0.00 | -1.99 ± 0.00 | 2.8 | 90.03 ± 0.00 | -8.90 ± 0.00 | 5.6 |
| Heur-AT | 97.41 ± 0.00 | -2.21 ± 0.00 | 5.0 | 94.09 ± 0.00 | -3.78 ± 0.00 | 3.6 |
| DQN | 97.27 ± 0.71 | -2.50 ± 0.89 | 4.0 | 92.59 ± 2.87 | -5.60 ± 3.57 | 4.0 |
| A2C | 97.71 ± 0.24 | -2.43 ± 0.30 | 2.8 | 80.52 ± 2.26 | -21.17 ± 2.82 | 7.0 |

| Method | Test Env. Seed 2 ACC (%) | BWT (%) | Rank | Test Env. Seed 3 ACC (%) | BWT (%) | Rank |
|---|---|---|---|---|---|---|
| Random | 93.74 ± 2.97 | -6.36 ± 3.71 | 5.2 | 94.12 ± 4.14 | -6.99 ± 5.16 | 3.7 |
| ETS | 86.72 ± 0.00 | -15.17 ± 0.00 | 7.0 | 89.44 ± 0.00 | -12.86 ± 0.00 | 6.6 |
| Heur-GD | 97.41 ± 0.00 | -1.87 ± 0.00 | 2.2 | 96.69 ± 0.00 | -3.68 ± 0.00 | 3.0 |
| Heur-LD | 97.30 ± 0.00 | -1.91 ± 0.00 | 3.2 | 90.61 ± 0.00 | -11.26 ± 0.00 | 5.2 |
| Heur-AT | 97.65 ± 0.00 | -1.58 ± 0.00 | 1.0 | 99.40 ± 0.00 | -0.26 ± 0.00 | 1.1 |
| DQN | 96.37 ± 0.84 | -3.17 ± 1.04 | 3.6 | 96.31 ± 2.98 | -4.16 ± 3.70 | 3.0 |
| A2C | 91.51 ± 1.43 | -7.20 ± 1.79 | 5.8 | 88.09 ± 5.91 | -11.21 ± 7.38 | 5.4 |

| Method | Test Env. Seed 4 ACC (%) | BWT (%) | Rank | Test Env. Seed 5 ACC (%) | BWT (%) | Rank |
|---|---|---|---|---|---|---|
| Random | 83.64 ± 5.22 | -16.86 ± 6.48 | 5.4 | 89.76 ± 2.46 | -7.91 ± 3.06 | 4.8 |
| ETS | 87.07 ± 0.00 | -12.61 ± 0.00 | 4.2 | 91.53 ± 0.00 | -5.61 ± 0.00 | 2.4 |
| Heur-GD | 91.29 ± 0.00 | -7.30 ± 0.00 | 2.2 | 90.33 ± 0.00 | -7.25 ± 0.00 | 4.6 |
| Heur-LD | 86.75 ± 0.00 | -12.87 ± 0.00 | 5.2 | 88.01 ± 0.00 | -10.02 ± 0.00 | 6.0 |
| Heur-AT | 83.28 ± 0.00 | -17.02 ± 0.00 | 6.4 | 90.53 ± 0.00 | -6.81 ± 0.00 | 3.6 |
| DQN | 88.26 ± 3.83 | -10.93 ± 4.79 | 3.6 | 88.45 ± 1.72 | -9.43 ± 2.16 | 5.6 |
| A2C | 96.56 ± 0.36 | -2.89 ± 0.45 | 1.0 | 97.70 ± 0.02 | -1.96 ± 0.02 | 1.0 |

| Method | Test Env. Seed 6 ACC (%) | BWT (%) | Rank | Test Env. Seed 7 ACC (%) | BWT (%) | Rank |
|---|---|---|---|---|---|---|
| Random | 95.34 ± 0.74 | -2.57 ± 1.01 | 2.0 | 94.40 ± 0.77 | -3.53 ± 0.94 | 3.6 |
| ETS | 95.48 ± 0.00 | -2.50 ± 0.00 | 2.2 | 95.31 ± 0.00 | -2.21 ± 0.00 | 2.2 |
| Heur-GD | 90.01 ± 0.00 | -9.09 ± 0.00 | 4.2 | 91.76 ± 0.00 | -6.84 ± 0.00 | 6.2 |
| Heur-LD | 85.44 ± 0.00 | -14.96 ± 0.00 | 5.2 | 95.14 ± 0.00 | -2.64 ± 0.00 | 3.2 |
| Heur-AT | 76.02 ± 0.00 | -26.58 ± 0.00 | 6.8 | 96.78 ± 0.00 | -0.67 ± 0.00 | 1.0 |
| DQN | 83.01 ± 6.53 | -18.05 ± 8.16 | 5.8 | 91.47 ± 1.84 | -7.28 ± 2.30 | 6.0 |
| A2C | 95.30 ± 0.47 | -2.22 ± 0.66 | 1.8 | 90.88 ± 2.58 | -10.15 ± 3.26 | 5.8 |

| Method | Test Env. Seed 8 ACC (%) | BWT (%) | Rank | Test Env. Seed 9 ACC (%) | BWT (%) | Rank |
|---|---|---|---|---|---|---|
| Random | 93.15 ± 4.08 | -7.22 ± 5.09 | 3.2 | 94.81 ± 3.66 | -5.14 ± 4.51 | 3.0 |
| ETS | 94.80 ± 0.00 | -5.18 ± 0.00 | 3.2 | 96.71 ± 0.00 | -2.76 ± 0.00 | 2.0 |
| Heur-GD | 95.00 ± 0.00 | -4.83 ± 0.00 | 2.2 | 95.27 ± 0.00 | -4.75 ± 0.00 | 3.2 |
| Heur-LD | 87.86 ± 0.00 | -13.66 ± 0.00 | 6.6 | 90.02 ± 0.00 | -11.16 ± 0.00 | 6.6 |
| Heur-AT | 93.93 ± 0.00 | -6.07 ± 0.00 | 4.2 | 90.29 ± 0.00 | -10.82 ± 0.00 | 5.6 |
| DQN | 93.38 ± 3.67 | -6.82 ± 4.61 | 3.2 | 95.33 ± 1.92 | -4.53 ± 2.40 | 2.4 |
| A2C | 91.66 ± 2.01 | -5.46 ± 2.50 | 5.4 | 90.66 ± 2.27 | -10.16 ± 2.85 | 5.2 |

**D.4   Task Splits in Test Environments in Policy Generalization Experiments**

Here, we provide the task splits of the test environments used in the policy generalization experiments in Section 4.2. We evaluated all methods using 10 test environments in all experiments. The test environments in the New Task Order experiments were generated with seeds 10-19. We show the task splits for the Split MNIST, Split FashionMNIST, and Split CIFAR-10 environments in Table 44, 45, and 46 respectively. The test environments in the New Dataset experiments were generated with seeds 0-9. We show the task splits for the Split notMNIST and Split FashionMNIST environments in Table 47 and 48 respectively.

Table 44: Task splits with their corresponding seed for test environments of Split MNIST datasets in the **New Task Orders** experiments in Section 4.2.

| Seed | Task 1 | Task 2 | Task 3 | Task 4 | Task 5 |
|------|--------|--------|--------|--------|--------|
| 10 | 8, 2 | 5, 6 | 3, 1 | 0, 7 | 4, 9 |
| 11 | 7, 8 | 2, 6 | 4, 5 | 1, 3 | 0, 9 |
| 12 | 5, 8 | 7, 0 | 4, 9 | 3, 2 | 1, 6 |
| 13 | 3, 5 | 6, 1 | 4, 7 | 8, 9 | 0, 2 |
| 14 | 3, 9 | 0, 5 | 4, 2 | 1, 7 | 6, 8 |
| 15 | 2, 6 | 1, 3 | 7, 0 | 9, 4 | 5, 8 |
| 16 | 6, 2 | 0, 7 | 8, 4 | 3, 1 | 5, 9 |
| 17 | 7, 2 | 5, 3 | 4, 0 | 9, 8 | 6, 1 |
| 18 | 7, 9 | 0, 4 | 2, 1 | 6, 5 | 8, 3 |
| 19 | 1, 7 | 9, 6 | 8, 4 | 3, 0 | 2, 5 |

Table 45: Task splits with their corresponding seed for test environments of Split FashionMNIST datasets in the **New Task Orders** experiments in Section 4.2.

| Seed | Task 1 | Task 2 | Task 3 | Task 4 | Task 5 |
|------|--------|--------|--------|--------|--------|
| 10 | Bag, Pullover | Sandal, Shirt | Dress, Trouser | T-shirt/top, Sneaker | Coat, Ankle boot |
| 11 | Sneaker, Bag | Pullover, Shirt | Coat, Sandal | Trouser, Dress | T-shirt/top, Ankle boot |
| 12 | Sandal, Bag | Sneaker, T-shirt/top | Coat, Ankle boot | Dress, Pullover | Trouser, Shirt |
| 13 | Dress, Sandal | Shirt, Trouser | Coat, Sneaker | Bag, Ankle boot | T-shirt/top, Pullover |
| 14 | Dress, Ankle boot | T-shirt/top, Sandal | Coat, Pullover | Trouser, Sneaker | Shirt, Bag |
| 15 | Pullover, Shirt | Trouser, Dress | Sneaker, T-shirt/top | Ankle boot, Coat | Sandal, Bag |
| 16 | Shirt, Pullover | T-shirt/top, Sneaker | Bag, Coat | Dress, Trouser | Sandal, Ankle boot |
| 17 | Sneaker, Pullover | Sandal, Dress | Coat, T-shirt/top | Ankle boot, Bag | Shirt, Trouser |
| 18 | Sneaker, Ankle boot | T-shirt/top, Coat | Pullover, Trouser | Shirt, Sandal | Bag, Dress |
| 19 | Trouser, Sneaker | Ankle boot, Shirt | Bag, Coat | Dress, T-shirt/top | Pullover, Sandal |

Table 46: Task splits with their corresponding seed for test environments of Split CIFAR-10 datasets in the **New Task Orders** experiments in Section 4.2.

| Seed | Task 1 | Task 2 | Task 3 | Task 4 | Task 5 |
|------|--------|--------|--------|--------|--------|
| 10 | Ship, Bird | Dog, Frog | Cat, Automobile | Airplane, Horse | Deer, Truck |
| 11 | Horse, Ship | Bird, Frog | Deer, Dog | Automobile, Cat | Airplane, Truck |
| 12 | Dog, Ship | Horse, Airplane | Deer, Truck | Cat, Bird | Automobile, Frog |
| 13 | Cat, Dog | Frog, Automobile | Deer, Horse | Ship, Truck | Airplane, Bird |
| 14 | Cat, Truck | Airplane, Dog | Deer, Bird | Automobile, Horse | Frog, Ship |
| 15 | Bird, Frog | Automobile, Cat | Horse, Airplane | Truck, Deer | Dog, Ship |
| 16 | Frog, Bird | Airplane, Horse | Ship, Deer | Cat, Automobile | Dog, Truck |
| 17 | Horse, Bird | Dog, Cat | Deer, Airplane | Truck, Ship | Frog, Automobile |
| 18 | Horse, Truck | Airplane, Deer | Bird, Automobile | Frog, Dog | Ship, Cat |
| 19 | Automobile, Horse | Truck, Frog | Ship, Deer | Cat, Airplane | Bird, Dog |

Table 47: Task splits with their corresponding seed for test environments of Split notMNIST datasets in the **New Dataset** experiments in Section 4.2.

| Seed | Task 1 | Task 2 | Task 3 | Task 4 | Task 5 |
|------|--------|--------|--------|--------|--------|
| 0 | A, B | C, D | E, F | G, H | I, J |
| 1 | C, J | G, E | A, D | B, H | I, F |
| 2 | E, B | F, A | H, C | D, G | J, I |
| 3 | F, E | B, C | J, G | H, A | D, I |
| 4 | D, I | E, J | C, G | A, B | F, H |
| 5 | J, F | C, E | H, B | A, I | G, D |
| 6 | I, B | H, A | G, F | C, E | D, J |
| 7 | I, F | A, C | B, J | H, D | G, E |
| 8 | I, G | J, A | C, F | H, B | E, D |
| 9 | I, E | H, C | B, J | D, A | G, F |

Table 48: Task splits with their corresponding seed for test environments of Split FashionMNIST datasets in the **New Dataset** experiments in Section 4.2.

| Seed | Task 1 | Task 2 | Task 3 | Task 4 | Task 5 |
|------|--------|--------|--------|--------|--------|
| 0 | T-shirt/top, Trouser | Pullover, Dress | Coat, Sandal | Shirt, Sneaker | Bag, Ankle boot |
| 1 | Pullover, Ankle boot | Shirt, Coat | T-shirt/top, Dress | Trouser, Sneaker | Bag, Sandal |
| 2 | Coat, Trouser | Sandal, T-shirt/top | Sneaker, Pullover | Dress, Shirt | Ankle boot, Bag |
| 3 | Sandal, Coat | Trouser, Pullover | Ankle boot, Shirt | Sneaker, T-shirt/top | Dress, Bag |
| 4 | Dress, Bag | Coat, Ankle boot | Pullover, Shirt | T-shirt/top, Trouser | Sandal, Sneaker |
| 5 | Ankle boot, Sandal | Pullover, Coat | Sneaker, Trouser | T-shirt/top, Bag | Shirt, Dress |
| 6 | Bag, Trouser | Sneaker, T-shirt/top | Shirt, Sandal | Pullover, Coat | Dress, Ankle boot |
| 7 | Bag, Sandal | T-shirt/top, Pullover | Trouser, Ankle boot | Sneaker, Dress | Shirt, Coat |
| 8 | Bag, Shirt | Ankle boot, T-shirt/top | Pullover, Sandal | Sneaker, Trouser | Coat, Dress |
| 9 | Bag, Coat | Sneaker, Pullover | Trouser, Ankle boot | Dress, T-shirt/top | Shirt, Sandal |

Table 49: The threshold parameter $\tau$ used for the ETS combined with heuristic scheduling baselines. The search range is $\tau \in \{0.90, 0.95, 0.999\}$ for all methods and we use 10 environments for validation to select the threshold parameter used in the test environments.

| | New Task Order | | | | New Dataset | |
|--------|---------|---------------|------------|-----------|------------|---------------|
| Method | S-MNIST | S-FashionMNIST | S-notMNIST | S-CIFAR-10 | S-notMNIST | S-FashionMNIST |
| ETS+Heur-GD | 0.999 | 0.999 | 0.999 | 0.999 | 0.95 | 0.999 |
| ETS+Heur-LD | 0.999 | 0.90 | 0.95 | 0.90 | 0.95 | 0.90 |
| ETS+Heur-AT | 0.90 | 0.90 | 0.90 | 0.95 | 0.999 | 0.95 |

### D.5 Results with Equal Task Scheduling Combined with Heuristic

In this section, we present the rankings and performance metrics of the RL experiments in Section 4.2 with additional baselines that combines Equal Task Scheduling (ETS) with the heuristic rules. These scheduling baselines replay the old tasks equally at every task, like ETS, but also checks whether their validation accuracy is below a threshold according to the heuristic rule to replay harder tasks more. We perform the same experiments as in Section 4.2 for ETS combined with Heur-GD, Heur-LD, and Heur-AT, and assess their generalization capability to new CL environments. The metrics for the RL policies (DQN and A2C) and the other baselines are the same as in Table 4 and 51 for the **New Task Order** and **New Dataset** experiments respectively. In the reported results, we indicate with red, orange, and yellow highlight the 1st, 2nd, and 3rd-best performing scheduling method based on its mean value on the metric for each dataset. Table 49 shows the selected threshold parameters for each method.

Table 50 shows the performance metrics on the **New Task Order** experiment. We observe that the RL policies (DQN and A2C) or the other baselines perform better than the ETS+Heur baselines on all datasets except Split FashionMNIST. In Table 51, we show the performance metrics on the **New Dataset** experiment. We see that the RL policies perform competitively against the ETS+Heur baselines on Split notMNIST, but are outperformed by ETS-Heur-GD on Split FashionMNIST which was also observed in Section 4.2. Since the results are similar to Section 4.2, this further shows the benefits of learning replay scheduling policies that selectively can turn on and off replaying old tasks to focus more on remembering hard tasks. Potentially, ETS combined with a carefully tuned heuristic would obtain more clear performance improvements against ETS and the heuristics alone for larger memory sizes than $M = 10$ that is used here.

Table 50: Performance comparison measured with average ranking (Rank), ACC (%), BWT (%) between the scheduling policies, including ETS combined with the heuristics, in the **New Task Order** experiment. Our learned policies using DQN and A2C performs competitively against the fixed policies (Random and ETS), the heuristics alone and combined with ETS across the 5-task datasets.

| Method | Split MNIST | | | Split FashionMNIST | | | Split notMNIST | | | Split CIFAR10 | | |
|---|---|---|---|---|---|---|---|---|---|---|---|---|
| | Rank (↓) | ACC (↑) | BWT (↑) | Rank (↓) | ACC (↑) | BWT (↑) | Rank (↓) | ACC (↑) | BWT (↑) | Rank (↓) | ACC (↑) | BWT (↑) |
| Random | 5.46 | 91.8 ± 4.7 | -9.7 ± 5.9 | 5.26 | 93.9 ± 3.8 | -5.6 ± 4.5 | 4.56 | 91.7 ± 2.8 | -4.8 ± 4.0 | 6.67 | 81.6 ± 6.1 | -17.2 ± 6.8 |
| ETS | 5.32 | 91.8 ± 5.0 | -9.7 ± 6.3 | 6.74 | 93.5 ± 3.0 | -6.1 ± 3.3 | 5.94 | 90.4 ± 3.1 | -6.3 ± 4.4 | 6.98 | 81.0 ± 5.9 | -18.1 ± 6.3 |
| Heur-GD | 5.73 | 91.3 ± 4.3 | -10.3 ± 5.4 | 6.03 | 91.8 ± 7.9 | -8.2 ± 9.0 | 4.04 | 92.3 ± 1.5 | -4.1 ± 1.7 | 5.33 | 83.2 ± 5.0 | -15.4 ± 5.0 |
| Heur-LD | 6.17 | 91.0 ± 4.1 | -10.7 ± 5.2 | 5.13 | 93.8 ± 5.0 | -5.7 ± 5.5 | 4.86 | 91.7 ± 2.0 | -4.8 ± 1.9 | 4.83 | 83.5 ± 4.2 | -15.1 ± 4.2 |
| Heur-AT | 5.58 | 91.5 ± 4.0 | -10.1 ± 5.1 | 5.91 | 92.9 ± 4.9 | -6.8 ± 5.1 | 7.15 | 89.7 ± 2.9 | -7.1 ± 3.7 | 4.43 | 83.7 ± 4.3 | -14.8 ± 4.3 |
| ETS+Heur-GD | 5.82 | 91.9 ± 4.0 | -9.6 ± 5.0 | 4.38 | 94.6 ± 3.0 | -4.7 ± 2.9 | 6.4 | 90.1 ± 2.7 | -6.8 ± 3.7 | 6.73 | 82.7 ± 4.2 | -16.1 ± 4.2 |
| ETS+Heur-LD | 5.62 | 92.1 ± 4.0 | -9.3 ± 5.0 | 5.2 | 95.0 ± 2.7 | -4.2 ± 2.6 | 6.72 | 90.4 ± 3.2 | -6.2 ± 3.9 | 5.36 | 83.1 ± 3.2 | -15.5 ± 3.2 |
| ETS+Heur-AT | 6.88 | 90.8 ± 5.0 | -10.9 ± 6.2 | 5.93 | 94.6 ± 2.8 | -4.7 ± 2.5 | 6.93 | 90.2 ± 3.3 | -6.6 ± 3.8 | 5.83 | 83.0 ± 4.1 | -15.6 ± 4.4 |
| DQN (Ours) | 4.46 | 93.0 ± 2.7 | -8.2 ± 3.4 | 5.14 | 94.2 ± 3.8 | -5.2 ± 3.9 | 4.19 | 91.9 ± 2.5 | -4.6 ± 2.8 | 5.31 | 83.0 ± 4.3 | -15.6 ± 4.3 |
| A2C (Ours) | 3.96 | 93.1 ± 3.7 | -8.1 ± 4.6 | 5.28 | 94.8 ± 3.3 | -4.5 ± 3.0 | 4.21 | 92.1 ± 1.8 | -4.3 ± 2.4 | 3.53 | 83.9 ± 3.8 | -14.5 ± 3.5 |

Table 51: Performance comparison with average ranking (Rank), ACC, and BWT between our RL policies and the baselines, including ETS combined with the heuristics, on the **New Dataset** experiment. Our policies generalize well on Split notMNIST, but is outperformed by ETS and Heur-GD on Split FashionMNIST.

| Method | Split notMNIST | | | Split FashionMNIST | | |
|---|---|---|---|---|---|---|
| | Rank (↓) | ACC (↑) | BWT (↑) | Rank (↓) | ACC (↑) | BWT (↑) |
| Random | 5.5 | 91.4 ± 2.7 | -5.5 ± 3.3 | 5.41 | 92.6 ± 4.7 | -6.9 ± 5.6 |
| ETS | 6.16 | 91.5 ± 1.7 | -5.5 ± 1.9 | 4.74 | 92.8 ± 3.7 | -6.5 ± 4.8 |
| Heur-GD | 6.91 | 89.8 ± 4.8 | -7.6 ± 6.5 | 3.78 | 94.1 ± 2.8 | -5.0 ± 2.4 |
| Heur-LD | 7.46 | 89.1 ± 4.7 | -8.4 ± 6.3 | 6.86 | 90.9 ± 4.1 | -8.9 ± 4.7 |
| Heur-AT | 6.29 | 90.2 ± 4.9 | -7.0 ± 6.6 | 4.83 | 91.9 ± 7.0 | -7.6 ± 8.0 |
| ETS+Heur-GD | 4.78 | 91.3 ± 4.3 | -5.6 ± 5.9 | 5.00 | 93.4 ± 3.1 | -5.7 ± 3.4 |
| ETS+Heur-LD | 4.44 | 91.3 ± 4.2 | -5.6 ± 5.7 | 6.75 | 92.4 ± 3.1 | -7.0 ± 3.0 |
| ETS+Heur-AT | 4.26 | 91.2 ± 4.5 | -5.7 ± 6.1 | 6.27 | 92.5 ± 4.1 | -6.9 ± 4.3 |
| DQN (Ours) | 5.06 | 91.7 ± 3.2 | -5.2 ± 4.5 | 5.72 | 92.2 ± 5.3 | -7.2 ± 5.9 |
| A2C (Ours) | 4.14 | 92.6 ± 1.6 | -4.1 ± 2.3 | 5.64 | 92.1 ± 5.5 | -7.5 ± 6.4 |

