# OpenReview forum: "Learn the Time to Learn: Replay Scheduling in Continual Learning"
_TMLR — Accepted by TMLR_

### Review · Reviewer_tXM7 · 2023-07-17

**Summary Of Contributions:**

This paper focuses on replay-based continual learning ans specifically when to replay previous data. Motivated by techniques for human learning, the authors suggest that forgetting can be mitigated by task replay schedules (learning the time to replay). Monte-Carlo-Tree-Search (MCTS) and Reinforcement Learning (RL) are then used to learn replay schedules. The authors empirically demonstrate improved performance over baselines for MCTS, thus highlighting the importance of replay schedule. Finally, the paper presents an empirical study for generalization to new CL scenarios via RL.

**Audience:**

Yes

**Broader Impact Concerns:**

N.A

**Claims And Evidence:**

No

**Requested Changes:**

I would like to request for the following changes that focuses more the RL formulation to address the concerns raised above:

1. How can the RL method be extended to addresses task-free and online setting.
2. More experiments on generalization on novel datasets.

The key concern I have is that while the manuscript clearly demonstrates replay schedule as an important direction, it does not offer a viable approach forward with the proposed RL approach.

**Strengths And Weaknesses:**

Strength:

1. The paper is well motivated. In particular, the empirical evaluation on MCTS illustrates the efficacy of altering replay schedules.
2. The paper is well written, and easy to follow.
3. The MCTS experiments validates the need for replay scheduling.
4. The connection to human learning schedules such as spaced repetition is interesting (e.g. the visualization of replay schedules in Fig. 3)

Weakness:
1. The setting considered in the paper is too narrow. Both MCTS and RL are mostly applied to a offline task-based continual learning setting, which allows the discretization of action space and the construction of states for RL. However, it is unclear that how the present method may be applied to online or task-free settings, both of which are generally regarded as more realistic in the literature. While there is some experiments on class-IL, the details on class-IL setting seems to be missing and the performance of all methods seem very poor in absolute terms, even though MCTS outperforms the baseline.

2. The proposed RL method does not appear to offer a practical solution for learning replay schedule. The generalizationt to new task orders is too narrow and could only operate on the same dataset. The generalization performance to novel dataset is mixed at the moment, with only improvement on the simpler Mnist-variant.

    It appears unclear that whether the proposed RL method could extend to online or task-free setting.

    The need to compute task accuracy for RL states may be computationally expensive, and may have high variance when the validation set is small. This could also negatively affect learning of the replay schedule.

    Training a generalizable RL policy may be prohibitively expensive, considering the vast combination of datasets and architectures that the policy needs to be trained on, in addition to the cost of RL training.

---

> ### Author Response · Authors · 2023-08-09
> **Rebuttal Reply 1/2: Weaknesses on extension to task-free settings and the proposed RL method**
>
> We thank the reviewer for the constructive feedback and comments.
>
> > **Weakness 1: Task-free and Online settings and Class-IL setup**
>
> **"...unclear that how the present method may be applied to online or task-free settings..."**
>
> We agree that learning the time to learn in online and task-free continual learning (CL) settings is currently an open research challenge. We mention this under Limitations and Future Work (Section 5) by referring to "unknown task changes" as the task-free CL setting. We will clarify in the revised manuscript that the challenge of applying replay scheduling to task-free and online settings remains an open question.
>
> Our suggestion to extend replay scheduling to task-free CL settings would be to compose the RL states with computed class accuracies rather than task accuracies to select which classes to replay at different times. Extending further to an online setting would require continual evaluation [1] of the states, probably combined with a continual collection of validation datasets.
> Another interesting direction would be to explore online fine-tuning RL policies where (similar as we do) the base policy is learned from offline training [2].
>
>
>
>  **"...the details on class-IL setting seems to be missing..."**
>
> We clarify the experimental setting of our Class Incremental Learning (Class-IL) experiments for MCTS:
> The replay memory is filled with at least 1 sample per class at every task, otherwise
> tasks that are not replayed would be fully forgotten, i.e., accuracy = 0.0\%. MCTS and the baselines then schedule which tasks to replay out of the remaining samples $M_{rest} = M - t \cdot n_c$,
> where $t$ is the task number and $n_c$ is the number of classes per task in the dataset. All other settings are the same as for Task-IL experiments, i.e., same learning rates, number of epochs, batch size, network architectures etc. We will clarify the replay memory setting in the Class-IL experiments in the revised manuscript.
>
>
> **"... class-IL setting... performance of all methods seem very poor in absolute terms..."**
>
> We pick the DER paper [3] for comparing the Class-IL results from their reported Experience Replay (ER) baseline against our Equal Task Scheduling (ETS) baseline, which are similar approaches, on the Split MNIST dataset. For memory size $M=200$, we observe that ER vs ETS obtains accuracy 80.43 +/- 1.89 [3, Table F.1] vs 77.73 +/- 1.31 (Table 21). These numbers are very close, but not exactly the same probably due to different CL hyperparameters (learning rate, batch size, etc.) and other replay settings.
>
> We ensure fair comparisons in our paper. Furthermore, we show that replay scheduling can be combined with any memory selection (Table 1) and replay method (Table 2), e.g., DER [3], to potentially obtain better CL performance.
>
>
> > **Weakness 2: Proposed RL method**
>
> **"The proposed RL method does not appear to offer a practical solution... generalization to new task orders is too narrow... generalization performance to novel dataset is mixed..."** and **"Training a generalizable RL policy may be prohibitively expensive... "**
>
>
> Using RL for learning the replay scheduling policy indeed comes with challenges, such as robustness to domain shifts.
> However, tackling inherited challenges in RL is out of scope for this work, but we believe that advances in RL research will also benefit our proposed framework.
> We intend to apply more complex RL methods in the future.
>
>
>
>
> **"...unclear that whether the proposed RL method could extend to online or task-free setting."**
>
> See our reply above under Weakness 1.
>
>
>  **"...need to compute task accuracy for RL states may be computationally expensive... high variance when the validation set is small..."**
>
> We agree with the reviewer that our assumption of access to validation sets at test time is a limitation. We will emphasize on this assumption in the revised version of the manuscript.
>
>
> **References:**
>
> [1] De Lange, Matthias, Gido van de Ven, and Tinne Tuytelaars. "Continual evaluation for lifelong learning: Identifying the stability gap." ICLR 2023.
>
> [2] Nair, Ashvin, Abhishek Gupta, Murtaza Dalal, and Sergey Levine. "Awac: Accelerating online reinforcement learning with offline datasets." arXiv preprint arXiv:2006.09359 (2020).
>
> [3] Buzzega, Pietro, Matteo Boschini, Angelo Porrello, Davide Abati, and Simone Calderara. "Dark experience for general continual learning: a strong, simple baseline." Advances in neural information processing systems 33 (2020): 15920-15930.

---

> > ### Author Response · Authors · 2023-08-09
> > **Rebuttal Reply 2/2: Requested Changes on task-free settings and more RL experiments in novel dataset setting**
> >
> > > **Requested Change 1: "How can the RL method be extended to addresses task-free and online setting."**
> >
> > Extending to task-free CL settings could be done by composing the RL states with computed class accuracies instead of task accuracies to select which classes to replay at different times. Extending further to an online setting would require continually computing the states and could be combined with a continual collection of validation datasets. We will add these suggestions to the revised manuscript as future work.
> >
> >
> > > **Requested Change 2: "More experiments on generalization on novel datasets."**
> >
> > Generalization in RL is an active research field on its own [4]. Tackling inherited RL challenges, such as robustness to domain shift, is orthogonal to this work, but we will apply more advanced RL methods
> > to our framework for learning the replay scheduling policies in the future.
> >
> >
> > **References:**
> >
> > [4] Kirk, Robert, Amy Zhang, Edward Grefenstette, and Tim Rocktäschel. "A survey of zero-shot generalisation in deep reinforcement learning." Journal of Artificial Intelligence Research 76 (2023): 201-264.

---

> > > ### Comment · Reviewer_tXM7 · 2023-08-11
> > > **Reply to authors' rebuttal**
> > >
> > > I thank the authors for the extensive clarification. Part of my concerns are addressed by the authors.
> > >
> > > The current manuscript has clear strengths and limitations. I think the strengths outweighs the limitations and would be interesting to the community as a new aveneue of research for continual learning. I thus support the paper's acceptance.
> > >
> > > At the same time, please do clearly highlight the limitations in the final manuscript as it does not offer a fully practical and viable approach for "learning the time to learn".

---

> > > > ### Author Response · Authors · 2023-08-11
> > > > **Authors to Reviewer tXM7: Thank you**
> > > >
> > > > Dear reviewer tXM7,
> > > >
> > > > We are happy to hear that you support accepting our paper to TMLR. We will update the limitations part accordingly in the camera-ready version. Thank you for your constructive and thoughtful feedback for improving our paper.

---

### Review · Reviewer_bt5V · 2023-07-18

**Summary Of Contributions:**

The paper highlights that the bottleneck of continual learning (CL) in real-world scenarios is mostly on computation rather than storage. Previous works on CL have focused on constructing replay memory for each task but have overlooked the efficient utilization of these memories. In contrast, this paper proposes to learn how to balance the memory across tasks using techniques such as MCTS or RL algorithms like DQN and A2C. The experimental results demonstrate that the proposed method outperforms intuitive baselines such as equal task schedule (ETS) or naively focusing on hard tasks (Heur-GD).

**Audience:**

Yes

**Broader Impact Concerns:**

Well done.

**Claims And Evidence:**

Yes

**Requested Changes:**

Rebuttal on [W1-2].

**Strengths And Weaknesses:**

**Strength**


[S1] Clear motivation\
The paper addresses an overlooked problem in continual learning (CL): how to balance the replay memory, while most prior works focus on how to construct these memories.

[S2] Reasonable baselines\
The paper compares the proposed method with reasonable baselines, including ETS and Heur-GD.

[S3] Extensive experiments\
The paper demonstrates experiments on various benchmarks, including MNIST, CIFAR, and mini-ImageNet.

[S4] Visualization of replay schedule\
The paper provides the learned replay schedule, not only the performance (ACC, BWT), which offers insights into how the proposed method works.

[S5] Limitations and broader impacts\
The paper appropriately discusses its limitations and broader impacts.


**Weakness**

[W1] MCTS vs. RL approaches?\
The paper suggests two ways to balance the replay memory, MCTS and RL approaches. However, it only shows the generalization to new tasks experiments for RL approaches. Why not compare MCTS and DQN/A2C under the same setups?

[W2] Maybe a sledgehammer to crack a nut\
The RL approach gets the task accuracies [A_{t,1},...,A_{t,t}] as a state and provides how to balance them. This problem may be solved by a properly designed heuristic, which is an interpolation between ETS and Heur-GD, assigning a higher portion to hard tasks but not being too concentrated. Some well-designed heuristic can be on par with the expensive MCTS and RL methods.

---

> ### Author Response · Authors · 2023-08-09
> **Rebuttal Reply: Added results with suggested baseline ETS combined with Heuristic scheduling**
>
> We thank the reviewer for the valuable feedback and suggestions.
>
>
> > **[W1]: "Why not compare MCTS and DQN/A2C under the same setups?"**
>
> We believe that there has been a misunderstanding. Rather than suggesting MCTS as a method for balancing the task proportion in the replay memory, we use MCTS as an exemplar method to show the benefits of replay scheduling in continual learning (CL). Since MCTS requires multiple episodes of CL training in any new CL scenario and dataset, we are prohibited from using MCTS in realistic CL settings. Therefore, we move forward to learning general replay scheduling policies with our RL framework that actually can be applied to new CL scenarios.
>
>
> > **[W2]: "... a properly designed heuristic, which is an interpolation between ETS and Heur-GD... can be on par with the expensive MCTS and RL methods".**
>
>
> We implemented the suggested baseline, which combines Equal Task Scheduling (ETS) with a heuristic rule, such as Heur-GD.
> This scheduling baseline replays the old tasks equally at every task, and also checks whether their validation accuracy is below a threshold according to the heuristic rule to replay harder tasks more.
>
> We compare ETS+Heur-GD against MCTS in the Task Incremental Learning (Task-IL) setting with same setup as in Table 1 (Section 4.1). In the table below, we report the ACC and BWT metrics on 5-task Split MNIST and the 20-task Split CIFAR-100. While ETS+Heur-GD outperforms ETS on both datasets, it is outperformed by both Heur-GD and MCTS.
> This shows the benefits of using replay scheduling policies that selectively can turn on/off replaying old tasks to focus more on remembering hard tasks.
>
>
> |            | **Split MNIST**  |        | **Split CIFAR-100** |             |
> |---------------|:----------------:|:----------------:|:-------------------:|:-----------------:|
> | **Method**      | ACC              | BWT              | ACC                 | BWT               |
> | **Random**      | 94.91 $\pm$ 2.52 | -6.13 $\pm$ 3.16 | 53.76 $\pm$ 1.80    | -35.11 $\pm$ 1.93 |
> | **ETS**         | 94.02 $\pm$ 4.25 | -7.22 $\pm$ 5.33 | 47.70 $\pm$ 2.16    | -41.69 $\pm$ 2.37 |
> | **Heur-GD**     | 96.02 $\pm$ 2.32 | -4.64 $\pm$ 2.90 | 57.31 $\pm$ 1.21    | -30.76 $\pm$ 1.45 |
> | **ETS+Heur-GD** | 95.19 $\pm$ 2.10 | -5.70 $\pm$ 2.64 | 51.00 $\pm$ 0.76    | -37.33 $\pm$ 0.58 |
> | **MCTS**        | 97.93 $\pm$ 0.56 | -2.27 $\pm$ 0.71 | 56.60 $\pm$ 1.13    | -31.39 $\pm$ 1.11 |
>
> We also present the rankings, ACC, and BWT for ETS combined with the heuristics from the RL experiment (Section 4.2) in the New Task Order setting on Split MNIST in the table below. We observe that the RL policies (DQN and A2C) gets better rankings than the ETS+Heuristic baselines, which further demonstrates the benefits of learning the scheduling policy.
> We will add results with the ETS+Heuristic for all datasets in the MCTS and RL experiments in the camera-ready version of our paper.
>
> |             | **Split MNIST** |            |             |
> |---------------|:---------------:|:--------------:|:---------------:|
> | **Method**      | Rank            | ACC            | BWT             |
> | **Random**      | 5.46            | 91.8 $\pm$ 4.7 | -9.7 $\pm$ 5.9  |
> | **ETS**         | 5.32            | 91.8 $\pm$ 5.0 | -9.7 $\pm$ 6.3  |
> | **Heur-GD**     | 5.73            | 91.3 $\pm$ 4.3 | -10.3 $\pm$ 5.4 |
> | **Heur-LD**     | 6.17            | 91.0 $\pm$ 4.1 | -10.7 $\pm$ 5.2 |
> | **Heur-AT**     | 5.58            | 91.5 $\pm$ 4.0 | -10.1 $\pm$ 5.1 |
> | **ETS+Heur-GD** | 5.82            | 91.9 $\pm$ 4.0 | -9.6 $\pm$ 5.0  |
> | **ETS+Heur-LD** | 5.62            | 92.1 $\pm$ 4.0 | -9.3 $\pm$ 5.0  |
> | **ETS+Heur-AT** | 6.88            | 90.8 $\pm$ 5.0 | -10.9 $\pm$ 6.2 |
> | **DQN**         | 4.46            | 93.0 $\pm$ 2.7 | -8.2 $\pm$ 3.4  |
> | **A2C**         | 3.96            | 93.1 $\pm$ 3.7 | -8.1 $\pm$ 4.6  |

---

> > ### Comment · Reviewer_bt5V · 2023-08-13
> > **Concerns addressed**
> >
> > Thank you for the rebuttal. After reading other reviews and the author's rebuttal, I am also leaning to accept the paper.
> >
> > Here are my post-rebuttal comments:\
> > I agree with QGFH that learning replay schedule gives a new angle to continual learning, which would indeed be a valuable contribution to the community. However, I believe that the concerns raised by tXM7 are valid, specifically regarding the extension to task-free CL and the limited evaluation of the RL' generalizability. While the first concern might fall outside the paper's scope, it would be great if the revised manuscript could address the second concern. This is important because users are expected to apply a pretrained general policy module to their own CL tasks in practical scenarios.
> >
> > Sincerely,\
> > Reviewer bt5V

---

> > > ### Author Response · Authors · 2023-08-14
> > > **Authors to Reviewer bt5V: Thank you and comment on RL generalizability as future work**
> > >
> > > Dear reviewer bt5V,
> > >
> > > Thank you for considering to accept our paper. We are glad to see that we have addressed your concerns.
> > >
> > > > **"...limited evaluation of the RL generalizability..."**
> > >
> > > Tackling challenges inherited in RL, such as robustness to domain shift, is orthogonal to this work as our contributions are to show the benefits of replay scheduling in continual learning. But we will apply more advanced RL methods for learning replay scheduling policies that better generalize in the future.
> > >
> > > Thank you again for your constructive feedback and comments to help us improve the paper.

---

### Review · Reviewer_QGFH · 2023-07-27

**Summary Of Contributions:**

This paper asks the question: “if in a continual learning experiment only a limited amount of data can be replayed, which data should be chosen?” In particular, the paper asks whether it is possible to design a replay schedule that performs better than always dividing the replay budget equally over all tasks seen so far.

In a first set of experiments the paper uses brute force (in the form of Monte Carlo Tree Search, MCTS) to find optimal replay schedules, in order to demonstrate that there exist replay schedules that are better than always equally replaying all past tasks. In a second set of experiments the paper employs reinforcement learning to learn replay schedules, and it is shown that these learned replay schedules generalize to new task orders and even to new datasets. This second set of experiments is motivated as a first step towards creating a method for replay scheduling that could be useful in the “real world”.

**Audience:**

Yes

**Broader Impact Concerns:**

No concerns in this regard.

**Claims And Evidence:**

No

**Requested Changes:**

For me to support acceptance of this paper, the following three issues should be addressed (more details on each issue can be found above):
- The main issue discussed above: in the first set of experiments, because a different optimal replay schedule is selected for each random seed, it is unclear whether the demonstrated benefits are due to replaying specific tasks or replaying a specific set of samples;
- Discuss more clearly that applying replay schedules, which were learned with RL, requires access to validation sets of all past tasks;
- Describe how the replay itself is implemented.

**Strengths And Weaknesses:**

I found this an intriguing paper. In recent years there have been several continual learning studies that explored whether benefits could be gained by deciding what to replay in a smart way. I myself have worked on this as well, and my impression is that generally the conclusion of these studies has been that it is very hard to do better than simply doing balanced, random sampling from all tasks so far. This paper takes an original new angle to this problem, and demonstrates that it is possible to gain benefits by being smart about what to replay.

I think this is an important demonstration that should be of interest to a large group of researchers working on continual learning. On top of that, the paper is pleasant to read, it is written in a clear style, most experimental details are reported and well-documented code is provided.

There is however one main issue, and two smaller ones, that must be addressed before I can support acceptance of this paper.

The main issue is that, in the first set of experiments, my impression is that it is not actually demonstrated that a schedule of replaying *specific tasks* can be beneficial. Rather, I think the current implementation of these experiments merely shows that it can be beneficial to replay a *specific set of past samples*. The reason for this is that to select the optimal replay schedule, the authors use MCTS to select 100 different replay schedules to try, and based on their performance on a validation set the authors then select the best performing one. It seems that the authors do this selection of a replay schedule again for each random seed. For the best performing schedule of each random seed, the performance on the test set is then reported. However, the 100 different replay schedules that are tried out for each random seed do not only differ in which tasks are replayed, they also differ in which specific samples are replayed. Therefore, by selecting the best performing schedule out of a 100 different schedules in this way, it is unclear whether this schedule is the best because of the tasks it chooses to replay or because of the specific samples it happened to select. If it is indeed the case that the authors select a different optimal replay schedule for each random seed (and based on the provided code, it is my impression that this is what is done), I think this is a critical issue that invalidates the current interpretation of these experiments. To fix this, after the presumed optimal replay schedule of tasks has been found with MCTS, the authors should apply that replay schedule with different samples from the selected tasks (i.e., new experiments with new random seeds should be run with the optimal replay schedule selected using a different random seed).

The two smaller issues that need addressing as well:
- In the second set of experiments, because the learned optimal replay schedule from one task sequence is applied to another task sequence, it seems to me that the above issue does not apply. However, there is still the lesser issue that in order to apply the learned optimal replay schedule, at each task switch, access to the validation sets of all past tasks is required. This is an important disadvantage from the perspective of practical applicability. Nevertheless, despite this issue, I still consider these experiments an important proof-of-principle demonstration. However, I think it is important to make the issue of requiring access to the validation sets of all past tasks more clear in the paper, and I think it would be good to include a balanced discussion of this issue.
- Another issue is that I was not able to find how the replay itself is implemented. I understand that for each new task a set of $M$ examples from past tasks are selected to be replayed during training on the new task, but I could not find exactly how these $M$ examples are replayed. Are these $M$ examples simply added to the training data of the new task? Or are these $M$ examples replayed in every training iteration of the new task? If the latter, how are the loss on the replayed samples and the loss on the samples from the current task weighted?

Finally, I noticed a few minor issues that do not prevent me from supporting acceptance of this paper, but addressing these might strengthen the paper:
- At several places, the authors make claims or suggestive remarks about applicability of their work to the “real world CL scenarios”. I think most of these remarks are not well justified. To me, the strength of this paper is predominantly academic in the sense that it provides a proof-of-concept demonstration that replay schedules can be beneficial for continual learning. I consider the real world value of the currently proposed algorithm rather limited. I would like to encourage the authors to discuss their work more in these terms.
- The reference list can be made neater and up-to-date (e.g., for several papers the arxiv version is cited while a published version is available as well);
- There is a small amount of grammatical errors (e.g., the word “data” is treated as singular twice on p4).

---

> ### Author Response · Authors · 2023-08-09
> **Rebuttal Reply: Experimental results on applying MCTS to new seeds, validation dataset assumption, and replay setting**
>
> We thank the reviewer for the valuable feedback and suggestions.
>
>
> > **Main Issue and Requested Change 1: "... demonstrated benefits are due to replaying specific tasks or replaying a specific set of samples."**
>
>
>  We have added results for the suggested experiment with applying the found MCTS schedules on new seeds to show the benefits of replaying specific tasks.
> We run experiments on 5-task Split MNIST and 20-task Split CIFAR-100 in the Task Incremental Learning (Task-IL) setting with same setup as in Table 1 (Section 4.1) with uniform memory selection.
> We select 5 new seeds (seed 10-14) and apply the replay schedules found by MCTS from independent runs separately on the new seeds. We compare against Random scheduling, Equal Task Scheduling (ETS), and Heuristic-GD (Heur-GD) scheduling executed on the same 5 seeds.
>
>
>
> We report the mean and std. of the evaluated ACC and BWT metrics in the table below. MCTS Seed X refers to applying the replay schedule found by MCTS from seed X to the CL experiments with the 5 new seeds.
> We observe that the found MCTS schedules outperform Random and ETS scheduling on Split CIFAR-100 on the 5 new seeds, and mostly exceeds the baselines on Split MNIST.
> Heur-GD is a competitive baseline against the MCTS schedules, especially on Split CIFAR-100.
> However, Heur-GD requires tuning a validation accuracy threshold for every dataset which can be difficult to tune for every new memory size $M$ and memory selection method.
>
>
> These new results further demonstrates the importance of scheduling which tasks to replay in CL to more effectively reduce catastrophic forgetting.
> We will add results for all datasets in the camera-ready version of our paper.
>
>
>
> |            | **Split MNIST**  |             | **Split CIFAR-100** |           |
> |---------------|:----------------:|:----------------:|:-------------------:|:-----------------:|
> | **Method**      | ACC              | BWT              | ACC                 | BWT               |
> | **Random**      | 96.04 $\pm$ 1.23 | -4.67 $\pm$ 1.51 | 51.29 $\pm$ 0.61    | -38.01 $\pm$ 0.61 |
> | **ETS**         | 96.35 $\pm$ 0.62 | -4.30 $\pm$ 0.79 | 46.69 $\pm$ 1.51    | -42.83 $\pm$ 1.66 |
> | **Heur-GD**     | 97.90 $\pm$ 0.47 | -2.31 $\pm$ 0.59 | 57.00 $\pm$ 2.01    | -31.25 $\pm$ 2.08 |
> | **MCTS Seed 1** | 97.71 $\pm$ 1.21 | -2.57 $\pm$ 1.51 | 54.58 $\pm$ 1.18    | -33.65 $\pm$ 1.54 |
> | **MCTS Seed 2** | 97.80 $\pm$ 0.67 | -2.42 $\pm$ 0.82 | 53.86 $\pm$ 1.11    | -34.48 $\pm$ 1.13 |
> | **MCTS Seed 3** | 96.83 $\pm$ 0.58 | -3.61 $\pm$ 0.74 | 53.21 $\pm$ 1.02    | -35.09 $\pm$ 0.81 |
> | **MCTS Seed 4** | 96.12 $\pm$ 2.00 | -4.54 $\pm$ 2.49 | 54.06 $\pm$ 1.69    | -34.52 $\pm$ 1.62 |
> | **MCTS Seed 5** | 97.07 $\pm$ 0.96 | -3.35 $\pm$ 1.19 | 55.23 $\pm$ 2.05    | -33.12 $\pm$ 2.03 |
>
>
>
> > **Smaller Issue 1 and Requested Change 2: "Discuss more clearly that applying replay schedules, which were learned with RL, requires access to validation sets of all past tasks."**
>
>  We agree with reviewer comment and will emphasize in the revised manuscript of our assumption about having access to a validation set in the test environment for computing the RL states.
>
>
>
> > **Smaller Issue 2 and Requested Change 3: "Describe how the replay itself is implemented."**
>
>  The $M$ examples are replayed in every training iteration of the new task. We concatenate the $M$ replay examples with the $B$ current task examples, where $B$ is the batch size. If $M > B$, we randomly sample $B$ replay examples among the $M$ examples and concatenate these to the current task examples. The $B + M$ examples are weighted equally when computing the loss, regardless of which task the examples are from. We will clarify this replay setting in the revised version of the manuscript.
>
>
> > **Minor Issue 1: "Discuss their work more in terms proof-of-concept demonstration rather than applicability to real world CL scenarios"**
>
>  We concur with the reviewer comment. We will adjust
> suggestive remarks on applicability of MCTS and the RL framework to real-world CL settings to avoid confusions, and make the discussions more focused on the benefits of replay scheduling in CL in the revised manuscript.
>
>
> > **Minor Issue 2 and 3: "Update reference list and fix grammatical errors"**
>
>  We will make the reference list more up-to-date by, for instance, citing the published versions rather than Arxiv versions, as well as run a spell checker through the manuscript for the camera-ready version.

---

> > ### Comment · Reviewer_QGFH · 2023-08-11
> > **Concerns addressed**
> >
> > Thank you for the to-the-point rebuttal. These additional experiments satisfyingly address my main concern. I think these are striking results.
> >
> > My smaller concerns are also adequately addressed. Congratulations on a neat and interesting paper.

---

> > > ### Author Response · Authors · 2023-08-11
> > > **Authors to Reviewer QGFH: Thank you**
> > >
> > > Dear reviewer QGFH,
> > >
> > > We are glad to hear that our rebuttal has addressed your concerns satisfyingly. Thank you for your constructive feedback and suggestions for improving our paper.

---

### Author Response · Authors · 2023-08-09
**Overall Response: Summary of Contributions and Revised Manuscript**

We thank all reviewers for their constructive feedback and suggestions. We thank reviewer **tXM7** for finding our paper well motivated and well-written,
reviewer **bt5V** for recognizing the importance of balancing the replay memory in continual learning (CL), and reviewer **QGFH** for finding our contribution intriguing and that it should be of interest for a large group of CL researchers.

We believe that some concerns might be related to misunderstandings regarding our contributions. Therefore, we revisit our contribution below.

**Contributions**

The three main contributions of this work are the following:

1) We challenge the current CL setting and propose a new, and slightly changed, setting where historical data is available at any time while the processing time is limited. This new CL setting aligns better with real-world needs given that new data arrive continuously but companies store their historical data rather than throwing it away.

2) We propose learning the time to learn, i.e., learning which tasks to replay at different times. As an exemplar method, we use MCTS to learn the time to learn and show the benefits of replay scheduling in CL, which has been overlooked by the current literature. However, MCTS has scalability limitations and requires multiple episodes of CL training in any new CL scenario, which is prohibited in CL settings.

3) We take a first step towards enabling replay scheduling for realistic CL settings by proposing a framework using reinforcement learning (RL) for learning general policies that generalize to new CL scenarios. The RL policy is applied to new CL scenarios without additional training in the test environment. We show that the RL policies can mitigate catastrophic forgetting for the CL system in 5-task scenarios with new task orders as well as new datasets, that are unseen during training.

We have revised the manuscript and color-coded the changes in blue accordingly.
Furthermore, we have added references in the right margin to the reviewer who requested the fix or add-on where it was appropriate.

We have replied to each reviewer separately below in the comment section. Please let us know if you have any further questions.

---

### Decision · Action_Editors · 2023-08-27

**Recommendation:** Accept as is

**Comment:**

The paper delves into the scheduling of experience replay for continual learning. All reviewers recognize that this approach offers a new angle on continual learning and could hold value for the community. Initially, reviewers raised concerns about the practicality of the proposed method and the analysis on the sources of its benefits. However, the reviewers found satisfaction in the authors' additional experiments during the rebuttal, leading to unanimous agreement on accepting the paper. The AE aligns with their assessments and consequently recommends accepting the paper as is.

**Audience:**

Meets the interests of audiences.

**Claims And Evidence:**

Well done.